# Meteorological history of low forest greenness events in Europe in 2002−2022

Mauro Hermann[1], Matthias Röthlisberger[1], Arthur Gessler[2,3], Andreas Rigling[2,3], Cornelius Senf[4], Thomas Wohlgemuth[2], and Heini Wernli[1]

[1]Institute for Atmospheric and Climate Science (IAC), ETH Zürich, Zurich, Switzerland
[2]Swiss Federal Institute for Forest, Snow and Landscape Research (WSL), Birmensdorf, Switzerland
[3]Institute of Terrestrial Ecosystems (ITES), ETH Zürich, Zurich, Switzerland
[4]Ecosystem Dynamics and Forest Management Group, Technical University of Munich, Freising, Germany

**Correspondence:** Mauro Hermann (mauro.hermann@env.ethz.ch)

**Abstract.**

Forest dieback in Europe has recently intensified and become more extensive. This dieback is strongly influenced by meteorological variations of temperature, $T2m$, and precipitation, $P$, and can be monitored with forest greenness. This study quantitatively investigates the three-year meteorological history preceding events of reduced forest greenness in Europe's temperate and Mediterranean biome with a systematic approach. A specific focus lies on the timing of unusually persistent and of unusually strong anomalies of $T2m$ and $P$, as well as their relation to synoptic weather systems. A pragmatic approach based on remote sensing observations of normalized difference vegetation index $NDVI$ serves to identify low forest $NDVI$ events at the 50 km scale in Europe in June to August 2002−2022. We quantify the impact of the hottest summer on record in Europe in 2022, which, according to our criteria, negatively affected 37% of temperate and Mediterranean forest regions, and thereby reduced forest greenness more extensively than any other summer in 2002−2022.

The low $NDVI$ events occurred in particularly dry and hot summers but their meteorological histories featured also significant anomalies further in the past, with clear differences between the temperate and Mediterranean biome. A key feature is the anomalous accumulation of dry periods (i.e., periods with a $P$ deficit) over the preceding 26 and 34 months in the temperate and Mediterranean biome, respectively. In the temperate biome only, $T2m$ was anomalously persistent during almost the same 26-month period and featured distinctive peaks late in the past three growing seasons. While anomalously strong hot-dry conditions were characteristic for temperate low $NDVI$ events already in the previous summer, we find hardly any other systematic meteorological precursor in the Mediterranean prior to the event year. The identified dry periods went along with reduced cyclone activity in the Mediterranean, and positive anticyclone frequency in the temperate biome, respectively. The occurrence of these two weather systems is locally more nuanced, showing, e.g., consistently increased and decreased cyclone frequency over western and northern Europe, respectively, in all event summers. Finally, the systematic meteorological histories are useful to test whether locally observed meteorological impacts, e.g., structural overshoot, systematically influenced the investigated events. In summary, systematic investigations of the multi-annual meteorological history provided clear evidence of how surface weather and synoptic-scale weather systems over up to three years can negatively impact European

forest greenness. The observation of the record-extensive low $NDVI$ event in summer 2022 underlines that understanding the forest-meteorology interaction is of particular relevance for forest dieback in a changing climate.

## 1 Introduction

European forest ecosystems have typically been in balance with their climatic environment and are, thus, largely adapted and acclimated to meteorological variability on a larger scale. This balance is increasingly disturbed by anthropogenic climate change (Seidl et al., 2017; McDowell et al., 2020). As a consequence, the exposure and vulnerability of European forests to climatic disturbances has increased in the recent two decades and forests were affected on up to the continental scale (Seidl et al., 2014; Bastos et al., 2020b). On that scale, drought has been a key driver of excess forest mortality in Europe (Senf et al., 2020). In addition to meteorological drought, i.e., reduced precipitation, high temperatures enhance the atmospheric water demand and, thus, water loss from the vegetated surface by evapotranspiration (Yuan et al., 2019; Grossiord et al., 2020). Heatwaves (Meehl and Tebaldi, 2004; IPCC, 2021), meteorological droughts (Trenberth et al., 2014; IPCC, 2021), and compound hot droughts (Allen et al., 2015) are expected to intensify future forest dieback (Brodribb et al., 2020). Such meteorological conditions, for example during 2003 and 2018, led to stem dehydration (Salomón et al., 2022), reduced forest growth (Ciais et al., 2005; Trotsiuk et al., 2020), hydraulic failure (Schuldt et al., 2020), and ultimately increased tree mortality in Europe (Allen et al., 2010; Hansen et al., 2013). Although some forest ecosystems are adapted to low water availability in summer, e.g., in the summer-dry Mediterranean, or throughout the year, e.g., in dry inner-Alpine regions, extended and more frequent hot droughts will strongly affect their dynamics including growth and survival (Rigling et al., 2013; Tague et al., 2019; Ogaya et al., 2020). In this context, forest greenness as measured by satellites is an effective measure related to forest vitality to monitor the recent forest dieback in Europe (Orth et al., 2016; Buras et al., 2020, 2021).

While forests can endure short-term weather extremes (e.g., an individual multi-day heatwave), they are more susceptible to longer-term extreme conditions. Particularly harmful long-term extremes include persisting or sequentially occurring droughts, whereby additional negative legacy effects are mediated through reduced tree resilience (Anderegg et al., 2015, 2020; Bose et al., 2020). However, drought legacy effects have also been suggested to provide acclimation to following droughts (Gessler et al., 2020). Furthermore, a stormy winter followed by a hot-dry growing season allows bark beetles to spread out and attack damaged and dying trees (Temperli et al., 2013; Biedermann et al., 2019; Jakoby et al., 2019). Additionally, persisting heat, and/or precipitation deficits can – given fuel availability – trigger forest fires year-round, occurring most intensely in the Mediterranean (Turco et al., 2017). In strongly fuel-limited regions, however, forest fires can in the long run negatively feed back on fire activity (Pausas and Ribeiro, 2013). Lastly, beneficial conditions in the past can exert a negative legacy on forest vitality in following dry periods through soil moisture depletion and structural overshoot, i.e., an excessive canopy buildup relative to average climatic conditions (Bastos et al., 2020a; Zhang et al., 2021). These examples highlight that in addition to the (co-)occurrence, magnitude, and duration of heat and drought, the position of such meteorological precursors in the

longer-term history likely modulates their impact on forest greenness.

In the context of interaction and legacy effects the point in time when such meteorological precursors are relevant to forest greenness is intensively discussed in the literature. Meteorological impact has often been investigated by considering the mean over the current growing season (e.g., Hlásny et al., 2017; Seftigen et al., 2018; Seidl et al., 2020). Senf et al. (2020) revealed that hot-dry conditions are particularly harmful for canopy mortality in March to July. Neumann et al. (2017) identified warm summer temperatures and high variability in seasonal precipitation as meteorological drivers of tree mortality. In a drought-prone region, forest drought stress was about equally determined by temperature in summer and the previous autumn, and precipitation during the cold season (Williams et al., 2013). More generally, drought-induced partial or complete tree mortality shows a threshold behaviour (Brodribb et al., 2020; Senf et al., 2020); however, water stress is not equally harmful at all times. Especially outside or at the bounds of the growing season there are complicating factors such as growth compensation, soil moisture coupling, and snow melt (e.g., Harpold et al., 2015; Bastos et al., 2020a).

The meteorological processes that are relevant for low forest greenness cover a wide range of timescales. Longer-term meteorological extremes are of particular interest, however, they are typically composed of multiple shorter-term anomalies, which are not necessarily extreme at their respective timescale (Röthlisberger et al., 2020). On the timescale of about $3-10$ days, atmospheric blocking can induce surface heatwaves and suppress precipitation by large-scale subsidence and high solar insolation (Pfahl and Wernli, 2012a; Zschenderlein et al., 2019). On somewhat longer time scales, recurrent and quasi-stationary Rossby wave patterns may also lead to co-occurring hot and dry conditions (Wolf et al., 2018; Röthlisberger and Martius, 2019; Ali et al., 2021). In central Europe, summer heatwaves can also arise from weak synoptic forcing, which, in combination with a Scandinavian blocking, allowed for widespread hot-dry conditions in 2018 (Spensberger et al., 2020). During heatwaves with no or reduced precipitation, the soil moisture-atmosphere coupling exacerbates the near-surface warming over drying soils (Fischer et al., 2007; Seneviratne et al., 2010). Accordingly, and especially in the Mediterranean, an extremely hot summer is more likely in years of a winter/spring precipitation deficit (Russo et al., 2019). As Europe hardly experiences drought over a longer (multi-annual) time scale (Schubert et al., 2016), seasonal meteorology, which is strongly linked to weather system dynamics, is of particular interest for forest greenness, and, therefore, in the focus of the present study.

Despite great progress in understanding the eco-hydraulic mechanisms linking drought to events of reduced forest greenness (Brodribb et al., 2020), a systematic analysis of the meteorological history of such events is still lacking. The purpose of this study is to systematically document and characterize significant aspects of these meteorological histories in Europe's temperate and Mediterranean forests. Specifically, this study seeks to identify meteorological precursors over the three years prior to reduced forest greenness in Europe. Hereby "precursors" are features in the meteorological histories that occur at a statistically significantly higher rate preceding reduced forest greenness events than in climatology, and that are shared among many events. We focus on the evolution of 90-day moving average 2-m temperature ($T2m_{90d}$) and precipitation ($P_{90d}$) as key variables and quantitatively address the following research questions: (1) When and how deviated $T2m_{90d}$ and $P_{90d}$ significantly from

climatology during the meteorological history of low forest greenness events in Europe? (2) Which anomalies in weather system frequencies went along with the meteorological precursors identified in (1)? To identify low forest greenness events, generally characteristic for low productivity crown defoliating and tree mortality (Buras et al., 2021), we use persistently low values of the normalized difference vegetation index ($NDVI$) in summer $2002-2022$ (Sects. 2.2 & 2.3). Based on a sub-sampling of the resulting low $NDVI$ events in combination with a bootstrapping test, we then identify meteorological precursors along their meteorological history, and further investigate the spatial variation of weather system frequencies in Sect. 3. Finally, we critically discuss our results and the limitations of our analyses in Sect. 4.

## 2 Data and methods

We differentiate broadly between the temperate and Mediterranean biome according to Schultz (2005) in the domain extending from $10°$ W to $45°$ E and $35°$ N to $65°$ N, excluding all boreal forests (Fig. 1a).

### 2.1 Forest cover data

Land surface cover observations are available from the Corine Land Cover (CLC) data set from the Copernicus Land Monitoring Service at $100\,$m horizontal resolution from the year 2012 (Büttner et al., 2004). For comparison with the other data sets introduced below, we first interpolate the surface cover to $250\,$m resolution by nearest-neighbor interpolation. Following that, we mask all pixels except forest land cover classes (coniferous, broad-leaved, and mixed forest) to retain only forest pixels. We then coarse grain fractional forest area to $0.5° \times 0.5°$ grid cells, which is the spatial resolution of the meteorological data set used here (ERA5, see below), hereafter denoted as $FA^{0.5}$. For our analysis we only consider grid cells with a significant fraction of forest cover, defined here as $FA^{0.5} \geq FA^{min} = 10\%$, and hereafter refer to these grid cells as "forest grid cells". According to this definition, there are 1'260 and 544 forest grid cells in the temperate and Mediterranean biome, respectively (Fig. 1a).

### 2.2 Normalized difference vegetation index

We use the 16-daily normalized difference vegetation index ($NDVI$) at $\sim 250\,$m horizontal resolution from March 2002 to August 2022 from NASA MODIS Terra (Didan, 2015). As mentioned in Sect. 2.1, we only use $NDVI$ at forest pixels according to CLC in order to minimize noise from other land cover types. The $NDVI$ is based on the red ($RED$) and near-infrared ($NIR$) spectral irradiance:

$$NDVI = \frac{NIR - RED}{NIR + RED} \tag{1}$$

The greener a forest pixel is, the closer its $NDVI$ is to +1 (Tucker, 1979). The $NDVI$ serves as a measure of vegetation greenness and has previously been used to assess drought impact on ecosystems (Anyamba and Tucker, 2012; Orth et al., 2016; Buras et al., 2020). The Application for Extracting and Exploring Analysis Ready Samples (AppEEARS; https:

 //appeears.earthdatacloud.nasa.gov) additionally provides MODIS pixel quality. In addition to masking non-forest land cover, we mask $NDVI$ values that are of poor quality due to snow and clouds, and only retain $NDVI$ values with good and marginal quality according to MODIS pixel quality. The resulting $NDVI$ time series contain missing values, which we linearly interpolate from neighbouring time steps as in Buras et al. (2021). Finally, we perform a linear detrending of the entire time series as in Buras et al. (2020) due to a detected greening trend (Bastos et al., 2017).

After this post-processing, at every pixel $j$ at time step $t$ in year $n$ we consider $NDVI$ anomalies ($NDVI'_{j,t,n}$) from the median in June−August (JJA). To later compare anomalies at different pixels, we standardize the anomalies by the local inter-quartile range $IQR_j(NDVI)$:

$$NDVI'_{j,t,n} = \frac{NDVI_{j,t,n} - \overline{NDVI_j}}{IQR_j(NDVI)} \tag{2}$$

The climatological median $\overline{NDVI_j}$ and $IQR_j(NDVI)$ are both calculated from all 126 $NDVI$ anomalies in JJA 2002−2022.

## 2.3 Identification of low $NDVI$ grid cells

The aim of the approach presented here is to identify persistently low $NDVI$ in JJA at the relatively large $0.5°$ scale of the meteorological reanalysis data, i.e., wide-spread low $NDVI$. In essence, the approach (1) considers all forest pixels $j$ within a forest grid cell $J$ and flags them if at least four out of the six $NDVI'$ values in JJA are negative, and (2) identifies an event

at forest grid cell $J$ (i.e., a $0.5° \times 0.5°$ grid cell with at least 10% forest cover) if at least 80% of forest pixels inside $J$ are flagged (Fig. 1b). Our identification scheme thus features three tuning parameters: (1) the $FA^{min}$, (2) the minimum count of negative $NDVI'$ per JJA at the pixel level ($c_{ev}^{min}$), and (3) the fraction of affected forest pixels per forest grid cell. An extensive sensitivity analysis to reasonable variations in these parameters is presented in Appendix A. Hereafter we detail the technical implementation of the approach.

In the first step (1 in Fig. 1b), we count the number of negative $NDVI'_{j,t,n}$ values for the six 16-daily values during JJA in year $n$ ($c_{n,ev}$). At every forest pixel $j$, an event flag $ev_{j,n}$ is determined according to the following criterion:

$$ev_{j,n} = \begin{cases} 1 & \text{if } NDVI'_{j,t,n} < 0 \text{ for } c_{n,ev} \geq c_{ev}^{min} = 4, \\ 0 & \text{otherwise.} \end{cases} \tag{3}$$

In the second step (2 in Fig. 1b), the total area of forest pixels with $ev_{j,n} = 1$ in $J$ ($A_{J,n}^{ev}$) has to be at least 80% of the total
145 forest pixel area in $J$ ($A_J^{for}$) - for which we use the term minimum affected ratio $AR^{min} = 80\%$ hereafter:

$$EV_{J,n} = \begin{cases} 1 & \text{if } A_{J,n}^{ev} \geq 0.8 A_J^{for}, \\ 0 & \text{otherwise.} \end{cases} \tag{4}$$

Lastly, for the identified low $NDVI$ grid cells, we calculate a measure of event intensity as the average of JJA minimum $NDVI'_{j,t,n}$ over all flagged forest pixels ($ev_{j,n} = 1$), and, hereafter, termed $NDVI'^{min}_{J,n}$. Subscripts are omitted whenever possible without loss of clarity.

The resulting low $NDVI$ grid cells ($EV_{J,n} = 1$) are tested for their sensitivity to the three threshold parameters $AR^{min}$, $FA^{min}$, and $c^{min}_{ev}$ in Appendix A. Our choice of $AR^{min} = 80\%$ and $FA^{min} = 10\%$ was guided by compromising sufficient low $NDVI$ grid cells for statistical evaluation and reasonable peculiarity of the low $NDVI$ events, which represent a form of extreme event. Furthermore, the main results of this study demonstrate a very low sensitivity to variation in these two threshold parameters, as to a reduction in $c^{min}_{ev}$. A substantial change of the identified low $NDVI$ grid cells results only when increasing $c^{min}_{ev}$ to five, i.e., almost uninterruptedly negative $NDVI$ in JJA. While these most extreme events would be worth studying, a robust statistical evaluation thereof would not be possible as the number of low $NDVI$ grid cells diminishes drastically (by almost a factor of ten).

## 2.4 ERA5 reanalysis data

Atmospheric fields are used from ERA5 reanalysis from the European Centre for Medium-Range Weather Forecasts (ECMWF, Hersbach et al., 2020) available hourly on 137 vertical levels and on a regular grid with 0.5° horizontal resolution.

### 2.4.1 Normalized meteorological 90-day anomalies

Our analyses focus on 90-day moving average values of 2-m temperature ($T2m_{90d}$), total precipitation ($P_{90d}$), cyclone frequency ($f_{90d}(C)$), and anticyclone frequency ($f_{90d}(A)$). The standard ERA5 variables $T2m_{90d}$ and $P_{90d}$ can directly be averaged, while cyclones and anticyclones are first identified from hourly sea level pressure ($SLP$) fields. These two most central weather systems are of interest to the low $NDVI$ events' meteorological history as they not only determine vertical and horizontal atmospheric transport of heat, momentum, and moisture, but are further of great importance to the temperature and humidity structure of the atmosphere. Cyclones (anticyclones) are identified according to Wernli and Schwierz (2006) and Sprenger et al. (2017) as objects of low (high) $SLP$ and are, hence, identified from the outermost closed $SLP$ isoline around local $SLP$ minima (maxima). From these hourly object masks we then calculate weather system frequencies over ninety days, i.e., $f_{90d}(C)$ and $f_{90d}(A)$. For all four variables, we calculate 90-day mean values as a right-aligned moving average. Each 90-day mean value, therefore, is labelled by the time step of the last value that contributes to the average. Leap days are discarded from the analysis to maintain consistency in each calendar day's climatology and the length of the meteorological histories. The climatologies of the four variables cover 90-day moving averages from 1 September 2001 to 31 August 2022.

Based on these 90-day mean values, we compute normalized anomalies at every forest grid cell for variables $X = T2m_{90d}$, $P_{90d}$, $f_{90d}(C)$, and $f_{90d}(A)$ as follows:

$$X'_{90d} = \frac{X_{90d} - \overline{X_{90d}}}{\sigma_{X_{90d}}} \tag{5}$$

where $X'_{90d}$ denotes the normalized anomaly, $\overline{X_{90d}}$ and $\sigma_{X_{90d}}$ denote the climatological seasonal mean and standard devia-
tion in the considered 21 years, respectively, i.e., are calculated over 21 values. Note that the normalization of $X_{90d}$ anomalies
is used merely for scaling with local variability. The scaling enables the spatiotemporal comparison of these anomalies and is
not used to estimate the anomalies' return period or likelihood.

For better interpretability of individual meteorological histories, we express $f_{90d}(C)$, and $f_{90d}(A)$ also as anomalies relative
to the climatological mean. These relative anomalies are calculated as follows, e.g.,:

$$f'^{rel}_{90d}(C) = \frac{f_{90d}(C) - \overline{f_{90d}(C)}}{\overline{f_{90d}(C)}} \tag{6}$$

### 2.4.2 Significance assessment

We conduct a bootstrapping test to identify statistically significant meteorological precursors that are shared among the low
$NDVI$ grid cells. The details of how the test is conducted are described in Appendix B. Broadly, the bootstrapping produces
1'000 synthetic samples of meteorological histories with a sample size equal to the number of low $NDVI$ grid cells under
consideration. These samples correspond to many realizations of meteorological histories that are expected in the climatological
reference period of 2002−2022, and are used to construct the null distributions for our statistical tests. We test the following
null hypothesis $H_{0,EV}$ at different time lags $\Delta t$ prior to the event time $t_{ev}$ with a significance level of $\alpha = 5\%$:

$H_{0,EV}$: The meteorological history at $t_{ev} - \Delta t$ is equal to a randomly sampled meteorological history.

We use different test statistics (with corresponding null distributions) to investigate different aspects of the meteorological
history under consideration. These statistics are the sample mean $T2m'_{90d}$, $P'_{90d}$, $f'_{90d}(C)$ and $f'_{90d}(A)$, respectively, and the
fraction of $\Delta t$ that is on average covered by warm ($T2m'_{90d} > 0$) and dry periods ($P'_{90d} < 0$), respectively. Moreover, our statis-
tical procedure is designed to retain the spatial correlation of the original meteorological fields (for details see Appendix B). In
the bootstrapping test, p-values are estimated from the percentiles of the 1'000 reference values. Values of the meteorological
history under consideration that lie outside the range of the 1'000 corresponding synthetic values of the reference meteorolog-
ical histories receive a p-value of 0 (Röthlisberger et al., 2016). We reject the above null hypothesis if an observed value lies
outside the $2.5^{th}$ to $97.5^{th}$ percentile of the null distribution (defined by the 1'000 values obtained from the bootstrapping).
That is, we reject the null hypothesis at the 5% significance level. At time lags $\Delta t$ when the meteorological history of the low
$NDVI$ grid cells lies outside the confidence interval, $H_{0,EV}$ is rejected for time lag $\Delta t$.

## 3 Results

### 3.1 Low $NDVI$ events in JJA 2002−2022

Low $NDVI$ events covered substantial parts of both biomes in JJA 2002−2022, and were by far the most frequent in 2022
(Fig. 2a,d; Appendix C). In the temperate biome, the years with most low $NDVI$ grid cells - in descending order - were 2022,

2019, 2018, and 2020 (Fig. 2a,d). Noteworthy, these four years all lie in the last five years of the study period. In 2022, 37%
of temperate forests were affected by the low $NDVI$ event, which far exceeded the previous record years 2019 and 2018 by
13% and 15%, respectively (Fig. 2d). The top years in terms of affected forest grid cells in the Mediterranean biome were
2022, 2008, 2005, and 2007, again sorted by decreasing area affected. Low $NDVI$ grid cells were on average almost twice as
frequent in the Mediterranean biome (9% yr$^{-1}$) compared to the temperate biome (5% yr$^{-1}$; Fig. 2d). Lastly, most low $NDVI$
events in each biome go along with increased disturbance area as measured by forest canopy mortality according to Senf and
Seidl (2021a, Appendix D). More specifically, in the overlap period of the two data sets, most low $NDVI$ grid cells are among
the top four and five ranks regarding the disturbance area in temperate and Mediterranean forests, respectively.

Mediterranean forests faced more but typically less intense events, as measured by the grid cell-wide average of summer
minimum $NDVI'$ (Fig. 2b, Sect. 2.3). $NDVI_J'^{min}$ was usually between $-1$ and $-2$, whereas it was about 0.5 lower for
temperate forests. Hot-spot regions with three or more events during the 21 years were northeastern Germany, the Balkans, and
large parts of the Mediterranean biome (Fig. 2c). In the latter, we find many low $NDVI$ grid cells in Spain in 2005, in Turkey
in 2007 and 2008, in Italy in 2017, and in 2022 in the northern Mediterranean (Appendix C). Central Europe was largely
affected in the past five years, while the 2018$-$2019 events extended further to Scandinavia and the Baltic, and 2020 and 2022
affected also parts of southern France and eastern Europe, respectively. Note that 26% of the forest regions in the study domain
never experienced an event in JJA 2002$-$2022 (Fig. 2c). These grid cells are in northeastern Europe or in mountainous regions
including the Alps, the Carpathians, the southern Dinaric Alps, and the Eastern Black Sea Mountains.

### 3.2 Examples of meteorological histories

Low $NDVI$ events affected different forest regions all over Europe with varying intensity, each with its own meteorological
history. We first present three examples of low $NDVI$ events that affected regions in Spain in 2005 (SPA05), in the Balkans
in 2013 (BAL13), and in France in 2022 (FRA22; Fig. 3). First, these examples illustrate that the low $NDVI$ events identified
in this study not necessarily featured a very low $NDVI_J'^{min}$, i.e., a strong magnitude of negative $NDVI'$. For example, some
grid cells of the SPA05 region were identified as low $NDVI$ grid cells with $NDVI_J'^{min}$ just above $-1$, while others with
$NDVI_J'^{min} < -1.75$ in northern Spain were not identified (Fig. 3a,d). In fact, this is an expected behaviour of our identifica-
tion scheme as low $NDVI$ grid cells are meant to indicate that a very large fraction of forest pixels in that grid cell experienced
persistently low $NDVI$ (Sect. 2.3). In many cases, however, event intensity as a JJA-integrated quantity was increased at low
$NDVI$ grid cells, compared to their event-unaffected surrounding, illustrated at the example of France in 2022 (Fig. 3c,f).
Lastly, another interesting case (not shown) occurred in 2014 in Slovenia, where an ice storm in DJF-6m caused a few low
$NDVI$ grid cells (Appendix C; Buras et al., 2021; Senf and Seidl, 2021c).

We now introduce the concept of a three-year meteorological history prior to low $NDVI$ events for these three examples.
For SPA05, the meteorological history was characterized by a shift from a precipitation surplus during SON-21m to JJA-12m
to a precipitation deficit during the year prior to the low $NDVI$ event (Fig. 4a). Note that we here use a notation for seasons

(e.g., SON-21m) that indicates their negative time lag to the events (i.e., SON-21m is the autumn 21 months prior to the low $NDVI$ event summer, termed JJA-ev). While the cyclone frequency was more than doubled during the former wet period, the period with negative $P'_{90d}$ featured almost no cyclones at all. The relative cyclone frequency anomaly $f'^{rel}_{90d}(C)$ was most negative ($-50\%$ to $-100\%$) in the climatological cyclone season from SON to MAM (Fig. 4a). For example in early MAM-3m, $P'_{90d} = -1.5$ coincided with $f'^{rel}_{90d}(C)$ of close to $-75\%$. The meteorological history of SPA05 further featured strong cold spells in MAM-15m to JJA-12m and in DJF-6m to MAM-3m with $T2m'_{90d}$ as low as $-2.6$ (Fig. 4b). Warm periods occurred in JJA-ev and DJF-18m and coincided with an increased frequency of anticyclones, i.e., positive $f'^{rel}_{90d}(A)$. In the second example, BAL13, the magnitude of negative $P'_{90d}$ was also a dominant feature of the meteorological history with even more distinctive persistence over the three years (Fig. 4c). Similar to SPA05, a precipitation deficit was often related to negative $f'^{rel}_{90d}(C)$ with some exceptions, e.g., in SON-9m. JJA-12m was characterized by the largest $T2m'_{90d} = +2.9$ of all examples and neither co-incided with substantial changes in $f'^{rel}_{90d}(A)$ or in $f'^{rel}_{90d}(C)$ (Fig. 4c,d). Despite continuously positive $T2m'_{90d}$ from JJA-12m to JJA-ev, the meteorological conditions became less dry and less hot than in JJA-12m, when, interestingly, the BAL13 region was mostly unaffected by low $NDVI$ events (Appendix C). The most recent event FRA22 had again a different meteorological history. It stands out with anomalously high $T2m'_{90d}$ over the six months preceding the event that was related to negative $f'^{rel}_{90d}(A)$ over considerable portions of that period (Fig. 4f). Moreover, $P'_{90d}$ during most of these six months was only slightly negative, and strongly positive when going further back in the meteorological history, e.g., in JJA-12m (+2.6; Fig. 4e). One last noteworthy disparity of FRA22 compared to BAL13 is that JJA-12m was persistently colder alongside negative $f'^{rel}_{90d}(A)$.

While they are illustrative, these exemplary meteorological histories of SPA05, BAL13, and FRA22 reveal great variability and clearly do not allow to draw any causal inferences about how certain aspects of these histories alter the likelihood of low $NDVI$ events. The events' meteorological histories share certain characteristics but also clear disparities emerge, for example, $P'_{90d}$ in JJA-ev or $T2m'_{90d}$ in the last year before the event. In the next section we thus systematically analyze the meteorological history of all identified low $NDVI$ events, and use our sub-sampling and bootstrapping procedure to identify statistically significant meteorological precursors to these events. Again recall that these precursors are features of the low $NDVI$ events' meteorological histories that were significantly more frequent than during any random meteorological history in the climatology.

### 3.3 Systematic meteorological precursors of low $NDVI$ events

The previous two sections have illustrated that (i) low $NDVI$ events were unequally distributed over the study period - especially in temperate forests -, and (ii) a more systematic analysis of meteorological histories is needed to assess their relevance for the low $NDVI$ grid cells. To account for the uneven distribution of events across years, we investigate the average meteorological history of a random sub-sample of low $NDVI$ grid cells in the temperate and Mediterranean biome separately. The sub-sample includes a maximum of ten randomly selected low $NDVI$ grid cells per year. Given the 21-year-long study period, there is a maximum of 210 contributing low $NDVI$ grid cells. For years with less than or exactly ten low $NDVI$ grid cells, all events of the respective year contribute to the sub-sample. The resulting average meteorological history of the

temperate and Mediterranean biome is a mean over 170 and 164 low $NDVI$ grid cells, respectively (see Appendix A). Due to the randomness involved in the sub-sampling, we repeat the procedure $n_{samp} = 10$ times to create a variety of average meteorological histories that account for the variability in years when many low $NDVI$ grid cells were identified. Our bootstrapping test (significance level $\alpha = 5\%$) is then applied to the anomalies of the averaged meteorological histories to identify statistically significant meteorological precursors of the low $NDVI$ events in $2002-2022$ (Sect. 2.4.2). In the first part of this section, we analyse the magnitude of $T2m'_{90d}$, $P'_{90d}$, $f'_{90d}(C)$, and $f'_{90d}(A)$ during the three years leading to low $NDVI$ events - similar to Fig. 4. Second, we investigate the persistence of dry ($P'_{90d} < 0$) and hot ($T2m'_{90d} > 0$) meteorological anomalies.

### 3.3.1 Magnitude of meteorological anomalies

The most remarkable meteorological precursors of low $NDVI$ events in the temperate biome are significantly negative $P'_{90d}$ and positive $T2m'_{90d}$ during the four months preceding the events (bold lines in Fig. 5a,c). Note that the 90-day anomaly four months prior to the event represents the conditions during the six to four months prior to JJA-ev, i.e., the time instances denoted here and in the following mark the end of the anomalous time periods. Furthermore, the meteorological history of low $NDVI$ events in temperate forests showed significantly reduced $P'_{90d}$ in JJA-12m (Fig. 5a). In between JJA-12m and JJA-ev, $P'_{90d}$ remained mostly negative but not with a statistically significant magnitude. Similar to $P'_{90d}$, also for $T2m'_{90d}$ significant signals along the meteorological history were always of the same sign (positive in the case of $T2m'_{90d}$). Further warm peaks occurred in MAM-3m, in SON-9m, in JJA-12m, in SON-21m, and in DJF-30m (Fig. 5c). Many of the highlighted periods when the magnitude of meteorological conditions were unusual coincided with larger spread of the ten low $NDVI$ grid cell sub-samples. For example, $P'_{90d}$ in JJA-12m ranged between $-0,2$ and $-0,4$, indicating that the years with many low $NDVI$ grid cells (e.g., 2022, 2019, and 2018) showed increased variability at that point in time (Fig. 5a). That is, some $NDVI$ grid cells showed a stronger, and others a weaker or no precipitation deficit in JJA-12m, respectively.

In Mediterranean forests, the magnitude of hot and dry anomalies in JJA-ev were comparable to those in temperate forests, with the difference that significantly negative $P'_{90d}$ emerged already eight months before low $NDVI$ events during DJF-6m (Fig. 5b,d). One more dry period in DJF-18m was significantly different from climatology, as was a wet and warm anomaly in DJF-30m (Fig. 5b). Apart from the few mentioned anomalies, the meteorological anomalies further back than three seasons (before SON-9m) were within the variability expected from the climatology. Similar to temperate forests, the uncertainty induced by the random sub-sampling is larger when anomalies were of greater magnitude. This applies in particular for the anomalies preceding the low $NDVI$ events by more than one year.

Most of the highlighted anomalies in surface meteorology went along with significant anomalies in the occurrence frequency of weather systems (Fig. 5e-h). Note that we here explore the median history of $f'_{90d}(C)$ and $f'_{90d}(A)$, i.e., of normalized anomalies that are comparable across space and time, and not of $f'^{rel}_{90d}(C)$ and $f'^{rel}_{90d}(A)$ as in Sect. 3.2, which are more meaningful in a local context (Sect. 2.4). In the temperate biome, the hot-dry conditions leading up to JJA-ev related to a concurrent

positive $f'_{90d}(A)$ (Fig. 5g). Already in JJA-12m, negative $P'_{90d}$ went along with significantly increased $f_{90d}(A)$. These positive anomalies thereby occurred in the season when anticyclones are climatologically the least frequent. Further back in time, negative $f'_{90d}(C)$ in MAM-15m had no direct connection to anomalies in surface meteorology, and the DJF-30m warm anomaly related to persistently negative $f'_{90d}(A)$. In the Mediterranean biome, there were four periods of significantly negative $f'_{90d}(C)$: two of them, in DJF-18m and during the 8-month long dry period before the event, coincided with $P'_{90d} < 0$ (Fig. 5f). Note that these were during periods when cyclones are climatologically the most frequent. Moreover, in the drier than usual DJF-18m, $f'_{90d}(A)$ was persistently positive (Fig. 5h). To summarize, significant changes in weather system frequencies often occurred simultaneously with the time periods when the magnitude of $P'_{90d}$ and $T2m'_{90d}$ were identified as meteorological precursors of low $NDVI$ grid cells.

The systematic assessment of meteorological histories across all low $NDVI$ grid cells in $2002-2022$ has revealed several meteorological precursors of low $NDVI$ events in temperate and Mediterranean forests. In addition to the mere magnitude of $T2m'_{90d}$ and $P'_{90d}$, some of these anomalies have co-occurred with a significantly altered frequency of cyclones and/or anticyclones. At the biome scale, anticyclones went along with locally drier conditions across all low $NDVI$ grid cells, e.g., in JJA-ev and in JJA-12m (temperate), and in DJF-18m (Mediterranean). Further, the lack of cyclones in the Mediterranean in their climatological peak season linked to significantly reduced $P'_{90d}$. Figure 5 reveals that some $T2m'_{90d}$ and $P'_{90d}$ signals were not significantly unusual in their magnitude, however, seem to have been of unusual persistence. For example, warm anomalies in temperate forests were hardly ever interrupted during the entire meteorological history (Fig. 5c). Therefore, we next analyze the persistence of dry and warm anomalies.

### 3.3.2 Persistence of dry and warm periods

The previous section pointed to not necessarily intense but unusually persistent dry and warm periods, which we investigate in more detail in Fig. 6. To do so we take the data displayed in Fig. 5 and compute the fraction of positive and negative $T2m'_{90d}$ and $P'_{90d}$, respectively, from a single 90-day average to three years prior to the low $NDVI$ events. This is done, again, for the ten sub-samples individually, of which we calculate a median time series. Moreover, analogously to Fig. 5, we contrast the respective fractions to values expected under the null hypothesis $H_{0,EV}$ that the fraction of dry/warm periods preceding the low $NDVI$ events (i.e., during $\Delta t$) was not different from a randomly sampled meteorological history (grey shading in Fig. 6; Sect. 2.4.2).

In the temperate biome, when going back more than 8-months prior to low $NDVI$ events, the persistence of warm and dry anomalies was each a statistically significant meteorological precursor to low $NDVI$ grid cells in $2002-2022$ (Fig. 6a,c). While $T2m'_{90d}$ and $P'_{90d}$ prior to this period were only shortly of significant magnitude (Fig. 5a,c), their persistence was unusual farther back along the meteorological history. When considering two years before events, dry periods accounted for $82-99\%$ of that time period, where the range is spanned up by the ten random sub-samples, which is significantly more than the climatological expectation (Fig. 6a). Similarly, the persistence of warm anomalies during $81-90\%$ of the preceding two

years were significantly anomalous (Fig. 6c) for nine out of the ten sub-sample meteorological histories (not shown). When considering all ten sub-samples separately, we find that the persistence of dry periods and warm periods was significantly different from climatology at least over 26 and 25 months prior to low $NDVI$ events, respectively. This is also the integration period when these two meteorological precursors were most distinct - i.e., when $H_{0,EV}$ is most clearly rejected. So an accumulation of both warm and dry periods over the about two previous years were peculiarities of events in the temperate biome.

In the Mediterranean biome, warm periods were more frequent than usual but not significantly so compared to the reference climatology (Fig. 6d), which is an interesting contrast to the temperate biome. None of the ten sub-samples indicates that positive $T2m'_{90d}$ were unusually persistent for any $\Delta t$ during the meteorological history. Similar to the temperate biome, dry periods accumulate to a highly unusual degree when going back more than eight months (Fig. 6b). The lowest p-value is reached 28 months prior to events, when dry periods persisted over $81-95\%$ of the time afterwards. Again considering each of the 10 sub-samples individually, we conclude that the persistence of dry periods was increased over at least 34 months preceding low $NDVI$ events, which is longer than for low $NDVI$ events in temperate forests. In contrast, the persistence of warm conditions does not emerge as a significant precursor to low $NDVI$ events in the Mediterranean.

### 3.4 Spatial patterns of weather system anomalies

Additional to the biome-wide averages shown in Fig. 5e-h, anomalies in weather system frequencies exert some typical spatial patterns along the meteorological history, which we illustrate at the example of the past year prior to JJA-ev. Based on our set of low $NDVI$ grid cells, we identify common patterns in the anomalies of weather system frequencies in the grid cells' meteorological history. Note, however, that the robustness of such an analysis is inherently low at the local scale due to the rarity of low $NDVI$ events (Sect. 3.1). Figure 7 shows the consistency in sign of $f''^{rel}_{90d}(A)$ and $f''^{rel}_{90d}(C)$ for the 90-day periods of approximately JJA-ev (Fig. 7a,b) and MAM-3m (Fig. 7c,d), respectively, for all forest grid cells that experienced at least two low $NDVI$ events in $2002-2022$. The results for DJF-6m and SON-9m are shown in Appendix E.

In the temperate biome, changes in weather system frequencies that were consistent among most or all low $NDVI$ events at one location occurred in $66-81\%$ and in $54-64\%$ of the considered forest grid cells in JJA-ev and MAM-3m, respectively. Most prominently, northeastern Europe showed a consistent increase of anticyclones and decrease of cyclones, respectively, in JJA-ev (Fig. 7a,b). Negative $f''^{rel}_{90d}(C)$ was consistent among all low $NDVI$ events at the southern edge of the storm track, i.e., south of the region of climatologically high $\overline{f_{90d}(C)}$ (Fig. 7b). In MAM-3m, however, positive $f''^{rel}_{90d}(C)$ occurred simultaneously with positive $f''^{rel}_{90d}(A)$ in Germany and parts of northwestern Europe. Positive $f''^{rel}_{90d}(A)$ in northern Europe was not only prevalent in JJA-ev, but also in MAM-3m (Fig. 7a), in DJF-6m, and partly in SON-9m (Fig. E1). In MAM-3m this signal further extended towards the Balkans (Fig. 7b). Southern France showed an opposite signal of increased cyclone frequency and negative $f''^{rel}_{90d}(A)$ in JJA-ev. Farther back in the meteorological history in DJF-6m and SON-9m $f''^{rel}_{90d}(C)$ was negative for most or all of the low $NDVI$ events in $56-58\%$ of the considered forest grid cells (Fig. E2). To summarize, positive $f''^{rel}_{90d}(A)$ and negative $f''^{rel}_{90d}(C)$ were prominent features in northern Europe in the warm and cold season, respectively. Other regions,

such as southern France, show interesting differences to this prominent signal.

The Mediterranean showed overall greater spatial coherence than its temperate counterpart. First of all, as Germany in MAM-3m, the Iberian Peninsula was a region of generally increased weather system activity in JJA-ev. There was a dipole pattern of consistently positive $f'^{rel}_{90d}(C)$ in the West, and a mostly to consistently positive $f'^{rel}_{90d}(A)$ in the East (Fig. 7a,b). This synoptic pattern fosters more frequent southerly advection. Note that the Iberian Peninsula was also a hot spot of low $NDVI$ events, i.e., results there were consistent among up to six events (Fig. 2c). The typical signal of positive $f'^{rel}_{90d}(A)$ and negative $f'^{rel}_{90d}(C)$ occurred in the central Mediterranean in JJA-ev, and in most of Mediterranean forest grid cells in MAM-3m (Fig. 7c,d) and in DJF-6m (Figs. E1 & E2). In these two seasons, only 5% and 6% of considered forest grid cells showed mostly or consistently positive $f'^{rel}_{90d}(C)$. Also in SON-9m, cyclones were mostly less frequent than usual (Figs. E2). Thus, Mediterranean low $NDVI$ grid cells very often experienced negative $f'^{rel}_{90d}(C)$ in the past year of the meteorological history, consistent with the results in Fig. 5f. Low $NDVI$ events in the western Iberian Peninsula hot spot region, however, experienced the opposite change in cyclone frequency in JJA-ev, which might be a signal of intensified Iberian thermal lows (Santos et al., 2015).

## 4    Discussion

### 4.1    Low $NDVI$ events

The low $NDVI$ grid cells identified in this study typically represented summers with a heat- and/or drought-induced loss of forest greenness (Anyamba and Tucker, 2012; Orth et al., 2016; Buras et al., 2020). Known drought and heat events were identified as low $NDVI$ events, e.g., the Iberian drought in 2005 (Gouveia et al., 2009), a two-year-long drought in $2007-2008$ in Turkey (Varol and Ertuğrul, 2016), the $2011-2013$ drought in the Balkans (Cindrić et al., 2016), the hot summer 2017 in Italy (Rita et al., 2020), and the Central European hot drought in 2018 (Schuldt et al., 2020; Senf and Seidl, 2021b). Additionally, we identify 2022 as record-breaking year of the most widespread low $NDVI$ events covering 37% of the Mediterranean and temperate forest biome each. In 2022, Europe experienced its hottest JJA on record alongside dry soils (Copernicus Climate Change Service, 2022), and the largest carbon emissions from wildfires since 2007 were recorded (Copernicus Atmosphere Monitoring Service, 2022). Specifically, among the countries with most low $NDVI$ grid cells were also those that faced extreme anomalies in wildfire activity; the burnt area in Romania, Germany, France, Spain, and Croatia was 11, 10, 7, 4, and 3 times larger in 2022, respectively, than the $2006-2021$ average (EFFIS, 2022). First observations of early leaf senescence as in 2018 are mentioned by Kittl (2022). The regions spared by the low $NDVI$ event in 2022 - mainly Scandinavia, parts of France, and a belt from the Austrian Alps to the Baltic - were the only regions that showed a surplus in surface soil moisture compared to $1991-2020$ conditions (Copernicus Climate Change Service, 2022). Our approach to identify low $NDVI$ events, therefore, not only identifies events due to heat and drought but also events due to positively interacting disturbances such as fire and insect outbreaks. Finally, however, there was at least one example of a drought-unrelated low $NDVI$ event, namely an ice storm that hit Slovenia in February 2014 (Senf and Seidl, 2021c; Buras et al., 2021). Consequently, we cannot rule out

that other disturbances that were not necessarily linked to heat or drought, e.g., also late frost (Bascietto et al., 2018; Vitasse et al., 2019), have impacted some of the low $NDVI$ grid cells.

As our approach identifies persistent and wide-spread $NDVI$ losses, more localized and potentially more extreme reductions in forest greenness are often not captured (Appendix C), e.g., logging in France in 2009 (Senf and Seidl, 2021a), or low $NDVI$ following the winter windstorm Gudrun in southern Sweden in 2005 (Buras et al., 2021). Interestingly, also the hot drought in 2003 hardly lead to low $NDVI$ grid cells in Europe. The $NDVI$ reduction in that summer was most prominent for grassland and crops but less so for forests (Buras et al., 2020). Forests are capable of resisting a temporally limited drought much better than grassland, as they can respond with reduced evapotranspiration and increased water use efficiency (Wolf et al., 2013). Note, however, that grassland is typically recovering better after long-lasting droughts than forests (Stuart-Haëntjens et al., 2018). The only forest regions that were affected by a low $NDVI$ event in 2003 are in southern France and Italy, where strongest growth reductions in forests were observed (Ciais et al., 2005). Our results, therefore, suggest that - compared to the events since 2018 - the impact of the 2003 drought on forest greenness was generally limited and scattered in space.

## 4.2 Meteorological histories and their inter-biome differences

The purpose of systematically analyzing meteorological histories of low $NDVI$ events was to identify statistically significant meteorological precursors to these events. Hereby, it should be noted that this statistical analysis alone does not allow to infer causation between the precursors and the low $NDVI$ events, but identifies unusual co-occurrence of these precursors and the low $NDVI$ events. The causation surmised in our interpretation of these precursors below is inferred from the large body of process-focused literature we cite. We neither identify a universally valid meteorological history leading to low $NDVI$ events nor establish hitherto unknown causal links between seasonal time scale meteorology and low $NDVI$ events. Rather, the value of our approach is that we can systematically examine which aspects of the meteorological history stands out of the noise and variability that are invariably present across the large set of meteorological histories (e.g., Fig. 4) identified here.

The case study of low $NDVI$ grid cells in Spain in 2005 (SPA05) is one example that illustrates this value of our approach (Sect. 3.2). The meteorological history of SPA05 showed a precipitation surplus in the previous spring and summer, which was not a significant meteorological precursor to low $NDVI$ events in the Mediterranean in general (Sect. 3.3.1). A preceding surplus in precipitation in a water-limited region such as Spain could cause structural overshoot, i.e., the build-up of large crowns with high water demand, which was suggested to worsen the following drought impact (Zhang et al., 2021). Long-term irrigation experiments at dry sites revealed reduced tree growth over several years as a response to ceasing irrigation (Rigling et al., 2003; Feichtinger et al., 2014). Furthermore, short-term irrigation causes more pronounced responses in tree growth than long-term irrigation, while both potentially increase the sensitivity to drought in the following years indirectly via increased leaf area and tree height (Feichtinger et al., 2015). So while there is strong evidence for lagged responses to a previous precipitation surplus due to structural overshoot, our approach shows that this process does not translate to a systematic meteorological

precursor at the biome scale in the Mediterranean.

The most striking meteorological precursors of low $NDVI$ grid cells were the persistence of a precipitation deficit (both biomes) and of positive temperature anomalies (temperate biome) over at least two years. Continuously dry conditions reached farther back in Mediterranean than in temperate forests, which might be an important difference due to year-round growth of widespread evergreen tree species in the Mediterranean (Camarero et al., 2021). Also, these conditions play an important role for forest fires, which likely aggravated the meteorological impact on $NDVI$ indirectly (Nagel et al., 2017; Turco et al., 2017). The identified extremely unusual accumulation of warm periods over around 25 months prior to events in the temperate biome points to its indirect effects on insect populations and fire, as well as to the joint amplification of drought impacts (Seidl et al., 2017; Sommerfeld et al., 2018; Seidl et al., 2020; Forzieri et al., 2021). Also, because the significantly pronounced warm periods occurred in the (late) growing season of all three preceding years, continuously increased temperatures might have worsened the impact of the event-concurrent hot drought through structural overshoot and soil moisture depletion (Bastos et al., 2020a; Zhang et al., 2021). This four-month-long hot drought in JJA-ev was of significant magnitude for both meteorological anomalies in the studied low $NDVI$ grid cells. In the Mediterranean, a large precipitation deficit preceded positive $T2m'_{90d}$ by another four months, which could follow from the fact that winter/spring drought in southern Europe increases the likelihood of a hot JJA through an enhanced soil moisture-atmosphere feedback (Seneviratne et al., 2010; Russo et al., 2019). More generally and in both biomes, the emergence of an unusually strong 90-day precipitation deficit already in spring can be particularly damaging (Senf et al., 2020; Bigler and Vitasse, 2021; Bose et al., 2021).

Apart from the accumulation of dry periods reaching far back in time in both biomes, primarily temperate forests show meteorological precursors that occurred more than one year in the past. Significantly reduced $P_{90d}$ and increased $T2m_{90d}$ occurred during the previous JJA, which points towards drought legacy effects (Anderegg et al., 2015). This legacy might not always be reflected in $NDVI$ (Kannenberg et al., 2019), however, it can indirectly affect future forest vitality via reduced tree resilience (Bose et al., 2020). Moreover, the succession of drought in consecutive summers is particularly harmful for temperate forests, while Mediterranean forests show a decreased sensitivity to the second drought (Anderegg et al., 2020). To summarize, the systematic meteorological histories of low $NDVI$ events and differences between the two biomes can be linked to much of the current mechanistic understanding of forest vitality in the two bio-climatic regions.

## 4.3 The role of weather systems

Our results highlight that the timing and positioning of weather systems is crucially determining their impact on surface meteorology relevant for low $NDVI$ grid cells. At the biome scale, the at least 34-month-long dry period in the Mediterranean is accompanied by reduced 90-day cyclone frequency, most so in DJF and MAM when cyclones are climatologically most frequent (Wernli and Schwierz, 2006). Cyclones are the main contributor to cold season precipitation in these forest regions (Rüdisühli et al., 2020), and also to extreme precipitation (Pfahl and Wernli, 2012b). Other water-limited forest regions show a similar sensitivity to cold season precipitation, and therefore, to the precipitation-causing weather phenomenon (Williams

et al., 2013). More frequent anticyclones were typical in JJA-ev and MAM-3m and relate to an upper-level subtropical ridge extending into the Mediterranean - a known driver of heat extremes in southern Europe (Sousa et al., 2018; Zschenderlein et al., 2019). Over the western Iberian Peninsula in JJA-ev, specifically, more frequent cyclones likely occurred as Iberian thermal lows that favor summer heat extremes through increased diabatic heating over the continent (Santos et al., 2015). Thus, reduced cyclone activity all along the meteorological history of Mediterranean low $NDVI$ events appears to have been the main contributor to the hot-dry meteorological precursors - with the exception of Iberian thermal lows in JJA.

In temperate forests, the JJA-ev and JJA-12m hot-dry conditions were both accompanied by more frequent anticyclones, especially in regions at the southern edge of the storm track. The accompanying reduction in cyclone frequency in northern Europe corresponds to a northward shift of the jet stream, which can lead to reduced forest greenness in these regions (Messori et al., 2022). More frequent anticyclones, on the other hand, often relate to an upper-level blocking that causes heat and precipitation suppression in central to northern Europe (Pfahl and Wernli, 2012a; Zschenderlein et al., 2019). A few regions in western Europe show an opposite signal, i.e., reduced anticyclone frequency in JJA-ev. This relates to the fact that summer precipitation there frequently occurs within high-pressure systems (Rüdisühli et al., 2020). In these cases, convective precipitation occurs in the moist and unstable inflow west of the anticyclone center (Mohr et al., 2020). So while in JJA a European-centered anticyclone can favor low $NDVI$ grid cells in northern Europe, it might be unfavorable for low $NDVI$ grid cells in western Europe. All in all, these considerations highlight the importance of weather systems and the necessity of considering their spatiotemporally varying impact on surface meteorology, also when interested in events of substantial forest impact.

### 4.4 Caveats

The two main caveats of this study are (i) the event aggregation to the comparably large scale, and (ii) the relatively short data record. The former implies that our analyses can neither account for species-specific drought responses (Scherrer et al., 2011; Vanoni et al., 2016), nor for the multi-dimensional nature of tree mortality (Allen et al., 2015; Etzold et al., 2016; Schuldt et al., 2020). The link between drought, drought response, and tree mortality is mediated by site, stand, and tree properties (Etzold et al., 2019; Vitasse et al., 2019; Frei et al., 2022), and can further be shaped by tree species diversity within a forest (Grossiord et al., 2014), its micro-climate (Buras et al., 2018), and legacies of changing environmental conditions due to, e.g., past forest management (Thom et al., 2018). This aggregation, however, is a central element of this study as we aimed to investigate the link of synoptic atmospheric variability with variability in forest $NDVI$, which both act on very different spatial scales. The event identification is, therefore, targeted to identify only spatially coherent losses of forest $NDVI$, which are meaningful to aggregate to the larger scale. Also, the sub-sampling of the identified low $NDVI$ grid cells ascertains that our results do not highlight meteorological precursors that are unique to very few events or regions. Nevertheless, the results of this study should be confronted with more specific and local impact assessments.

The main consequence of the relatively short data record is that the normalization of meteorological anomalies suffers from significant sampling uncertainty, which renders any comparison over space and time rather difficult. The normalized $P'_{90d}$

and $T2m'_{90d}$ then not necessarily represent the actual site-level temperature and precipitation values and their interpretation requires care (Zang et al., 2020). This specifically applies when comparing the meteorological history of the temperate with the

515 Mediterranean biome, respectively, as the latter climatologically receives little precipitation during summer (Schultz, 2005). The normalization, however, is a way to use basic meteorological variables that can readily be interpreted and linked to weather system dynamics, which is of great importance to the novelty of this study.

## 5 Conclusions

This study identified specific aspects of the meteorological history (the three-year evolution of 90-day temperature, $T2m'_{90d}$,

and precipitation anomalies, $P'_{90d}$), which are systematically shared characteristics of events of persistently low summer forest greenness at the 50 km scale in Europe in 2002−2022. Forest greenness as measured by the $NDVI$ is also used as an early warning mechanism of forest dieback (Buras et al., 2021). First and foremost, in the temperate and Mediterranean biome the regions with low $NDVI$ events in 2022 exceeded the previous record summers 2018 (temperate) and 2008 (Mediterranean) by far with regard to spatial extent. In the hottest summer in Europe on record, 37% of both forest biomes were affected by

persistently low $NDVI$, which is about +13% more than during the previous records. In contrast, our approach classifies the impact of the hot-dry summer 2003 on forests as very limited and, if so, scattered in space.

The approach used in this study identifies and quantifies those meteorological features that preceded many of the events in the same way and reveals considerable inter-biome differences. The persistence of dry periods was significantly increased

over at least 26 and 34 months prior to low $NDVI$ events in the temperate and Mediterranean biome, respectively. In contrast, the persistence of hot periods was only significantly increased (at least for 25 months preceding the events) in the temperate biome, but not the Mediterranean biome. Closer to the event summer, negative $P'_{90d}$ and positive $T2m'_{90d}$ were significantly anomalous in magnitude. In the temperate biome, both anomalies acquired statistically significant magnitudes in spring, four months before the low $NDVI$ event. In the Mediterranean biome, negative $P'_{90d}$ arose another four months earlier, i.e., eight

535 months prior to low summer $NDVI$. Note that a single $P'_{90d}$ value that was anomalous, e.g., eight months prior to the low summer $NDVI$, denotes an anomaly that refers to a 90-day period, i.e., to the eight to ten months prior to the event. Lastly, the systematic meteorological histories are able to verify whether meteorologically related processes from local observations apply to an entire biome. We discuss structural overshoot (Zhang et al., 2021), which is plausible to systematically affect low $NDVI$ events in the temperate biome through warmer or extended growing seasons. In contrast, structural overshoot due to

540 more precipitation in the previous year is highly plausible for a case study in the water-limited Mediterranean, which, however, does not translate to the biome scale.

Finally, we provide clear evidence on the spatially varying impact of synoptic-scale weather systems on the important meteorological precursors. At the biome scale, the prominent dry periods are often caused by a significantly reduced cyclone

frequency in the Mediterranean biome, and by increased anticyclone frequency in the temperate biome, respectively. This ef-

fect can, however, differ at a local scale, depending on which weather system is locally relevant for precipitation. For instance, western Europe often receives summer precipitation from convective cells in anticyclones and, thus, hot-dry conditions in the event summer go along with reduced anticyclone frequency.

The important differences between the meteorological histories impacting temperate and Mediterranean forests as identified in this study provide a better understanding of European forests' response to multi-seasonal meteorology. Moreover, we, for the first time, quantify and assess the impact of the extremely hot summer 2022 and compare it with that of the preceding twenty years. Finally, the presented systematic investigations bridge the gap between forest dynamics and atmospheric dynamics, and, thereby, constitute progress in how expected forest dieback can be linked to changing meteorological and climatic conditions
under global warming.

## Appendix A: Sensitivity to threshold parameters

The event identification is based on three threshold parameters, namely the minimum affected ratio $AR^{min} = 80\%$, the minimum forest area $FA^{min} = 10\%$, and the minimum number of time steps in JJA with negative $NDVI'$ $c_{ev}^{min} = 4$ (Sect. 2.3). Parameter $AR^{min}$ refers to the fraction of forest pixels that has to show persistently negative $NDVI$ per $0.5° \times 0.5°$ grid cell for that grid cell to be identified as low $NDVI$ grid cell. Persistently here is defined via a lower threshold $c_{ev}^{min}$ for $c_{n,ev}$, where the latter refers to the number of time steps out of a total of six in JJA that show negative $NDVI$. Lastly, $FA^{min}$ sets a minimum forest cover per grid cell to filter out those with only very few forest $NDVI$ pixels. We vary $AR^{min}$ and $FA^{min}$ by $\pm5\%$ and test different combinations thereof. We vary $c_{ev}^{min}$ by $\pm1$ only for the setup "80_10" used in the study ($AR^{min} = 80\%$, $FA^{min} = 10\%$) as the identification scheme depends strongly on this parameter. Table A1 shows the number of events ($n_{tot}$), the number of years with at least ten low $NDVI$ grid cells in $2002-2022$ ($n_{yr}$), and also the number of events per sub-sample ($n_{ev}$) that result from varying the threshold parameters. Large $n_{yr}$ is important to the sub-sampling of the low $NDVI$ grid cells, which is used to retrieve more systematic results, and would optimally be as close to the total years of 21 as possible. The $n_{tot}$ is particularly strongly reduced when increasing $AR^{min}$ to 85%, resulting in $n_{yr} \leq 10$. Aiming to optimize $n_{yr}$ by using looser thresholds, however, would misconceive a typical characteristic of extreme events such as low $NDVI$ events, namely that they occur concentrated in individual years and not in others. So looser thresholds have the disadvantage of reducing the peculiarity of low $NDVI$ grid cells. This is illustrated by the $\sim 1.5\times$ increase in $n_{tot}$ when reducing $AR^{min}$ from 80% to 75%. While reducing $c_{ev}^{min}$ in the study setup (80_10) from four to three has only minor effects, $n_{tot}$ and consequently $n_{yr}$ and $n_{ev}$ are drastically reduced when increasing $c_{ev}^{min}$ to five (Table A1). For example, in the temperate biome, only three years would contribute at least ten low $NDVI$ grid cells if $c_{ev}^{min} = 5$ was used. So while the number of events is not very sensitive to reductions in $c_{ev}^{min}$, increasing $c_{ev}^{min}$ would make a systematic assessment impossible.

**Table A1.** Number of events ($n_{tot}$), number of years with at least ten low $NDVI$ grid cells ($n_{yr}$), and number of events per sub-sample ($n_{ev}$) in the temperate and Mediterranean biome for different combinations of threshold parameters. The column title indicates the setup with $c_{ev}^{min} = 4$ and different $AR^{min}$ and $FA^{min}$ combinations separated by an underscore - except for the last two columns. These denote tuning the threshold parameter $c_{ev}^{min}$ to three and five while $AR^{min} = 80\%$ and $FA^{min} = 10\%$.

| | **Sensitivity** | | | | | | | | | | |
|---|---|---|---|---|---|---|---|---|---|---|---|
| | **75_5** | **75_10** | **75_15** | **80_5** | **80_10** | **80_15** | **85_5** | **85_10** | **85_15** | **3of6 (80_10)** | **5of6 (80_10)** |
| $n_{tot}$ **in temp.** | 2294 | 1998 | 1744 | 1580 | **1386** | 1204 | 929 | 809 | 707 | 1770 | 144 |
| $n_{yr}$ **in temp.** | 17 | 17 | 17 | 16 | **15** | 15 | 10 | 9 | 9 | 16 | 3 |
| $n_{ev}$ **in temp.** | 187 | 185 | 183 | 173 | **170** | 164 | 138 | 132 | 124 | 179 | 35 |
| $n_{tot}$ **in Med.** | 1701 | 1319 | 1078 | 1287 | **989** | 808 | 861 | 661 | 529 | 1181 | 195 |
| $n_{yr}$ **in Med.** | 18 | 16 | 15 | 16 | **14** | 14 | 12 | 12 | 11 | 14 | 6 |
| $n_{ev}$ **in Med.** | 195 | 187 | 177 | 177 | **164** | 160 | 155 | 142 | 136 | 177 | 93 |

The sensitivity of the main result to these two parameters is illustrated at the example of Fig. 5. As in the original Fig. 5, we perform a random sub-sampling of up to ten low $NDVI$ grid cells per year for each biome and compute an average meteorological history from the resulting samples (Sect. 3.3). This sub-sampling is done ten times for each biome, and Fig. A1 shows the median of these ten equivalent average meteorological histories for every of the eleven combinations of $AR^{min}$, $FA^{min}$, and $c_{ev}^{min}$ listed in Table A1. The sub-sampling for each combination of the two threshold parameters is of course dependent on the identified low $NDVI$ grid cells and, therefore, dependent on $n_{tot}$ and $n_{yr}$ (Table A1).

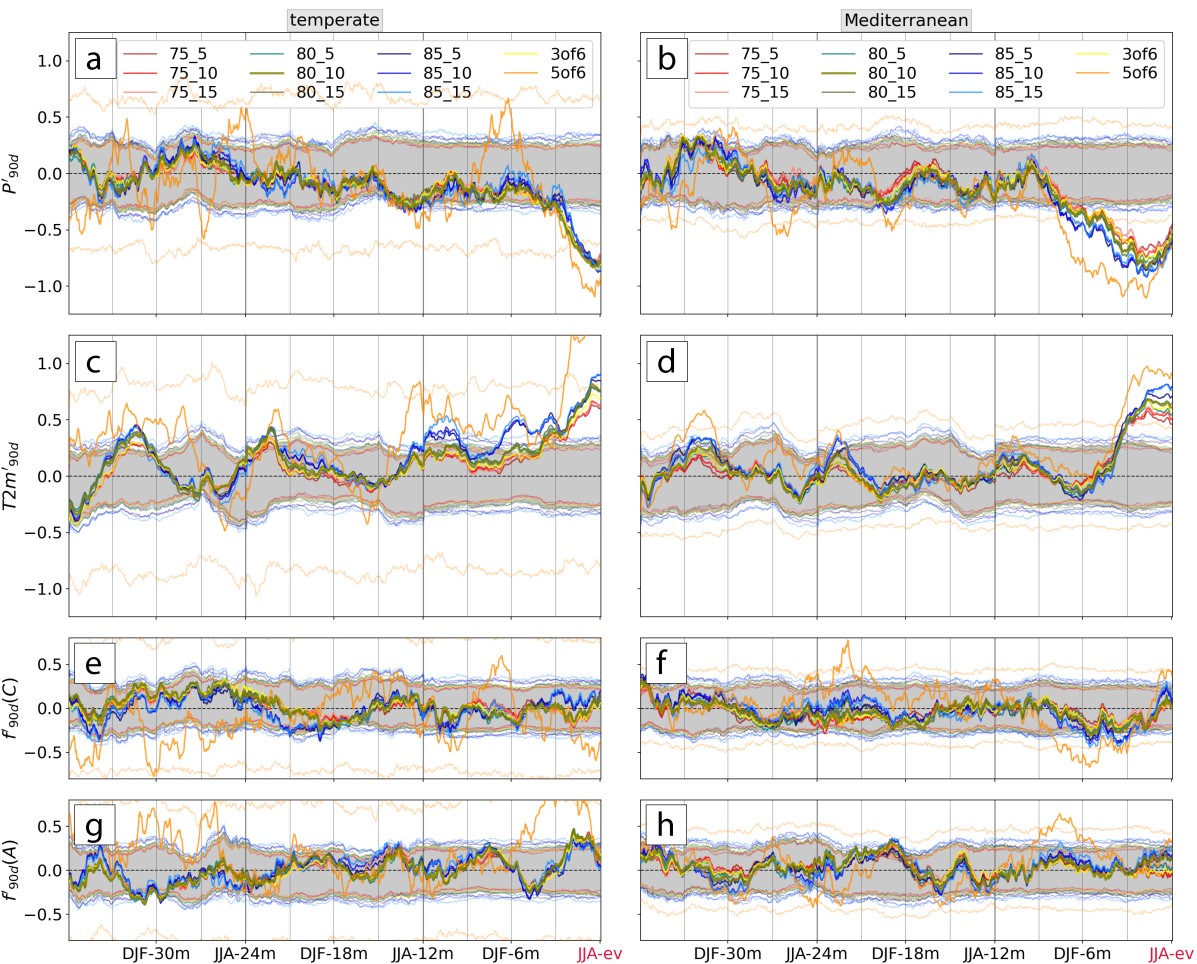

**Figure A1.** Same as Fig. 5 but for different combinations of $AR^{min}$, $FA^{min}$, and $c_{ev}^{min}$. The thicker olive line and the grey shaded 95% confidence interval (CI) correspond to the median setup shown in the study (Fig. 5). The other thick lines show the meteorological histories of the other combinations of threshold parameters, and the corresponding CI is shown with thin lines of the same color. The normalized 90-day mean (a,b) precipitation, (c,d) temperature, (e,f) cyclone frequency, and (g,h) anticyclone frequency anomalies $f'_{90d}(C)$ and $f'_{90d}(A)$, respectively, are shown as line plots. The legend indicates the combination of threshold parameters used as in Table A1.

Figure A1 overall highlights low sensitivity of various aspects of the meteorological history on the two threshold parameters. First, the number of events per sub-sample and thus per average meteorological history $n_{ev}$ differs for every setup of threshold parameters depending on variations in $n_{tot}$. Consequently, setups with more events per sample (loose thresholds) lead to smaller magnitudes of the averaged meteorological anomalies, and, hence, also a more narrow confidence interval than a setup with fewer events per sample (stricter thresholds). The comparison here, therefore, focuses mostly on aspects such as the timing and evolution of significant anomalies instead of their exact magnitude. The statistically significant anomalies highlighted in our study, e.g., negative $P'_{90d}$ in JJA-12m and JJA-ev, and positive $T2m'_{90d}$ that emerged in MAM-3m in temperate forests, respectively, would also result from other parameter setups (Fig. A1a,c). Especially the timing when meteorological anomalies were significantly different from climatology are consistent within almost all eleven setups. Some of the highlighted anomalies persisted longer and emerged more clearly when using stricter thresholds, e.g., $AR^{min} = 85\%$ and $FA^{min} = 15\%$. Positive $T2m'_{90d}$ followed JJA-12m into SON-9m and also the warm period prior to JJA-ev reached farther into the past (Fig. A1c). With that setup, also the negative $f'_{90d}(C)$ in MAM-3m prior to low $NDVI$ events in the Mediterranean biome were more distinct than for the setup used. With stricter parameter setups, however, $n_{ev}$ is unfavorably reduced as the number of years contributing the maximum of ten low $NDVI$ events ($n_{yr}$) is greatly reduced (Table A1). This is strongly pronounced when using the setup "5of6", for which, e.g., in the temperate biome, only three years (2018, 2019, and 2022) contribute substantially to the average meteorological history shown in Fig. A1. Considering the numbers in Table A1 this setup can clearly not provide a meaningful evaluation of low $NDVI$ events over the study period. Apart from that setup, any larger deviation between the results from the different parameter setups typically occur within the respective confidence intervals - e.g., $f'_{90d}(A)$ in JJA-ev - and are, hence, not highlighted in the analysis and interpretation of Fig. 5. To summarize, the sensitivity analysis supports the chosen setup with $AR^{min} = 80\%$, $FA^{min} = 10\%$, and $c_{ev}^{min} = 4$, and generally demonstrates low sensitivity of the main results to reasonable variations in the three parameters.

## Appendix B: Bootstrapping tests

In the bootstrapping test we want to test the null hypothesis $H_{0,EV}$ that a given aspect $X$ of the meteorological history at $t_{ev} - \Delta t$ of low $NDVI$ events is equal to that of any random meteorological history. For $X$ we use the meteorological fields of $T2m'_{90d}$, $P'_{90d}$, $f'_{90d}(C)$, $f'_{90d}(A)$, as well as the fraction of $\Delta t$ where $T2m'_{90d} > 0$ and $P'_{90d} < 0$, respectively, covering $1999-2022$ and the study domain (Fig. 1a). The fields are used here for the period 1999—2022, in order to compute three-year meteorological histories for all low $NDVI$ events in 2002-–2022. Figure B1 illustrates the procedure of retrieving event mean meteorological histories as well as the way the bootstrapping is constructed.

First, the sub-sampling of all low $NDVI$ grid cells results in an event set $EVS$ of low $NDVI$ grid cells that can be represented as three-dimensional binary $21 \times n_{lat} \times n_{lon}$ event mask, which is equal to 1 at every low $NDVI$ grid cell (Fig. B1a). Further, for every $X$ we retrieve an $n_{ev} \times n_{dt}$ matrix - i.e., one with a time series with $n_{dt} = 3 \times 365$ daily time steps for every of the $n_{ev}$ selected low $NDVI$ grid cells - by extracting the fields of $X$ where $EVS$ equals 1. The average time series of $X$ for

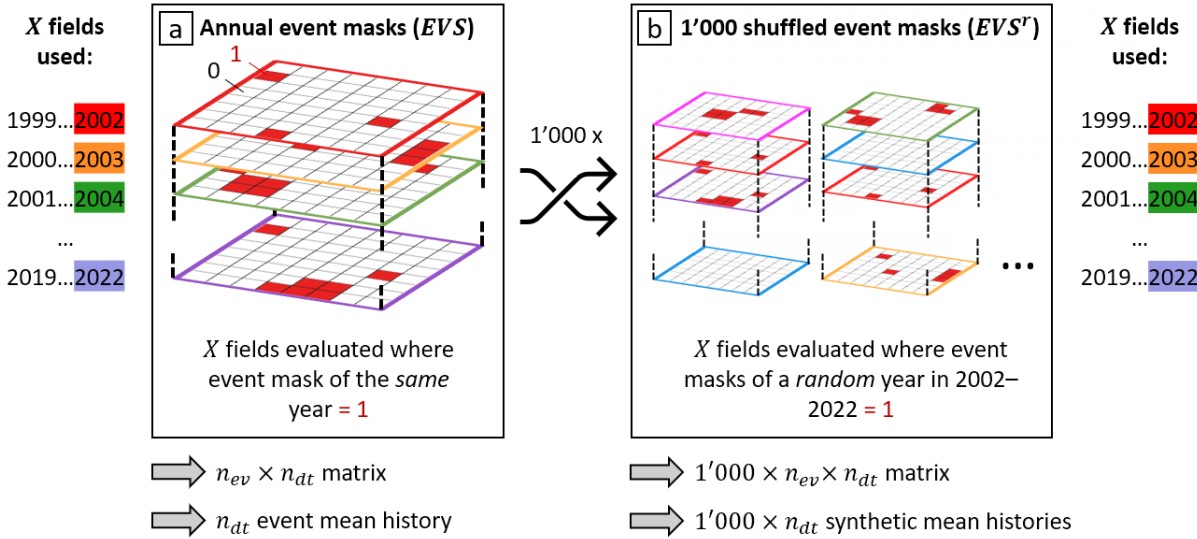

**Figure B1.** Schematic of the construction of the event-mean meteorological histories (a) of event set $EVS$, and of (b) their synthetic analogues $EVS^r$ used for the bootstrapping test. The rows on the side show the order in which fields of $X$ are used to extract the time series for $EVS$ and each $EVS^r$. The annual binary event masks of $EVS$ in (a) are colored according to the year of occurrence. The shuffled $EVS^r$ have a different order of these masks, however, use the $X$ fields in the same order as $EVS$.

one sample results from taking the mean along the first dimension of this matrix, as shown in Fig. 5. For the bootstrapping test, we generate 1'000 synthetic event sets ($EVS^r$) by randomly shuffling the 21 annual masks of $EVS$ 1'000 times (Fig. B1b). This shuffling process is best visualized by shuffling a deck of cards, whereby each card corresponds to a the binary $n_{lat} \times n_{lon}$ event mask of a specific year. Specifically, when constructing the random event set with number $r$ we assign a randomly se-

620 lected year $y_i^r$ to all low $NDVI$ grid cells occurring in year $y_i$ and then repeat the process for all remaining years. Hereby, the random years are chosen such that each year occurs once in every $EVS^r$. Consequently, each reference event set contains the same number of low $NDVI$ grid cells as $EVS$ but in a different year-location combination. Afterwards, the synthetic meteorological histories are generated by first retrieving $X^r$ from extracting $X$ fields for $EVS^r$, i.e., using the shuffled deck of annual binary event masks. Then, the resulting $1'000 \times n_{ev} \times n_{dt}$ matrix is averaged along its second dimension to retrieve a

625 set of 1'000 synthetic event-mean time series for every $X^r$. We, thereby, create 1'000 meteorological histories that are equally plausible in the climatological reference period without the prerequisite of a following low $NDVI$.

We then compare event-mean time series of $X$ of low $NDVI$ grid cells to the 1'000 synthetic event-mean time series of $X^r$. Values of $X$ outside the range of $X^r$ receive a p-value of 0 (Röthlisberger et al., 2016). The remaining p-values are estimated

from the percentiles of the $1'000 \times n_{dt}$ synthetic matrix along its first dimension. At the significance level of $\alpha = 5\%$, $H_{0,EV}$ is rejected at time lags $\Delta t$ if the event value of $X$ is outside the 95% confidence interval, i.e., outside the $2.5^{th} - 97.5^{th}$ percentile range, of the 1'000 reference values of $X^r$. Note that the shuffling of years is done prior to extracting the spatial fields of $X$

from the ERA5 data set. This has – in contrast to a random sampling of all forest grid cells – the convenient effect that spatial correlation in these meteorological variables is retained. Thus, synthetic meteorological histories of $X^r$ are constructed from a data set with exactly the same spatial correlation as the original data sets of $X$.

## Appendix C: Maps of low $NDVI$ grid cells

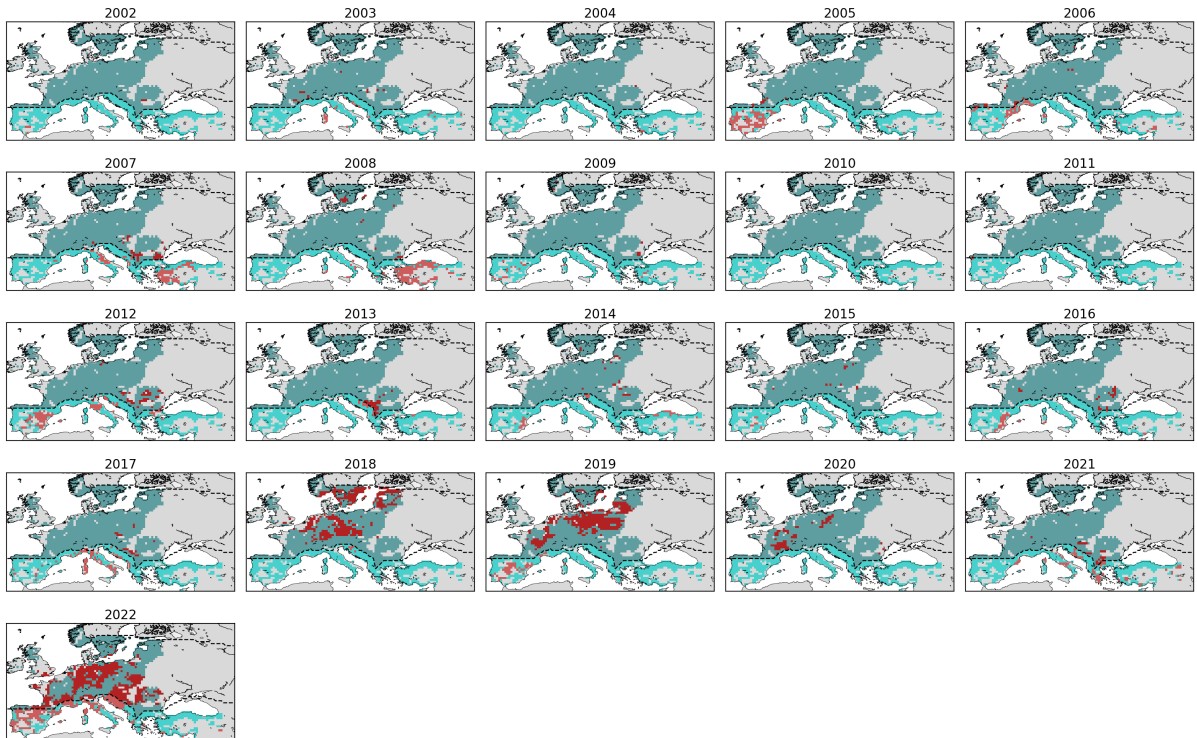

**Figure C1.** Low $NDVI$ grid cells in $2002-2022$ (red) in forest grid cells of the temperate (turquoise) and Mediterranean biome (cyan). The dashed lines delineate the two biomes.

## Appendix D: Low NDVI events and forest disturbance

We provide a brief and qualitative comparison of our set of low $NDVI$ grid cells with the independent disturbance data set of Senf and Seidl (2021a). The comparison is useful to put the identified low $NDVI$ events into perspective regarding existing knowledge on forest disturbances.

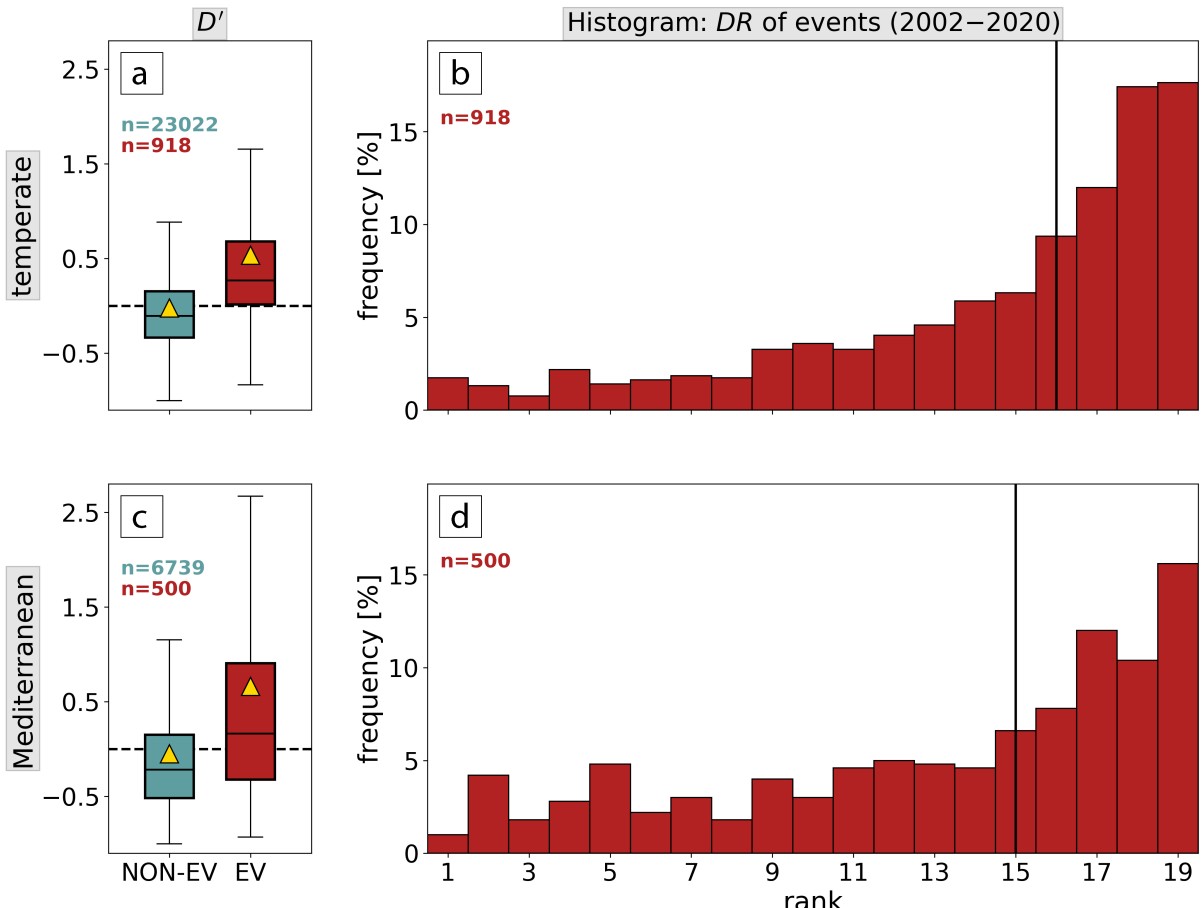

**Figure D1.** Event comparison for (a,b) the temperate and (c,d) the Mediterranean biome in the period 2002−2020. (a,c) box plots of the disturbance anomaly $D'$ of low $NDVI$ grid cells (red) and non-event grid cells (turquoise). The distribution mean is shown by a yellow triangle, outliers are omitted. (b,d) histograms of ranks 1−19 of disturbance area $DR$ of low $NDVI$ grid cells in 2002−2020. The median is shown by the vertical line.

## D1 Forest disturbance data set

We use the forest disturbance data set by Senf and Seidl (2021a) with an original resolution of 30 m. It is based on a time-series segmentation approach called LandTrendr (Kennedy et al., 2010) and identifies tree canopy mortality in 1986−2020. The approach uses two spectral bands (shortwave infrared I and II) and two spectral indices (tasselled cap wetness and normalized burn ratio) from Tier 1 Landsat 4, 5, 7, and 8 images in Jun−Sep. For more details see Senf and Seidl (2021a). From this data set we use the annual disturbance area $D_{J,n}$, which is aggregated for every $0.5° \times 0.5°$ forest grid cell. We only use years and grid cells that overlap with our study period and forest grid cells as identified in Sect. 2.1. Our event data set overlaps with the disturbance data set in the time period of 2002−2020 at 91% of forest grid cells as $D$ does not cover Turkey. Consequently,

66% and 51% of all low $NDVI$ grid cells in the temperate and Mediterranean biome are compared to the disturbance data
set. More specifically, we use two measures of $D$: the disturbance anomaly $D'$, and the rank of $D$ among the 19 annual values
$DR_{J,n}$ in 2002$-$2020:

$$D'_{J,n} = \frac{D_{J,n} - \overline{D_J}}{\overline{D_J}} \tag{D1}$$

$$DR_{J,n} = rank(D_{J,n}) \tag{D2}$$

at forest grid cell $J$ in year $n$, with $\overline{D_J}$ denoting the climatological mean disturbance area in 2002$-$2020. When referring to
low $NDVI$ grid cells in the following, we thereby only address those that spatially overlap with $D$ data in 2002$-$2020.

## D2  Qualitative comparison

In 70% of all low $NDVI$ grid cells the disturbance area $D$ is larger than on average in 2002$-$2020 - more often in the temperate
(76%) than in the Mediterranean biome (59%; Fig. D1a,c). The median disturbed area increases by +27% and +16% during
low $NDVI$ events in the temperate and Mediterranean biome, respectively. Furthermore, non-events typically go along with
negative $D'$ in the temperate (61% of non-events) and the Mediterranean biome (66%). Figure D1b,d additionally shows the
disturbance area rank, $DR$, from 1 (smallest $D$ in 2002$-$2020) to 19 (largest $D$). With 1$-$6 events per affected forest grid cell
(Fig. 2c) a low $NDVI$ grid cell would go along with $DR$ 14$-$19 if the event years were equal to the years of largest disturbed
area. The majority of low $NDVI$ grid cells indeed cover ranks 16$-$19 and 15$-$19 in the temperate and Mediterranean biome,
respectively. We conclude that low $NDVI$ grid cells tend to go along with more forest disturbances, i.e., enhanced canopy
mortality, and rank among the largest forest disturbances at forest grid cells.

## Appendix E:  Weather system anomalies

The following Figures E1 & E2 show the spatial pattern of weather system anomalies. For all forest grid cells with at least two
low $NDVI$ events in 2002$-$2022, we show how many of these events were linked to positive or negative anomalies in $f'^{rel}_{90d}$.
Additionally, we calculate the average anomaly over all events that had the same sign of the anomaly and highlight those with
mean changes of at least 25%. For each season of the past year, from JJA-ev backward to SON-9m, we use the value of $f'^{rel}_{90d}$
at the last day of the season, which is approximately equal to the seasonal average over the three preceding months.

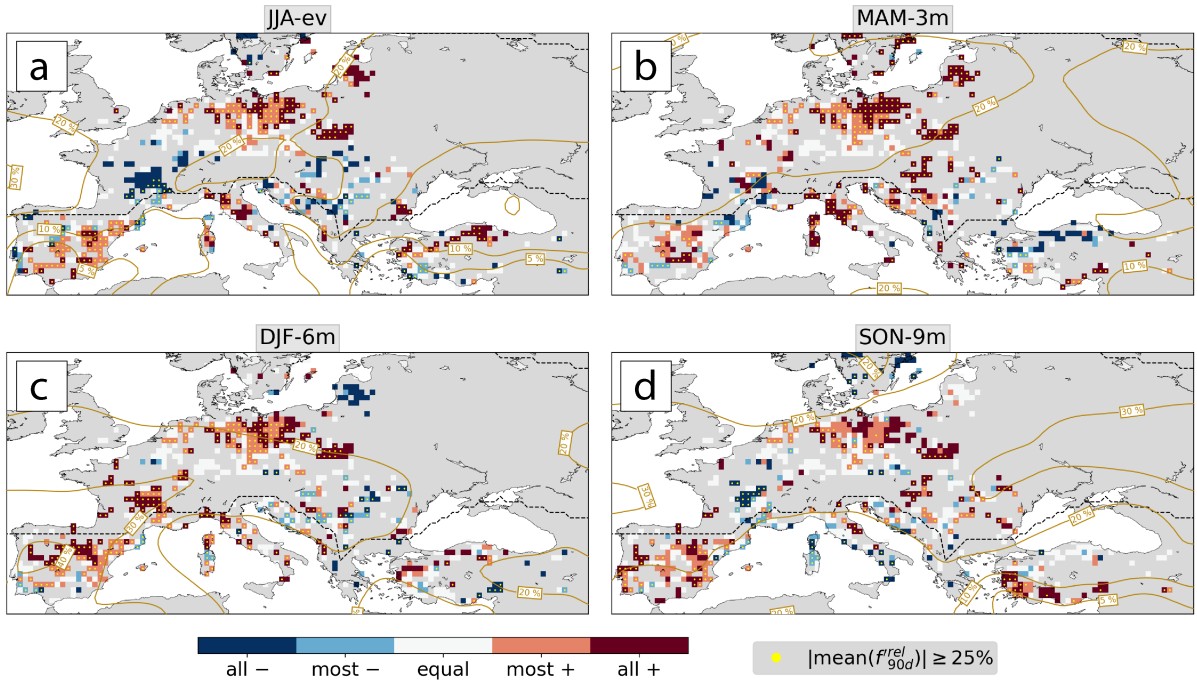

**Figure E1.** The same as Fig. 7 but for the relative anticyclone frequency anomaly $f_{90d}^{\prime rel}(A)$ in (a) JJA-ev, (b) MAM-3m, (c) DJF-6m, and (d) SON-9m.

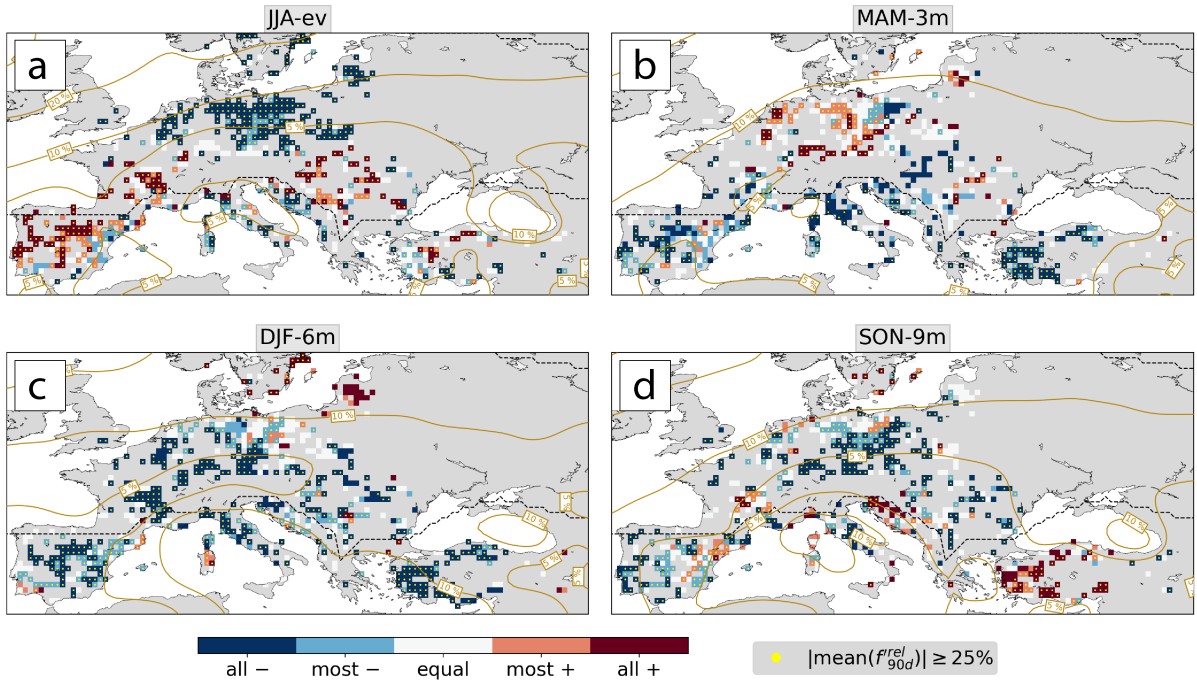

**Figure E2.** The same as Fig. 7 but for the relative cyclone frequency anomaly $f'^{rel}_{90d}(C)$ in (a) JJA-ev, (b) MAM-3m, (c) DJF-6m, (d) SON-9m.

*Code and data availability.* We uploaded the low $NDVI$ events in JJA $2002-2022$ as identified in this study to the ETH Research Collection (https://doi.org/20.500.11850/505559). The data sets used in this study are freely available, namely 16-daily $NDVI$ data from the NASA Application for Extracting and Exploring Analysis Ready Samples (AppEEARS; https://appeears.earthdatacloud.nasa.gov), global forest cover area by Büttner et al. (2004, https://land.copernicus.eu/pan-european/corine-land-cover), and atmospheric fields of ERA5 from the ECMWF (https://cds.climate.copernicus.eu/cdsapp#!/dataset/reanalysis-era5-pressure-levels?tab=form). In Appendix D, we use the updated version 1.1 of forest disturbance data (Senf and Seidl, 2021a) and aggregate the data to the ERA5 grid. Version 1.0 is available at https://doi.org/10.5281/zenodo.3925446. All other data and code is available upon request.

*Author contributions.* MH performed most of the analyses and wrote a first version of the manuscript in close exchange with MR and HW. CS contributed processed data to the study. All authors contributed to the design of the study, the interpretation of the results, and the writing.

*Competing interests.* The authors declare that they have no conflict of interest.

*Acknowledgements.* We acknowledge the NASA Application for Extracting and Exploring Analysis Ready Samples (AppEEARS), the European Centre for Medium-Range Weather Forecasts (ECMWF), and the Copernicus EU for providing all the essential data sets for this study. MH is grateful to Benjamin Stocker and Ana Bastos for fruitful discussions. Finally, we thank two anonymous reviewers for constructive comments on previous versions of the manuscript as well as the associate editor Trevor Keenan.

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

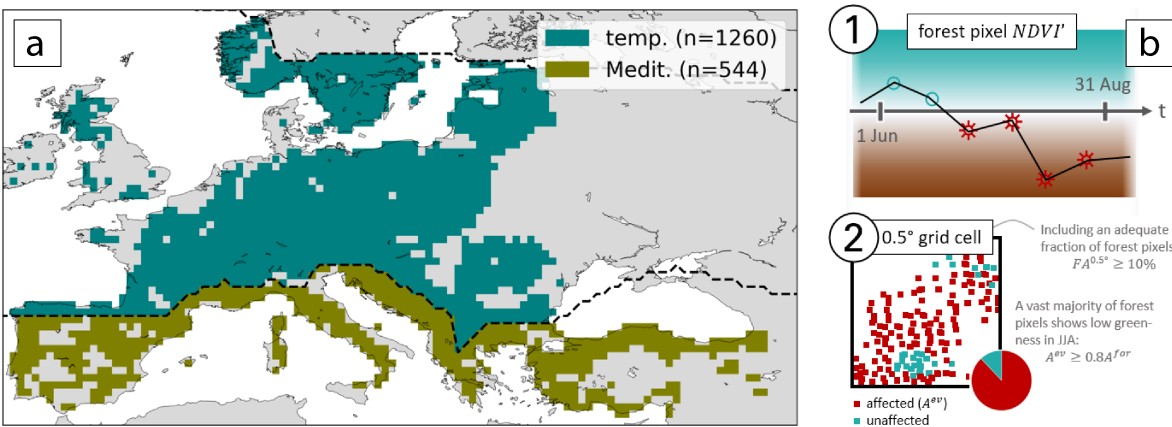

**Figure 1.** (a) forest grid cells ($FA^{0.5} \geq 10\%$) in the study domain, separated into temperate and Mediterranean forests by the black dashed line. The boreal biome is cropped by the second dashed line in the Northeast of the domain. (b) an example of the identification of low $NDVI$ grid cells, where (1) forest pixels are flagged if at least 4/6 time steps show negative $NDVI'$, and (2) $0.5° \times 0.5°$ forest grid cells are flagged if more than 80% of the forest pixels within are flagged (details provided in the text).

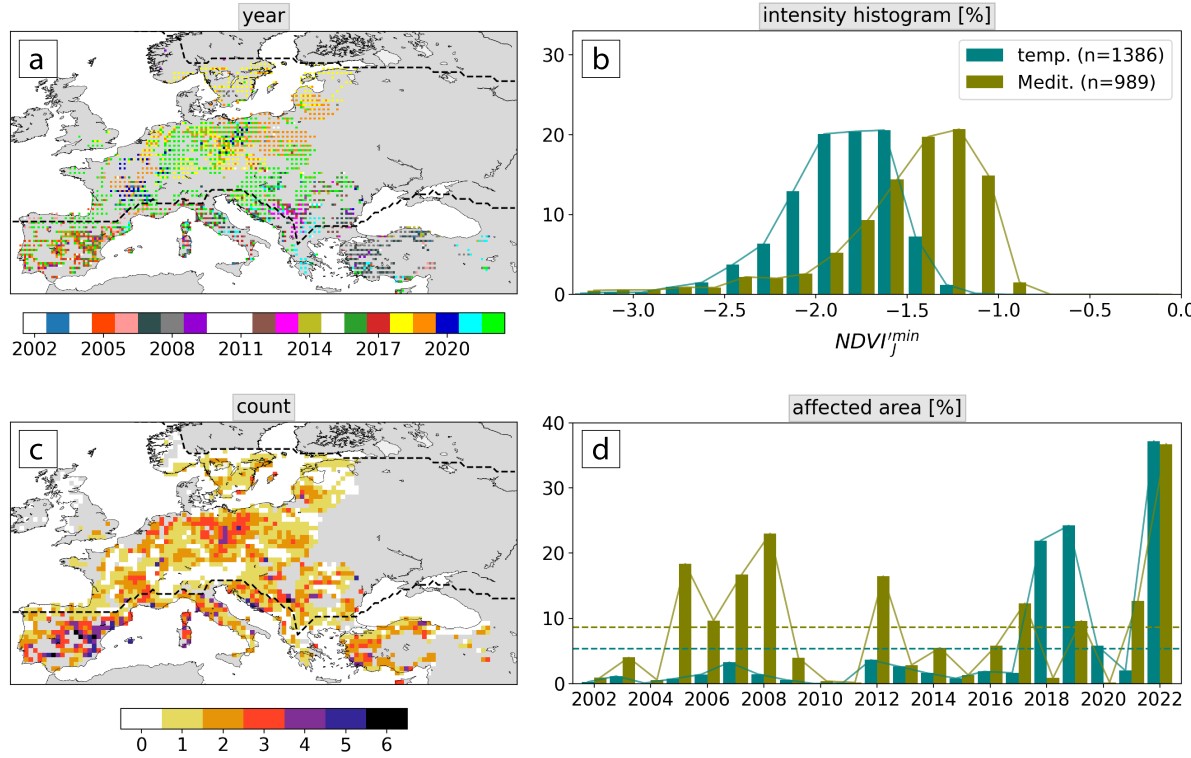

**Figure 2.** Maps of (a) years with a low $NDVI$ event (Appendix C), and (c) total number of events. (b) Histogram of event intensity as measured by $NDVI_J'^{min}$ in the two biomes, and (d) time series of the biome-integrated area of low $NDVI$ grid cells relative to all forest grid cells (in %). Years with few low $NDVI$ grid cells are shown in white in (a). In (a) each grid cell is split into four quads, showing the event year of up to the four most intense events. Dashed lines in (d) show the average over all years. Dashed lines in (a,c) delineate the temperate and Mediterranean biome.

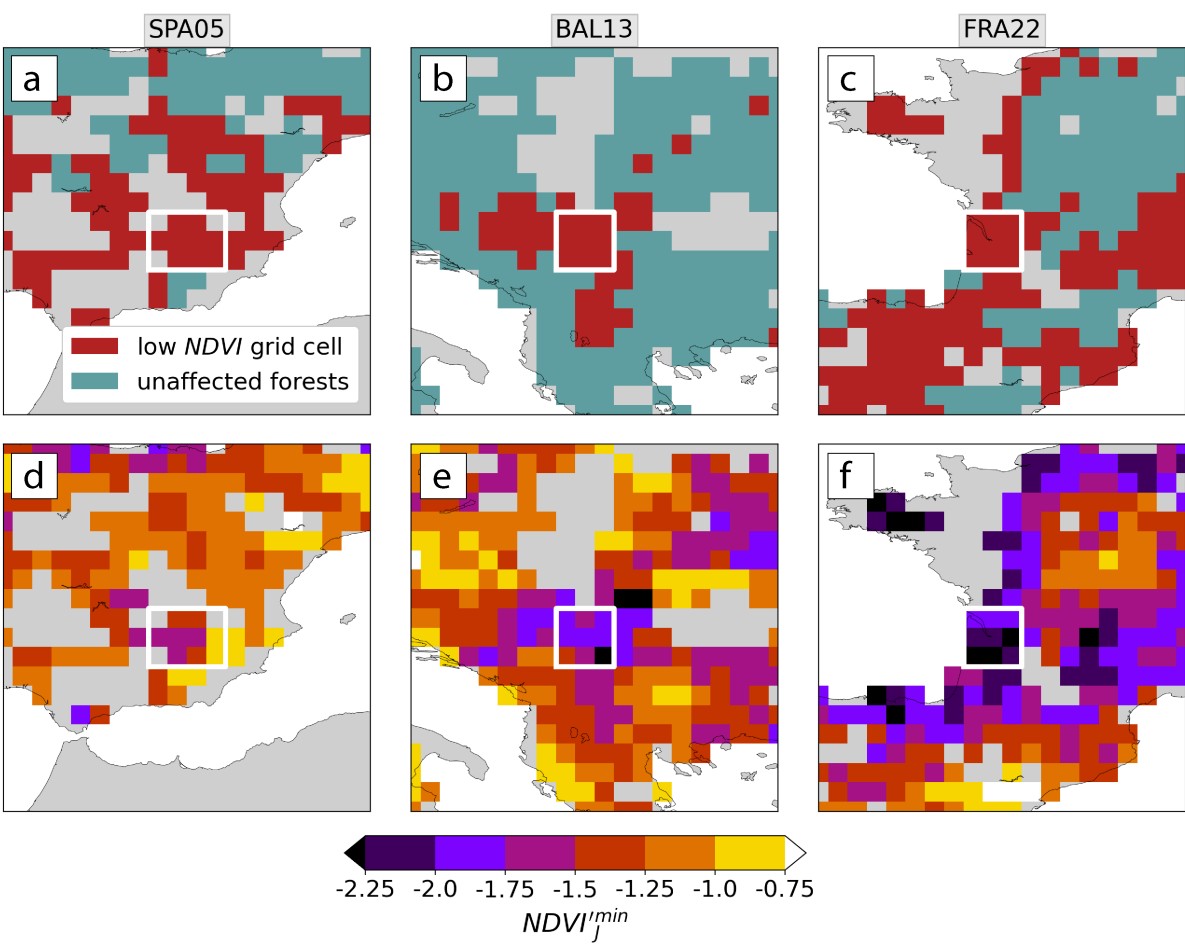

**Figure 3.** (a-c) Low $NDVI$ grid cells and (d-f) the event intensity measured by $NDVI'^{min}_J$ in (a,d) Spain in 2005, (b,e) the Balkans in 2013, (c,f) France in 2022. The focus regions of SPA05, BAL13, and FRA22 are framed with white boxes.

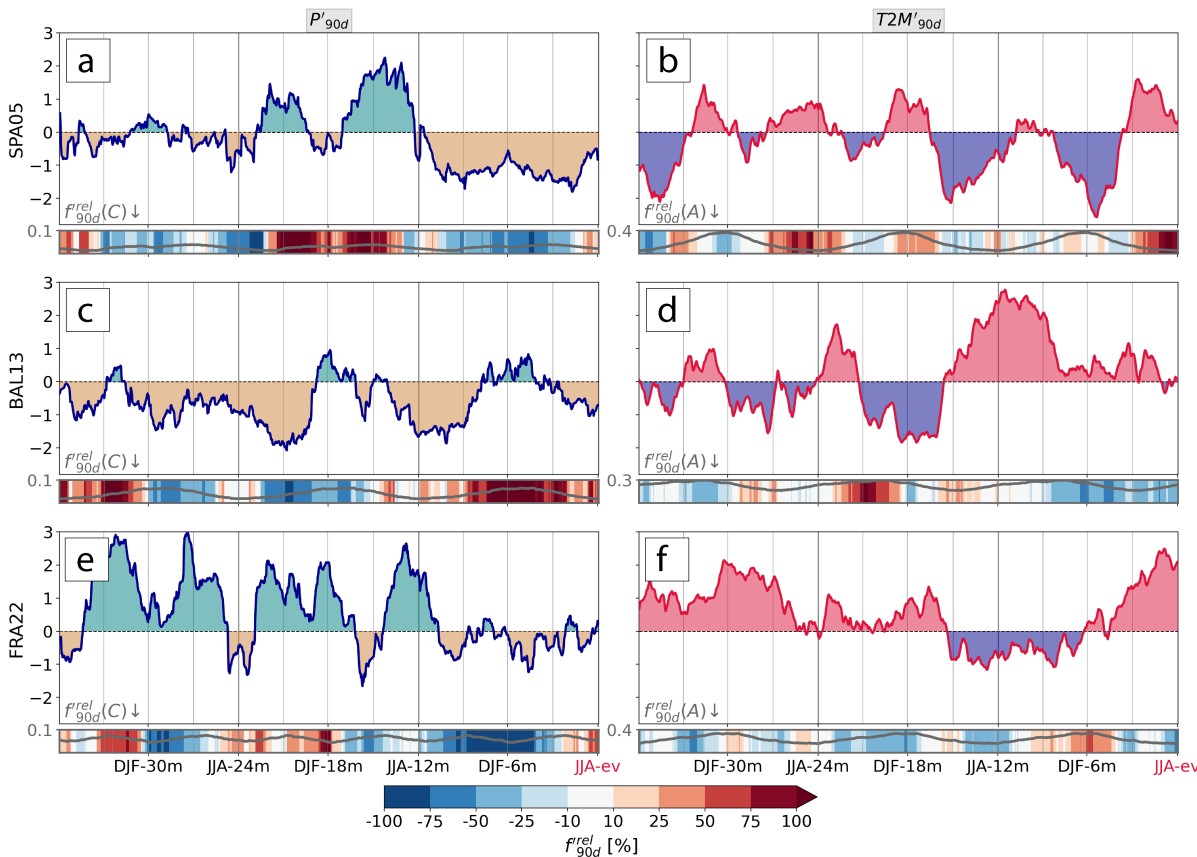

**Figure 4.** Three-year evolution of (a,c,e) $P'_{90d}$ and (b,d,f) $T2m'_{90d}$ leading up to low $NDVI$ events in (a,b) Spain in 2005, (c,d) the Balkans in 2013, (e,f) France in 2022. The relative anomaly of (a,c,e) cyclone frequency, and of (b,d,f) anticyclone frequency is shaded and their climatological mean is shown as grey line in the heat map panels.

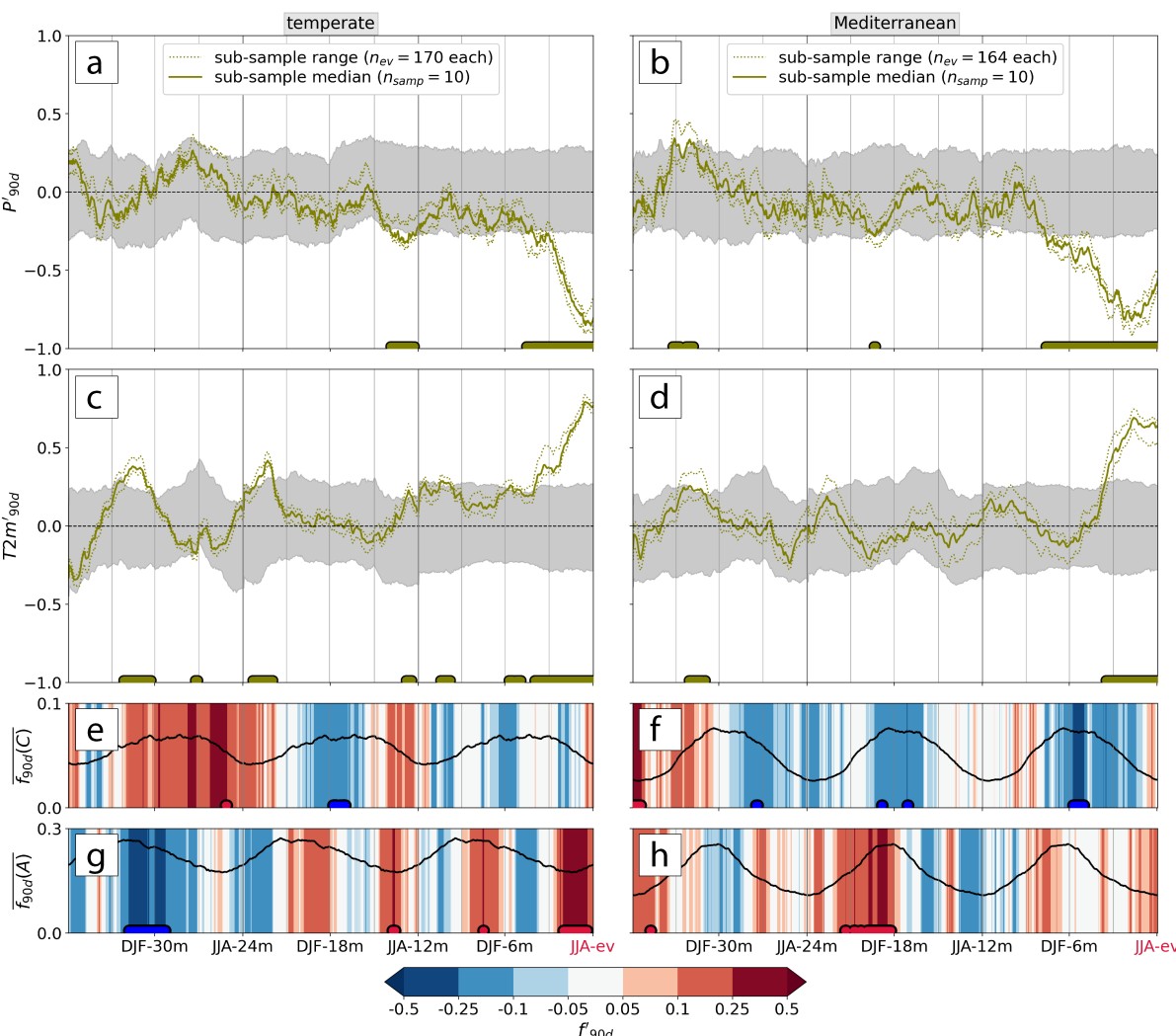

**Figure 5.** Average three-year evolution of (a,c) $P'_{90d}$ and (b,d) $T2m'_{90d}$ at low $NDVI$ grid cells (olive lines). The range spanned up by the $n_{samp} = 10$ sub-samples (of $n_{ev}$ low $NDVI$ grid cells each) is dotted, their median in solid. The confidence interval (CI), i.e., the $2.5^{th} - 97.5^{th}$ percentile of the reference climatology is shaded grey (see Sect. 2.4.2). The normalized (e,f) cyclone and (g,h) anticyclone frequency anomalies $f'_{90d}(C)$ and $f'_{90d}(A)$, respectively, as median over the 10 samples are shaded in colors. The 90-day climatology of the weather system frequencies is displayed as solid line. Plots apply to events in the (a,c,e,g) temperate, and (b,d,f,h) Mediterranean biome. Statistically significant median values outside the 95% CI are marked by colored dots at the bottom of each panel.

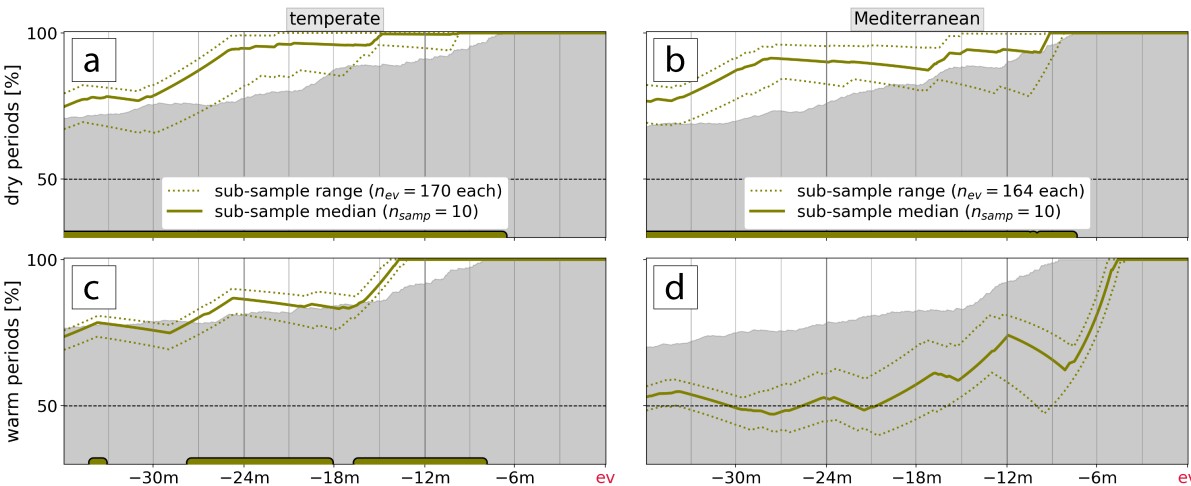

**Figure 6.** The average fraction of the integration period with a (a,c) dry ($P'_{90d} < 0$) and (b,d) warm period ($T2m'_{90d} > 0$) for decreasing integration period $\Delta t$ prior to low $NDVI$ events. The range spanned up by the $n_{samp} = 10$ sub-samples (of $n_{ev}$ low $NDVI$ grid cells per biome) is dotted, their median in solid. The grey shading displays the 95% confidence interval (CI) of the reference climatology. Statistically significant median values outside the 95% CI are marked by colored dots at the bottom of each panel.

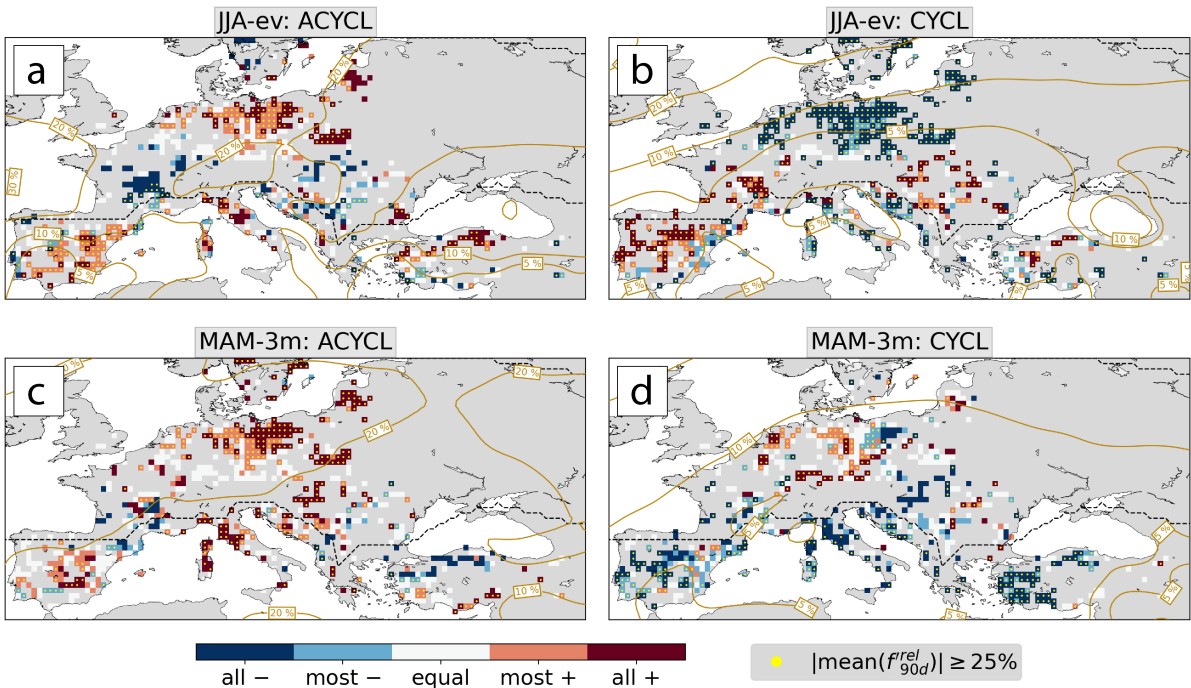

**Figure 7.** The consistency in sign of (a,c) $f_{90d}^{\prime rel}(A)$, and (b,d) $f_{90d}^{\prime rel}(C)$ for all forest grid cells with at least two low $NDVI$ events in $2002-2022$. Maps are shown for the last day of (a,b) JJA-ev, and (c,d) MAM-3m. Stippling indicates that the absolute average over all anomalies of the same sign is at least 25%. 2D-Gaussian-smoothed ($2\sigma$) climatological weather system frequencies are shown in beige contours.