# Peer review of "Meteorological history of low forest greenness events in Europe in 2002–2022"

_EGUsphere, 2022_

## Referee Comment (RC2)

Herrmann et al. analyse storylines of low summer NDVI events in Europe for the period 2000-2020. The storylines are estimated for up to two years prior to the low NDVI events and are analysed separately for the temperate and Mediterranean biomes. This is an extremely relevant topic, since the ongoing disturbances in Europe are in many regions unprecedented (Senf & Seidl, 2021) and the drivers of tree mortality are still elusive (Hartmann et al., 2022). The use of storylines to evaluate drivers of low NDVI events is an innovative and well-fitting approach that could have a high potential to shed light into the general processes driving these events. After reading the manuscript in detail, I believe the study unfortunately falls short of this ambition for generality, due to a number of fundamental issues with the specific methodological approach, underlying data, and the general lack of mechanistic insights, as discussed in more detail below. Moreover, the reading is at times engaging but at times it can be quite cumbersome due to the (heavy) use of confusing notation, unclear grammar and some distracting sentences that read as if the authors were addressing specific reviewers' comments, perhaps from a previous submission. That said, I believe that the analysis performed here can be of great value to the community, should these issues be addressed.

Major comments
1) Generality of the storylines & mechanistic interpretation
At the moment, the results and discussion are presented as if the authors can derive general lessons about meteorological storylines leading to reduced forest greenness in summer. Given the short length of the time record (3 years), and that very few (1-3) large-scale events dominate the overall number of events detected – 42% of individual events correspond to the 2018 drought/heat event – it is hard to accept that the conclusions can be generalized to different event types. This is even more problematic for the "consecutive events", which refer basically to the 2018/19 event (making up 82% of the "individual events"). Therefore, the event samples cannot be treated as individual events. If I understood well (see my comment below) the authors calculate the anomalies for the climate variables by selecting 10k randomly selected samples and calculate their means over the different events for each of the variables. By using a random sampling of $n$ "pixel" events that are dominated by a single large-scale event (or two for the repeated events), a large fraction of the 10k samples will stem from this single event, so that their sampling will necessarily be biased. It follows that the meteorological anomalies estimated for these storylines will strongly reflect the anomalies of those 1-2 synoptic-scale events. This results in the authors possibly overinterpreting short-term departures of the storylines from the 95% confidence interval, particularly the rainy previous winter season – that occurred in much of the region affected by the 2018 event (https://climate.copernicus.eu/european-wet-and-dry-conditions) – as a general percursor of low summer NDVI events. It is unclear what would be the mechanistic link between rainy winters and dry/hot summers leading to low NDVI events and the authors do not offer satisfactory explanations (rather list a number of hypothetical processes in the discussion).

I would be curious to whether the authors could find significant signals, especially the rainy winter peak before the low NDVI events in temperate regions, if events occurring in 2018 are removed from the analysis. The authors could also try an alternative sampling approach that would explicitly correct for the sampling bias due to the predominance of few large-scale synoptic events, but this would reduce drastically the number of events (by about half for EV10 and by 80% for EV11). An alternative could be to tone down the ambition to derive general conclusions for 2000-2020 and rather focus on highlighting the value of the storyline approach to learn about anomalies in

vegetation activity and, specifically, extremes. While the results for 2018/19 in central Europe are not necessarily new, the fact that the approach proposed here shows consistent results past studies while providing a means to assess more generally past history is a strength that could/should be stressed. For the Mediterranean case, the results might also reflect the year 2017 the most (given figure 3d), so that a similar approach/reasoning can be applied.

2) Spatial resolution
The authors analyse storylines leading to low NDVI extremes, focusing on forest ecosystems. To do this, they rely on NDVI data from MODIS at 0.05km. This is already a relatively coarse resolution for the analysis proposed, as pointed out by Reviewer 1, but the authors then further coarsen the data in the analysis to 0.5 degree spatial resolution. The authors tried to minimize this issue by estimating low NDVI values first at the 5km resolution and then imposing the condition that at least 50% of the coarser grid cell needs to register low NDVI events. However, their definition of "forest pixels" includes pixels with less than 50% of forest, and even as low as only 20% (Section 2.1). This results in many areas that are dominated by croplands or mixed tree/herbaceous cover (parts of central and eastern Europe and of Italy, southern Sweden), being included as "forests".
While I understand that the authors could not perform this analysis at the higher MODIS spatial resolution given the lack of such high-resolution climate data over Europe, it is unclear why the authors chose to aggregate results to 0.5 degrees, since ERA5's native resolution is 0.25 degree, and ERA5 land provides downscaled fields at 0.1 degree spatial resolution, much closer to the spatial scale used to define events and to distinguish (at least better) forest from non-forest pixels. The same for E-OBS, and the group of COSMO-REA reanalyzes provide temperature and precipitation fields at 6km or even 2km.
Surely, repeating the analysis at such fine resolutions (2km) would be more computationally expensive, so that keeping the 5km scale of the initial steps is probably the most feasible option.

3) Synthetic event sets and meteorological means
In appendix A and the methods, it is generally unclear if the 10k event sets are derived per pixel individually, as suggested by keeping $n$ in subscript (appendix A), or across pixels or across individual events (i.e. pairs of (pixel, year)). It is also unclear how the authors calculate the means and climatologies of the meteorological variables (Line 562-63). Generally, this makes it difficult to evaluate and reproduce the analysis.
The authors state that they take 10k synthetic event sets and then (1) take the mean among all events, then (2) take a distribution of these mean values that is used to calculate climatologies. I have several questions about this step:
-  It is not clear what is the size of the event set matrix, is it 3d (10k, 21, n)?
- And over which dimension is the first mean calculated, the second?
- What is the size of the resulting vector/matrix?
- Do the climatologies in T and P have seasonality?
- Over which dimension is the standardization performed?
For the consecutive event sets, how exactly is this done, do you select from EV10 subsets of synthetic events that happen consecutively? Where are the values of the different n values derived from?
Finally, the authors mention in the methods section that the sampling strategy preserves the spatial correlation of T and P fields, but this is not discussed in Appendix A.

As a consequence of these issues, it is difficult to evaluate the robustness of the method described. I suggest either describing the steps using explicitly mathematical notation, reporting the sizes of the vectors/matrices, and/or a flowchart to facilitate understanding of the exact steps performed here. It would also be good to discuss the rationale behind using bootstrapping in the methods and start of Appendix A.

4) Use of forest disturbance dataset
It is unclear why the authors introduce a whole new dataset that is only used for cross-comparison purposes. Low NDVI events do not necessarily have to correspond to crown-mortality events (D), and conversely crown mortality events might not necessarily result in low NDVI if understory vegetation benefits from the canopy opening. This discrepancy is briefly mentioned by the authors, and very clear in Figure 4, but not fully addressed. Another point not really mentioned is that D can be affected strongly by anthropogenic signals, such as management and selecting logging. Since the authors presented the goals of the analysis as to evaluate extremes in vegetation greenness, a clearly well-defined and constrained problem, I find that the comparison with D adds more confusion, rather than clarity to the study.

Specific comments:

"forest performance" is not a standard expression and is rather unclear whether the authors mean vitality, health, or any other aspect reflected by NDVI (vegetation cover, LAI, …). To be accurate, the best would be to stick to "forest greenness".

Line 23: Over what time-scales is this sentence referring to? This is not true for the past several decades – acid rain, changes in management, reforestation, elevated CO2, nutrient deposition, … there is a long list of processes that have been destabilizing forests in Europe.

Line 48-50: but in Mediterranean regions, drought can also reduce fuel load (Pausas and Ribeiro, 2013).

Line 56: intensively discussed… in the literature?

Line 59: "stressed", or "identified"?

Line 60: "drought prone region", such as the Mediterranean?

Line 63: "margins" of the growing season is an unusual expression

Line 65: increasing understanding "of …", specify what is meant here. Furthermore, the concept of storylines should be described in more detail and appropriately referenced (e.g. Shepherd et al. 2018)

Line 73-74: Other studies have attributed this to a Wave-7 pattern and a positive NAO phase (Drouard et al., 2019 and Kornhuber et al., 2019)

Line 84: what do the 90 in subscript stand for, 90-day moving average not yet mentioned, making it confusing.

Line 85: why 3 year only?

Lines 87-88: I propose swapping (2) and (3)

Lines 101-102: why the choice of this specific domain and why ignoring boreal forests?

Figure 1 caption: please add a brief description of panel b) and it is impossible for the reader to understand it without reading the methods.

Line 106: no justification about why 0.5 degree is used.

Line 116: "at forest pixels" does not seem grammatically correct

Line 119: this is the first time missing values are referred to. Where do they stem from? The use of quality control flags? And what if there are two consecutive months missing?

Line 123: in mathematical notation, the apostrophe is usually used to express the first derivative, so this notation is confusing. Why not a for anomaly?

Line 132: replace "at" by "for" or "in"

Line 134: there is no scheme presented here. Do the authors mean the "approach presented here"?

Line 135: is it a "forest grid cell" or an "atmospheric grid cell over forested pixels"? Overall, I find these definitions confusing.

Line 136 and Equation 2: how does the flag work when there at 3 months in the season? Since the authors take the minimum value over the season, does this mean that it is enough that 1 month in JJA is flagged as low NDVI? How can the authors be confident that this is a "low NDVI season"?

Line 148: correct to "reanalysis", singular

Line 149: why interpolating ERA5 to 0.5 degree?

Line 151: if this refers to seasonal averages, should the subscript be 90d or "season"?

Line 151-159: it is not fully clear if the standardization is done also for the 90d moving windows, please clarify.

Line 167: so only 40% of the values are "not extreme"?

Line 167-169: give correlation values and respective significance

Figure 2: what do the colorbars indicate? Not mentioned in the caption.

Line 181: this is not described in the Appendix A.

Line 191: for a Biogeosciences audience, is would be good to explain what the "outermost closed SLP contour …" means.

Line 191-195: more generally, for a Biogeosciences audience it would be good to explain what additional information does this analysis bring.

Line 197: grammar "To evaluate", "for" + "ing" does not express purpose/intention.

Line 201: which is aggregated and normalized. Why is the normalization now done at the coarser resolution?

Line 216: correct "succeeding" to "subsequent"

Section 3.1.1 – if the purpose of using the D dataset is for evaluation of low NDVI events, why not compare the annual variability in D as well here and in Figure 3?

Figure 3b: the colors in the 4 quads are hardly distinguishable

Figure 3d: if the authors decide to keep D, then add the extent affected by D events in this panel as well.

Line 249: these "conceptual, technical and physical reasons" are not really thoroughly discussed in Sec 41.

Figure 4: What is D'[]?

Figure 5: mark in shaded areas the periods when the event mean is outside of the 95% CI. What do the vertical lines in a-d indicate?

Line 269: negative, but still within the 95% CI. Here, and elsewhere, the authors over-emphasize non-significant results.

Line 271: for short periods. Add "is significant for Xdays … "

Line 274: continuously negative, but still within the 95% CI. Please give duration of the periods when event mean is outside of the 95% CI.

Line 276: negative in the previous winter, but not extreme.

Line 280: DJF of which year?

Line 284: can you give a mechanistic explanation for this?

Line 304: add ", respectively," between "P'90d" and "from"

Line 315: the accumulation of dry periods is not significant

Section 3.2.3: please state clearly that this applies basically to 2018/19 in temperate regions, and make a similar assessment for the Mediterranean biome.

Line 331: the information about the fraction of these years to the event samples needs to be given much earlier in the manuscript, and please add information for the Mediterranean biome too.

Line 334: why is 2020 excluded?

Line 340: "hot" anomalies?

Line 350: again, I find in-depth mechanistic interpretation of these patterns lacking in the discussion.

Line 355: "exerts" is not applicable here, since the NDVI events have no influence on T90.

Line 356: I quite like that the authors here give specific values of the anomalies discussed. This should be done throughout the whole results section.

Line 358: what does "small" mean? That the absolute value is close to zero?

Line 364: only T, see comment above for line 315.

Figure 8: please explain what the different color shades mean (95%CI, I believe)

Discussion Section: it is surprising that the limitations related to the short temporal records and the dominance of single years in the events analysed are not discussed.

Lines 414-325: How does the analysis done here "help to characterize the nature of these events"? If this would be true, I would expect separate analyses for pixels with low NDVI and no crown mortality and pixels with low NDVI and crown mortality. Overall, this paragraph is quite distracting given that the main goal of the paper was to analyse low NDVI extremes.

Line 432-433: the grammar can be improved

Line 437: regadless of what?

Line 437-438: do the authors mean that drought early in the growing season directly "damages" forests, or simply that low P in spring promotes drier summers?

Line 440: unclear why warming in the previous 3 years would affect an instantaneous process like fire.

Line 485: can you explain in more detail how acclimation results in reduced leaf area and productivity?

Line 488-490: isn't this simply a consequence of the fact that only low NDVI events were selected? The authors did not evaluate post-event recovery trajectories separately, so that they cannot know whether increased vulnerability out competes acclimation.

Line 491: sensitivity to drought is not shown in the results, what do the authors mean here?

Line 501: what does "superior statistical modelling" mean?

Line 508: "not shown", please add these results to the supplement and this is an important point.

Conclusions
I find that the authors overemphasize the winter wet signal in the temperate biome, which first is rather short (a couple of days outside of the CI) and second cannot likely be generalized for all events. This should be toned down.

Appendix A
Line 557: the superscript $r$ should be placed above $n$, and be defined again in the text here, for those readers who might start here.

Line 571: Add "for" before the "null hypothesis". Also, EV10 and EV11 have not been defined previously in the appendix, making reading confusing.

References
Shepherd, T.G., Boyd, E., Calel, R.A. *et al.* Storylines: an alternative approach to representing uncertainty in physical aspects of climate change. *Climatic Change* **151,** 555–571 (2018). https://doi.org/10.1007/s10584-018-2317-9

Pausas, J.G. and Ribeiro, E. (2013), Fire and productivity. Global Ecology and Biogeography, 22: 728-736. https://doi.org/10.1111/geb.12043

Drouard, M., Kornhuber, K., & Woollings, T. (2019). Disentangling dynamic contributions to summer 2018 anomalous weather over Europe. Geophysical Research Letters, 46, 12537–12546. https://doi.org/10.1029/2019GL084601

Kornhuber, K., Osprey, S., Coumou, D., Petri, S., Petoukhov, V., Rahmstorf, S., & Gray, L. (2019). Extreme weather events in early summer 2018 connected by a recurrent hemispheric wave-7 pattern. *Environmental Research Letters*, *14*(5), 054002.

---

## Author Comment (AC1)

We acknowledge the reviewers for their careful reading and detailed commenting of our manuscript. With the final author comments below we reply to the major reviews of both reviewers including suggestions on how to revise the discussed aspects of our study. Additionally, due to the profound critiques raised by the reviewers and the substantial amount of planned revisions, we first provide an overview of the major revisions we have in mind. Major revisions planned (in the following referred to as MRP) include:

1. **Updated MODIS NDVI data set:** We address the probably most major critique by downloading NDVI at ~250m spatial resolution and 16-day temporal resolution from 2002–2022 (Didan, 2015) – instead of using the 0.05° resolution data set with monthly resolution from 2000–2020. As suggested by Reviewer 1, this enables us to better identify forest-related low-NDVI grid cells. Consequently, the identified events, the climatological reference period, and some of the results of our study will change.

2. **New subsampling to acquire the central Figs. 5 & 6:** To account for the weight of individual event years in the meteorological storylines (because there are much more low-NDVI grid cells in some years than in others), we will do a random subsampling of our event data. For each biome separately, it selects randomly 10 grid cells affected by events to produce a meteorological storyline as in Fig. 5. In some years there will be less than 10 events – depending on the event definition – but the majority of years typically contributes equally to our results.

   1. **Drop Sect. 3.2.3 (event sequences):** As a consequence of the new subsampling strategy, event sequences outside the 2018–2020 period will become extremely scarce. Thus, a robust analysis as for single events will not be possible anymore.

   2. **No wet winter signal:** Our first results, which will be included in the revised version, indicate that the wet winter is not a statistically significant meteorological precursor anymore – mostly due to the new subsampling that reduces the weight of years with many low-NDVI grid cells (e.g. 2018).

3. **New scaling of NDVI data:** Instead of scaling the NDVI anomaly from the climatological mean with the local standard deviation, we will use the anomaly from the median and scale the magnitude of the anomaly by the local interquartile range. This accounts for the bounded nature of NDVI data. Further, we will reduce the dependence of our event definition on scaled anomalies (see next paragraph) and only use them for event characterization.

4. **Measures to reduce the number of threshold parameters used:** Before, we relied on four threshold criteria, namely $FA^{0.05°} \geq 50\%$ (min. forest coverage of NDVI pixels used), $FA^{0.5°} \geq 20\%$ (min. forest coverage of 0.5° grid cells used), $NDVI'^{min} \leq -2$ (event intensity) and $A^{ev} \geq 0.5 * A^{for}$ (min. area affected by an event per 0.5° grid cell). Given the fact that we will completely redo our analyses, we came up with the following measures to reduce the sensitivity of the identified events on threshold parameters:

   1. We will get rid of the condition $FA^{0.05°} \geq 50\%$ by using the **land-cover data set CORINE by Copernicus** (Büttner et al., 2004). It offers a categorical classification of NDVI pixels into forest and non-forest pixels.

   2. We will reduce the remaining three thresholds mainly used for aggregation to two thanks to **a revised event definition** (see also Figure FAC 1 in C3). The new definition is independent of NDVI magnitude, and focuses on the persistence and large-scale nature of the low NDVI events in our focus instead: The identification still follows a two-step approach. First, we identify an event at a forest pixel (250m) if at least four out of six time steps in JJA show a negative NDVI anomaly from the median ("low greenness"). Second, we aggregate to the 0.5° scale by requesting a

minimum of 80% of the forest pixels per 0.5° grid cell to be affected by low greenness (first threshold: $A^{ev} \geq 0.8 * A^{for}$). Lastly, we ensure coherence more robustly by filtering out 0.5° grid cells with low information of forest NDVI, i.e., such cells that have a forest cover lower than 10% (second threshold: $FA^{0.5°} \geq 10\%$).

5.  Both thresholds mentioned in MRP 4.2 will be varied in a **sensitivity analysis** to assess the robustness of our results. The new criteria to identify low-NDVI grid cells also has the advantage of targeting the method even more on our study goal: identifying low forest greenness on a relatively large spatial scale, on which the analysis of meteorological conditions using ERA5 data is meaningful.

In the following, the comments of the reviewers are shown in black and our replies in blue. We group and number reviewer comments for referencing purposes throughout the document (comment 1 = C1, etc.). We further introduce three figures (Figs. FAC 1–3 ) to better illustrate some of the replies. References are listed at the very end of the document.

**Reviewer 1**

**1.** In their paper, Herrmann et al. use MODIS NDVI at 0.05° (roughly 5 km) resolution to characterize 'meteorological storylines' preceding extraordinary forest summer NDVI over the period 2000-2020. The defined events are backed-up using a Landsat-based product which allows for identifying forest disturbance following an event. In addition to studying single events, they also investigate on two consecutive events in a row. The meteorological storylines of the identified events – which represent the core of the study – are quantified using ERA5 reanalysis temperature and precipitation integrated over 90 day seasons to better understand the triggers of extremely reduced canopy greenness. Moreover, they study sea-level pressure-based cyclone-anticyclone frequencies to get an idea of predominant circulation patterns of identified events. In general, the topic of investigation –describing triggers of extreme drought impacts on forest ecosystems – is of broad interest to the public and scientific community and thus deserves publication.

Many thanks for this positive assessment of our overall objective.

However, I yet recognize some major issues that have to be addressed to allow for a scientifically sound research paper. I have to mention, that I already reviewed an earlier submission of this manuscript to another journal, where I already raised quite a few of those issues. Altogether, I acknowledge that the authors have put quite some efforts into the manuscript to tackle some of the points I raised previously. However, some major points of my earlier review have unfortunately not been considered in the revision of the manuscript, which I still deem mandatory for a solid and sound analysis. I will outline those in the following:

**2.** The authors based their analysis on monthly MODIS NDVI with a spatial resolution of 0.05°. As mentioned in my previous review, I wonder why the authors do not use MODIS NDVI at 250 m resolution. In combination with a fine-grained land-cover map (as used in their study) this would allow for masking most of the non-forest areas within the grid-cells under investigation. At current, the roughly 5 x 5 km grid-cells may still contain up to 50% of non-forest area, which may substantially affect the corresponding grid-cell NDVI. As shown in other studies (some of which are referenced by Herrmann et al.), the response of different land-cover types to drought varies substantially. In particular, agricultural land-cover – which probably dominates the noise in the considered pixels – is known to respond earlier to extreme drought and consequently the effect of drought on forests might be overestimated not to mention the effect of different harvest dates if different crops were planted. The authors admit this caveat in section 4.5 and mention that increasing the threshold of forest proportion did not change the identified event hot spots while reducing the sample size. However, reducing the sample size would possibly remove some of the spotted just-significant 'pre-cursors' of droughts (e.g. the just significant percentage of dry periods 24 months prior to an event in Fig. 6) and thus, the meteorological storylines might be interpreted differently. If MODIS at 0.05° resolution were the only available remote sensing product for these analyses, I would probably accept it. But since there is a relatively simple solution to the problem, I believe the authors should do their best to maximize their analytical precision. That is, instead of using the coarse MODIS resolution, simply analyze MODIS at 250 m resolution, mask non-forest areas and then – if desired (but I doubt this is needed and you would lose information) – aggregate the remaining pixels to the target resolution, i.e. 0.05°. By doing so, the precision of identified forest stress would certainly increase, likely resulting in a clearer picture of the whole study since artefacts resulting from non-forest land-cover can be largely ruled out.

We fully appreciate the relevance of the horizontal resolution of the NDVI data. Therefore, we will download and process the 250m resolution NDVI to identify low NDVI events. We will hold on to the aggregation to the 0.5° scale, as our interest and expertise lies in the field of synoptic and mesoscale

weather dynamics. Therefore, we want to contribute to understanding widespread low forest greenness and the degree to which it relates to synoptic meteorology (see MRP 4, MRP 5).

**3.** A less critical – but yet important – point refers to the statistical pre-processing of data, i.e. a standardization of non-normally distributed data (NDVI and precipitation). While the authors stress that the standardization is not used to derive any probabilities or return intervals, relying on normal-distribution related parameters such as mean and standard deviation is inappropriate, even if Shapiro-test indicates normal distribution. A bounded distribution as NDVI (between -1 and +1) cannot be normally distributed. Also, the arbitrarily chosen threshold of -2 suggests that the authors are targeting at the lower margin outside the 95 % confidence interval (-1.96). While this probably does not severely affect the outcome of the analyses, appropriate data treatment should nevertheless be the aim in a scientific study (and is actually easy to obtain). For the NDVI an approach based on proportional differences to the median might render an appropriate alternative solution.

> The reviewer raises a valid concern. Taking several comments of both reviewers together, we decided to use a different event definition that is not based on any intensity measure (see MRP 4.2, Figure FAC 1). This will also make the somewhat arbitrary threshold of what an "intense" or rare event should be – also mentioned by the reviewer – superfluous. We will still use negative anomalies as an indication of low forest greenness, now defined as deviation from the median (see MRP 3). We would like to stress again (as in the paper) that the real challenge is the sparsity of data and that any statistics with few data is prone to substantial sampling uncertainty, i.e., any statistic derived from such small samples must be treated with care.

[Figure]

*Figure FAC 1: Schematic of the two-step event identification. (1) the requirement for a 250m forest pixel to be marked as an event pixel: at least four negative NDVI anomalies in June–August. (2) the aggregation of event pixels to 0.5° grid cells including two thresholds: one for the minimum fractional forest area ($FA^{0.5°}$), and one for the fraction of affected forest area ($A^{ev}$) among the total forest area ($A^{for}$).*

> And lastly, to avoid confusion, note that the Shapiro-Wilk test was used to test whether the normal distribution should be rejected, however, failure to reject that null hypothesis does not *indicate* that the null hypothesis is true.

**4.** In terms of selecting a threshold for defining events a sensitivity analysis should be carried out to reflect the dependence on the selected threshold. This should in any case be done, if selecting any threshold (i.e. also for the currently chosen z-transformed data and -2).

> See C3 above. Our new event identification does not rely on an intensity threshold. For other thresholds used, we will conduct a sensitivity analysis (MRP 4).

**5.** Regarding precipitation, I was wondering why the authors did not utilize some of the more frequently used drought-metrics, e.g. SPI or SPEI which automatically standardize the water balance between precipitation and PET and consequently better resemble actual plant water availability. Recall that 100 mm of precipitation in northern Europe means a lot more water available as plant water in comparison to the same amount in the Mediterranean just because of the large differences in PET.

> This is why we normalized P and T2m with local variability. Assuming that forests have adapted to the local variability in P and T2m over the long run and on the large scale of 0.5° (mentioned in the Sect. 1), standardized deviations from climatology can be equally problematic. We agree that SPI and SPEI would be appropriate alternatives, however in light of our overarching research question regarding "meteorological" storylines, we wanted to start with fundamental meteorological parameters (P and T2m). The variables you suggest are more involved, more directly relevant for plant water availability, but less straightforward for meteorological interpretation.
>
> Note that standardization of meteorological variables is a standard procedure as we use 90-day mean values of all meteorological anomalies. Given the central limit theorem (e.g., Dodge, 2008), 90-day mean values are obtained from many degrees of freedom and – no matter the distribution of the independent (daily to weekly) values – are expected to be close to normally distributed.

**6.** I am not convinced by the meteorological storylines, at least not in the way the authors interpret them. To me, it reads as if the authors believe to have found a 'recipe' for single and consecutive drought events.

> We apologize if we created this impression. This is not at all our understanding of our results. We don't try to present these results as a recipe or "the universal storylines" of low NDVI events. Rather, we present what the data tell us. The storylines (a term, which will be replaced by "history", see C28) are necessarily distinct to a certain degree for each and every event (this is how meteorology works), nevertheless it is valid to search for „statistically significant precursors", i.e., features that are often shared between distinct low NDVI events and thus occur more frequently than expected under climatological conditions. When revising the paper, we will make sure to avoid the impression that we consider our results as a general recipe.

**7.** While I agree to the finding that concurrent (spring-summer) drought results in extraordinarily low NDVI (this has been shown in several studies before) I doubt that a Central European single drought event per se needs previous summer precipitation to be below average and previous winter precipitation to be above average as suggested by Fig. 5 (or taking Fig. 6 a high chance of drought occurrence 2 years in advance of a drought).

> We believe there is a misunderstanding here (see also C6). Our analyses do not indicate that "a Central European single low NDVI event *per se needs* previous summer precipitation to be below average and previous winter precipitation to be above average". That is, no inferences can be drawn from Fig. 5 about strictly necessary conditions for a low NDVI event. Rather, the analyses presented in Fig. 5 aim to identify *statistically significant precursors*. The concept of meteorological precursors is very well established in the meteorological literature and refers to a feature that occurs with a

statistically significantly higher frequency in a certain set of meteorological histories (e.g., those preceding low NDVI events) compared to climatology (e.g., Martius et al., 2008). When attempting to improve the physical understanding of any meteorological phenomenon such precursors are thus interesting features, because they stand out of the noise that invariably exists in any set of meteorological histories preceding certain events. They inform about *shared* characteristics of these histories that possibly (but not necessarily) are physically related to the events of interest. However, such precursors need to be identified first using statistical methods, which is what Fig. 5 presents. In that sense, Fig. 5 simply indicates that the features mentioned by the reviewer (below and above average P in the summer and winter preceding low NDVI events, respectively) occurred at a higher magnitude in the meteorological storylines of our event set compared to what could be expected under climatological conditions.

**8.** It is at least very counterintuitive why an overshoot precipitation in the previous winter should be an important 'ingredient' for a drought event, since it in fact would rather replenish soil-water resources. In contrast, a dry winter prior to an event seems more meaningful as described for the Mediterranean.

> As mentioned in C6 & C7, we documented what the data revealed. We agree that a wet winter is a counterintuitive meteorological precursor for low NDVI events. As counterintuitive results bear the potential of new insight, one should not directly dismiss them – especially given the fact that they emerge from using the same method like other, more intuitive results (concurrent summer drought). Therefore, we consulted the existing literature to suggest two possible mechanisms in Sect. 4.2 via which wet (and mild) winters could negatively affect forest greenness in the following summer. Our first results, which will be included in the revised version, indicate that the wet winter is not a statistically significant meteorological precursor anymore (see MRP 2.2).

**9.** A reason for these partly counterintuitive results is probably the limited number of events underlying these analyses. As pointed out by the authors, 42% of single events refer to the 2018 drought, indicating that this specific event (with a wet winter in advance) has a strong fingerprint on the storylines. Even stronger is the effect of the 2018/2019 consecutive event which renders 82% of consecutive events. Consequently, the meteorological conditions prior to the 2018/2019 event dominate the meteorological storyline for consecutive events, but I am convinced that also other constellations can lead to consecutive extraordinary NDVI.

> The reviewer raises a valid concern here and in the following paragraphs. First, however, we also think there is a misunderstanding, as the numbers above are not cited correctly. We point out that 42% of the event grid cells referred to the low NDVI grid cells in 2018, 2019, and 2020 – not only 2018. Furthermore, 82% of the consecutive events refer to *two* two-year event sequences, namely 2018/19 and 2019/2020 – not only 2018/19.

> Nevertheless, we fully agree that we look at extreme (i.e., rare) events in a short period and therefore individual years can have a strong weight in the overall results. The shorter the data set, the higher the relative contribution from individual years. During our revisions we will make sure that we account for the weight of individual years and stress their contribution to the overall picture more clearly (see MRP 2).

**10.** Given this dominance of the 2018/2019 event on these analyses, I was surprised to see precipitation to be above average in the winter 2018/2019 since in fact this winter was drier than usual in Central Europe, explaining why the drought actually simply went on in 2019 since soil water was not replenished – as usually

– in winter. Therefore, I would be very careful with talking about legacy effects but rather interpret the drought 2019 as an ongoing drought. This can for instance be seen when studying soil water deficit indices or other soil-related drought metrics over the winter 2018/2019. Again, this makes me wonder whether the used climate parameter precipitation is the right candidate for quantifying drought impact. I have to stress, that the authors do discuss the disadvantage of having only a short period for their overall analyses in section 4.5, resulting in the inability to 'perform superior statistical modelling or resolve species specific responses' (the coarse resolution of 0.05° would anyhow not allow for the latter). Unfortunately, they do not mention the potential effect of single events dominating their meteorological storylines even though the numbers are given in section 3.2.3. To account for the dominance-effect, it would be interesting to see the storyline analyses based on a weighed random subsampling of events to avoid dominance of single events in the storylines.

> Thank you for this suggestion. We will use a subsampling method that has the same goal as the suggestion by the reviewer (see MRP 2).

**11.** But all in all, I have doubts that an average storyline– i.e. a 'recipe' – for extreme NDVI exists beyond concurrent spring/summer conditions. The droughts of the past two decades all had in common that summers were extraordinarily warm and dry. However, the timing of drought differed and so did the conditions in the previous winter (warm-wet vs. cool-wet vs. warm-dry) as well as the previous summer. Thus, even if 100-year lasting NDVI time-series were available, I would be surprised if we were able to find a specific storyline resulting in single or consecutive drought events.

> We think we offer a careful analysis that is able to test how far back in time meteorological anomalies prior to low NDVI events stood out from the noise. Again, we thereby do not speculate about the "generic, globally valid storyline for low NDVI events" but rather we evaluate the available data for Europe in the last two decades and with – we think – sophisticated resampling, we can show that, with a relatively high degree of confidence, the signals we find are not just noise. With the new subsampling applied (MRP 2), we strengthen the robustness of our results. In the revised version of the manuscript, we will also discuss the degree to which the interpretation of our results is limited, which is that they are derived from the past 20 years of observations only.

**12.** Finally, I miss the mention of potentially additional noise on the analyses. For instance, in 2019 a late-frost event stroke parts of Central Europe which potentially affected NDVI in summer 2019. Same holds true for the winter-storm early in 2018. This also refers to my mention on the uniqueness of meteorological storylines preceding extraordinary NDVI values. While I agree that the main cause of shown events is spring/summer drought, the possible additional effects of late-frost/storm impacts and others – which is briefly mentioned at the initiation of the discussion – is not well elaborated, at least not in context of the meteorological storylines.

> Summer NDVI is definitely also affected by other disturbances, as highlighted, e.g., in Buras et al. (2021). This is why we think the way we assess significance is such an appropriate tool to *only detect shared meteorological precursors* of many low NDVI grid cells. The more other factors were driving low NDVI events, the less significant T2m and P signals would result from our analysis in Figs. 5 & 6. Our study, however, clearly highlights that there are some shared meteorological precursors. We will try to be, again, more precise with the goal of the presented study, which is to detect plausible meteorological large-scale precursors (magnitude and persistence of T2m and P anomalies) of low forest greenness. Also, we will make sure that other disturbances are discussed to an appropriate degree regarding the scope of our study, as intended in Sect. 4.1 and Appendix B.

I regret, that I cannot be more positive. As mentioned earlier in this review, I appreciate that the authors have refined their analyses in comparison to the last time I reviewed this manuscript: compared to the previous version of the paper, it now reads much clearer and the analyses as well as their presentation have certainly improved. But given the major revision they undertook, I wonder what made the authors decide against a higher spatial resolution of MODIS data, which is freely accessible and also why they stick to the standardization of non-normally distributed data. Both issues could have been dealt with in course of the revision without extreme efforts and would allow for a more precise picture of forest drought meteorological storylines. At current, the selection of events is blurred by effects from non-forest land-cover and potentially inadequate standardization resulting in different absolute variations being treated equally (for instance if the standard deviation differs between pixels which can easily happen given the varying contribution of non-forest land-cover to the pixels but also when comparing different forest types).

**13.** As a way forward, I (again) recommend to utilize high-resolution MODIS products and mask non-forest areas from the analyses. Inappropriate standardization of data should be avoided (suggestions are given in point 2 above). Regarding the meteorological storylines, I recommend to randomly sub-select the data to avoid single events – such as 2018 or 2018/2019 – dominate the storylines.

This may result in a lower overall sample size but that could be accounted for by bootstrapping the random subsamples over several iterations (just make sure this does not result in a stronger contribution of single events, i.e. keep the weighing for each iteration constant).

> Thank you for these suggestions. It is indeed very useful to redo the storylines (Fig. 5) with omitting single years or a subsampling approach as discussed previously (see also MRP 2).

**14.** When interpreting corresponding results, I would highly appreciate if the authors communicate their findings in a less straightforward way, i.e. avoid selling a 'recipe for droughts' but rather tone down to something like: meteorological conditions that have preceded extreme forest drought impacts over the last two decades and recall that under anticipated climate change 'storylines' may change if drought frequencies increase.

> Thanks, as noted in C6, C7, and C11, we will carefully reconsider our wording to avoid the impression of "selling a recipe".

I hope my comments help the authors refining their analyses towards a more robust interpretation. At this stage, I refrain from commenting on textual aspects of the manuscript since I believe that the revisions change a larger part of the methods, results, and discussion.

**Reviewer 2**

**15.** Herrmann et al. analyse storylines of low summer NDVI events in Europe for the period 2000-2020. The storylines are estimated for up to two years prior to the low NDVI events and are analysed separately for the temperate and Mediterranean biomes. This is an extremely relevant topic, since the ongoing disturbances in Europe are in many regions unprecedented (Senf & Seidl, 2021) and the drivers of tree mortality are still elusive (Hartmann et al., 2022). The use of storylines to evaluate drivers of low NDVI events is an innovative and well-fitting approach that could have a high potential to shed light into the general processes driving these events.

> Thank you for this positive overall assessment of the objectives of our study. Note that we will from now on use the term "meteorological history" instead of "meteorological storyline" to more clearly refer to our intention of assessing the characteristics of the meteorological past of low NDVI grid cells (see C28 for a detailed introduction of this term).

After reading the manuscript in detail, I believe the study unfortunately falls short of this ambition for generality, due to a number of fundamental issues with the specific methodological approach, underlying data, and the general lack of mechanistic insights, as discussed in more detail below. Moreover, the reading is at times engaging but at times it can be quite cumbersome due to the (heavy) use of confusing notation, unclear grammar and some distracting sentences that read as if the authors were addressing specific reviewers' comments, perhaps from a previous submission. That said, I believe that the analysis performed here can be of great value to the community, should these issues be addressed.

Major comments

**16.** 1) Generality of the storylines & mechanistic interpretation. At the moment, the results and discussion are presented as if the authors can derive general lessons about meteorological storylines leading to reduced forest greenness in summer.

> This comment was also raised by Reviewer 1. Please refer to C6 & C7 above. In short, we apologize for creating a false impression, as we just let the data tell their story. We do not want to claim any generality for earlier or future events, or for events in other parts of the world.

**17.** Given the short length of the time record (3 years), and that very few (1-3) large-scale events dominate the overall number of events detected – 42% of individual events correspond to the 2018 drought/heat event – it is hard to accept that the conclusions can be generalized to different event types. This is even more problematic for the "consecutive events", which refer basically to the 2018/19 event (making up 82% of the "individual events").

> Unfortunately, there was a misunderstanding here, the above reference of our numbers is not correct (see C9). We will better emphasize in the revised version how the number of events are distributed across the 20 years.

**18.** Therefore, the event samples cannot be treated as individual events. If I understood well (see my comment below) the authors calculate the anomalies for the climate variables by selecting 10k randomly selected samples and calculate their means over the different events for each of the variables. By using a random sampling of $n$ "pixel" events that are dominated by a single large-scale event (or two for the repeated events), a large fraction of the 10k samples will stem from this single event, so that their sampling will necessarily be biased.

We created Figure FAC 2 to increase the comprehensibility of this reply (it shows how the bootstrapping is done for precipitation; see C24 for a new description of the bootstrapping). We think there are two misunderstandings here. First, we do not only sample the low NDVI grid cells during the year of the event in the bootstrapping. We keep the annual chunks of low NDVI grid cells, i.e., the annual event masks (1=event, 0=no event), to extract meteorological anomalies of a randomly selected year in 2000–2020 where mask=1. So the spatial coherence of meteorological anomalies in a specific year is retained over an area defined by low NDVI grid cells in a different year. Therefore, the anomalies in that region are sampled in a year that is not necessarily (and most likely is not) followed by a low NDVI event due to the random shuffling. All in all, our method allows for spatially coherent patterns in the meteorological fields of the randomly sampled years.

[Figure]

*Figure FAC 2: Schematic illustrating of how we acquire the meteorological history (with $n_{dt} = 1'095$ time steps) of the $n_{ev}$ low NDVI grid cells and of the reference for bootstrapping. The outermost columns indicate the time period of the meteorological fields extracted at low NDVI grid cells indicated by the gridded masks in the same row. The gridded masks are colored according to the year when the low NDVI grid cells were identified. Note that for the bootstrapping, the color of the masks and the meteorological fields do not necessarily correspond.*

Second, even if it was the case that we sample grid cells mostly from a single large-scale event, then we would not find *statistically significant* anomalies, because then the random sampling would identify the same "large-scale signal". Fig. 5 clearly looks different than expected in this case.

**19.** It follows that the meteorological anomalies estimated for these storylines will strongly reflect the anomalies of those 1-2 synoptic-scale events. This results in the authors possibly overinterpreting short-term departures of the storylines from the 95% confidence interval, particularly the rainy previous winter season – that occurred in much of the region affected by the 2018 event (https://climate.copernicus.eu/european-wet-and-dry-conditions) – as a general percussor of low summer NDVI events. It is unclear what would be the mechanistic link between rainy winters and dry/hot summers leading to low NDVI events and the authors do not offer satisfactory explanations (rather list a number of hypothetical processes in the discussion).

We agree that due to the weight of 2018, the wet winter signal stood out as a significant meteorological precursor that is shared among the low NDVI grid cells averaged in Figs. 5 & 6. In contrast, the new subsampling of low NDVI grid cells promoted by both reviewers results in a meteorological history without the counterintuitive wet winter signal (see MRP 2). However, this is a result of more carefully assessing the meteorological history that is tested in the bootstrapping,

and not a shortcoming of the bootstrapping test. Because generally, we are confident that departures from the confidence interval are significant because the resampling is done in a way that we use the same area affected in certain years also in the bootstrapping reference. Therefore, a large-scale anomaly can affect also the resampled histories, in a similar fashion as is the case in the observed event history (see C18 and C24). Finally, even if one year contributes strongly to the meteorological histories, they need not be the same everywhere within the event. For example, in 2018 grid points from southern Scandinavia to France and to the Balkans were affected (Fig. 3a).

We also agree that we could not offer an explanation for the role of the mild and wet winters and rather refer to existing literature (see C8). But this is a very common feature in climate science that some studies identify signals in observations and/or simulations (e.g., many important trend studies), and these studies are valuable although they hardly ever can "explain" the reasons for the identified signals. So, in our view, there is nothing wrong / bad with presenting our results to the forest and climate community as an invitation to either falsify or explain our results. We further elaborate on this in similar comments of Reviewer 1 in C6-C9.

**20.** I would be curious to whether the authors could find significant signals, especially the rainy winter peak before the low NDVI events in temperate regions, if events occurring in 2018 are removed from the analysis.

We like this suggestion, which is similar to a suggestion from Reviewer 1 and we will use a random sampling of similar sample sizes per year (see MRP 2).

The authors could also try an alternative sampling approach that would explicitly correct for the sampling bias due to the predominance of few large-scale synoptic events, but this would reduce drastically the number of events (by about half for EV10 and by 80% for EV11).

**21.** An alternative could be to tone down the ambition to derive general conclusions for 2000-2020 and rather focus on highlighting the value of the storyline approach to learn about anomalies in vegetation activity and, specifically, extremes. While the results for 2018/19 in central Europe are not necessarily new, the fact that the approach proposed here shows consistent results past studies while providing a means to assess more generally past history is a strength that could/should be stressed. For the Mediterranean case, the results might also reflect the year 2017 the most (given figure 3d), so that a similar approach/reasoning can be applied.

Thank you, we will rephrase our conclusions to reflect this valuable input in addition to the new sampling method applied.

**22.** 2) Spatial resolution. The authors analyse storylines leading to low NDVI extremes, focusing on forest ecosystems. To do this, they rely on NDVI data from MODIS at 0.05km. This is already a relatively coarse resolution for the analysis proposed, as pointed out by Reviewer 1, but the authors then further coarsen the data in the analysis to 0.5 degree spatial resolution. The authors tried to minimize this issue by estimating low NDVI values first at the 5km resolution and then imposing the condition that at least 50% of the coarser grid cell needs to register low NDVI events. However, their definition of "forest pixels" includes pixels with less than 50% of forest, and even as low as only 20% (Section 2.1). This results in many areas that are dominated by croplands or mixed tree/herbaceous cover (parts of central and eastern Europe and of Italy, southern Sweden), being included as "forests".

It is worth mentioning first that we are using 250m resolution NDVI data in the first place (MRP 1). Furthermore, we will now use a new land cover data set to categorically identify forest areas (MRP

4.1). Consequently, we can much more reliably identify 250m pixels and 0.5° grid cells where forest was affected by low NDVI, and, thus, these concerns no longer apply to the new methodology.

Nevertheless, for clarification, we think that there was a misunderstanding in the above-cited numbers. We only considered 0.05°x0.05° pixels that are at least 50% covered by forests. The threshold of 20% forest area applied to the 0.5° grid cells considered. Only 0.5° grid cells with more than 20% forest area are considered at all. The eventual NDVI signal considered within these grid cells includes forest pixels only, i.e., they each must have at least 50% forest coverage.

**23.** While I understand that the authors could not perform this analysis at the higher MODIS spatial resolution given the lack of such high-resolution climate data over Europe, it is unclear why the authors chose to aggregate results to 0.5 degrees, since ERA5's native resolution is 0.25 degree, and ERA5 land provides downscaled fields at 0.1 degree spatial resolution, much closer to the spatial scale used to define events and to distinguish (at least better) forest from non-forest pixels.

The official ECMWF webpage states that "The data [ERA5] cover the Earth on a 30km grid". The grid spacing is not the same as effective resolution, which is typically about 4 times the grid spacing (since 4 grid points are required to resolve a wavelength). In principle one can download ERA5 data on any grid, e.g., a 0.25° instead of a 0.5° grid, but this does not increase the information content of the data, it rather oversamples the data. This is also why we only found small differences in a comparison of seasonal mean P and T2m fields in ERA5 data at 0.25° vs. at 0.5°.

Since ERA5-Land is based on the same spectral truncation of the ECMWF model, it is not clear whether the 0.1° grid provides meaningful additional information. Note that ERA5-Land does not include any correction of the precipitation by the ERA5 reanalysis (Muñoz-Sabater et al., 2021).

The same for E-OBS, and the group of COSMO-REA reanalyzes provide temperature and precipitation fields at 6km or even 2km. Surely, repeating the analysis at such fine resolutions (2km) would be more computationally expensive, so that keeping the 5km scale of the initial steps is probably the most feasible option.

**24a.** 3) Synthetic event sets and meteorological means. In appendix A and the methods, it is generally unclear if the 10k event sets are derived per pixel individually, as suggested by keeping *n* in subscript (appendix A), or across pixels or across individual events (i.e. pairs of (pixel, year)). It is also unclear how the authors calculate the means and climatologies of the meteorological variables (Line 562-63). Generally, this makes it difficult to evaluate and reproduce the analysis.

Thank you for mentioning the difficulties in understanding the details of our method. We provide detailed information below and will include more details in the revised manuscript.

**24b.** The authors state that they take 10k synthetic event sets and then (1) take the mean among all events, then (2) take a distribution of these mean values that is used to calculate climatologies. I have several questions about this step:

- It is not clear what is the size of the event set matrix, is it 3d (10k, 21, n)?
  - The reference event matrix is a 3D $s \times g \times n_{evs}$ matrix, where $s = 10'000$ corresponds to the number of random samples, $g = 3$ corresponds to the grid point identifiers year, latitude and longitude, and $n_{evs}$ corresponds to the total number of low NDVI grid cells (which was 1'228 and 303 for the temperate and Mediterranean biome, respectively).

- And over which dimension is the first mean calculated, the second?
  - For every sample we extract meteorological fields in the respective year, lat, lon combination indicated by the matrix above. The resulting matrix, e.g., for temperature, is 3D again with size $s \times n_{evs} \times n_{dt}$. It includes $T2m'_{90d}$ for $s = 10'000$ synthetic samples, including $n_{evs}$ events, over $n_{dt} = 1'095$ timesteps (three years). The first mean is taken along the second axis (event mean, including lat-weighing). The resulting $s \times n_{dt}$ matrix is used to calculate the 2.5th and 97.5th percentile along its first axis. The outcome is a time-evolving confidence interval with $n_{dt}$ entries (see the more elaborate description below).
- What is the size of the resulting vector/matrix?
  - See above
- Do the climatologies in T and P have seasonality?
  - Yes they do: For each calendar day, the climatology $\overline{P_{90d}}$ is equal to the mean over the 21 annual $P_{90d}$ values at that calendar day. $P_{90d}$ itself is the mean over the past 90 days, exerting seasonality as it is a moving average. Consequently, also $\overline{P_{90d}}$ is changing for every calendar day (see also Sect. 2.4.1).
- Over which dimension is the standardization performed?
  - The standardization is performed as initial data treatment prior to the bootstrapping. At every grid point there are 21 $P_{90d}$ values per calendar day which are used to calculate mean and standard deviation for local standardization. As for the climatologies above, the period used is Sep 2000 – Aug 2020.

For the consecutive event sets, how exactly is this done, do you select from EV10 subsets of synthetic events that happen consecutively? Where are the values of the different n values derived from? Finally, the authors mention in the methods section that the sampling strategy preserves the spatial correlation of T and P fields, but this is not discussed in Appendix A.

**24c.** As a consequence of these issues, it is difficult to evaluate the robustness of the method described. I suggest either describing the steps using explicitly mathematical notation, reporting the sizes of the vectors/matrices, and/or a flowchart to facilitate understanding of the exact steps performed here. It would also be good to discuss the rationale behind using bootstrapping in the methods and start of Appendix A.

> This reviewer comment clearly indicates that our explanation of the method and statistical approach was not sufficiently clear, and we thank the reviewer for her/his detailed questions. We hope to have answered the questions above sufficiently and will include a revised description – drafted below – including a schematic to better explain the bootstrapping, including some more details as suggested by the reviewer.

[Figure]

**Figure FAC 3:** *Updated Figure FAC 2 for the new study period and resampling number. The retrieval of the meteorological histories is shown for $P'_{90d}$ and is done analogously for $T2m'_{90d}$. The number of low NDVI grid cells is referred to as $n_{ev}$ and the number of time steps of each meteorological history (of the events or the bootstrap reference) is $n_{dt} = 1'095$.*

*Text to be included in the revised manuscript:* Evaluating the meteorological fields for event set $EVS$ results in an $n_{ev} \times n_{dt}$ matrix for each variable, $T2m'_{90d}$ and $P'_{90d}$. Together the two matrices represent the meteorological history of all $n_{ev}$ low NDVI grid cells including $n_{dt} = 3 \times 365$ daily time steps. In Fig. 5, for example, we display the mean along the first dimension of these matrices – the event-mean meteorological storyline.

For null hypothesis $H_{0,EV}$, we generate 1'000 synthetic event sets $EVS^r$ by randomly shuffling the 21 annual chunks of $EVS$ 1'000 times. This shuffling process is best visualized by shuffling a deck of cards, whereby each card corresponds to a the binary event map of a specific year. Specifically, when constructing the random event set with number $r$ ($EVS^r$) we assign a randomly selected year $y_i^r$ to all low NDVI grid cells occurring in year $y_i$ and then repeat the process for all remaining years. Hereby the random years are chosen such that each year occurs once in every $EVS^r$. Consequently, each reference event set contains the same number of low NDVI grid cells as $EVS$ but in a different year-location combination. Afterwards, the synthetic meteorological histories are generated by first extracting ERA5 fields for $EVS^r$ (i.e., using the shuffled deck of annual binary event maps). Then, the resulting $1'000 \times n_{ev} \times n_{dt}$ matrix is averaged along its second dimension to retrieve a set of 1'000 synthetic event-mean histories. We, thereby, create 1'000 meteorological histories that occurred at some time in the climatological reference period without the prerequisite of a following low NDVI.

We then compare event-mean $T2m'_{90d}$ and $P'_{90d}$ of low NDVI grid cells to the 1'000 synthetic event-means $T2m''^r_{90d}$ and $P''^r_{90d}$ respectively. Values outside the range of $T2m''^r_{90d}$ and $P''^r_{90d}$ receive a p-value of 0. The remaining p-values are estimated from the percentiles of the synthetic data set along its first dimension. At the significance level of $\alpha = 5\%$, $H_{0,EV}$ is rejected at time lags $\Delta t$ if the event value, e.g., $P'_{90d}$, is outside the 95% confidence interval of the 1'000 reference values, e.g., $P''^r_{90d}$.

Note that the shuffling of years is done prior to extracting the spatial fields of $T2m'_{90d}$ and $P'_{90d}$ from the ERA5 data set. This has – in contrast to a random sampling of all forest grid cells – the convenient effect that spatial correlation in these two meteorological variables is retained. Thus, synthetic meteorological histories are constructed from a data set with exactly the same spatial correlation of $T2m'_{90d}$ and $P'_{90d}$ as the original data set.

Note to the reviewer: We reduce the number of reference histories in the bootstrapping to 1'000, as the confidence intervals are hardly changing when further increasing the number of samples.

**25.** 4) Use of forest disturbance dataset. It is unclear why the authors introduce a whole new dataset that is only used for cross-comparison purposes. Low NDVI events do not necessarily have to correspond to crown-mortality events (D), and conversely crown mortality events might not necessarily result in low NDVI if understory vegetation benefits from the canopy opening. This discrepancy is briefly mentioned by the authors, and very clear in Figure 4, but not fully addressed. Another point not really mentioned is that D can be affected strongly by anthropogenic signals, such as management and selecting logging. Since the authors presented the goals of the analysis as to evaluate extremes in vegetation greenness, a clearly well-defined and constrained problem, I find that the comparison with D adds more confusion, rather than clarity to the study.

We think that this brief comparison is useful, because it supports the notion that observed forest disturbances are indeed much more likely to occur in grid cells where we identify low NDVI events. However, to accommodate the reviewers' comment, we will shift this comparison to an Appendix as it is indeed not essential for constructing the meteorological histories.

**26.** Specific comments:

Thank you very much for your careful reading and the suggestions below. We will consider them all in our revision and reply to them individually together with the submission of the revised manuscript as many text passages are likely to change. At the current stage we only address some of the more general comments below.

**27.** "forest performance" is not a standard expression and is rather unclear whether the authors mean vitality, health, or any other aspect reflected by NDVI (vegetation cover, LAI, ...). To be accurate, the best would be to stick to "forest greenness".

Thank you, this is a good suggestion, and we will stick to forest greenness.

Line 23: Over what time-scales is this sentence referring to? This is not true for the past several decades – acid rain, changes in management, reforestation, elevated CO2, nutrient deposition, ... there is a long list of processes that have been destabilizing forests in Europe.

Line 48-50: but in Mediterranean regions, drought can also reduce fuel load (Pausas and Ribeiro, 2013).

Line 56: intensively discussed... in the literature?

Line 59: "stressed", or "identified"?

Line 60: "drought prone region", such as the Mediterranean?

Line 63: "margins" of the growing season is an unusual expression

**28.** Line 65: increasing understanding "of ...", specify what is meant here. Furthermore, the concept of storylines should be described in more detail and appropriately referenced (e.g. Shepherd et al. 2018)

> We realize our use of "storyline" can be confused with the established storyline concept of Shepherd et al. (2018). Even though we use an (extreme) event-oriented approach, we do not intend to address the plausibility and the causality of the meteorological storylines we find, nor to entangle *all* the drivers of low NDVI events. Therefore, we will no longer use the terminology "meteorological storyline" but rather "meteorological history", which better focuses on the past evolution and the occurrence of statistically significant precursors (of T2m and P).

Line 73-74: Other studies have attributed this to a Wave-7 pattern and a positive NAO phase (Drouard et al., 2019 and Kornhuber et al., 2019)

Line 84: what do the 90 in subscript stand for, 90-day moving average not yet mentioned, making it confusing.

**29.** Line 85: why 3 year only?

> This choice is guided by two opposing constraints: (1) Going back longer potentially shows more significant results in Figs. 5 & 6. However, (2) the length of the data record becomes more critical the longer you go back. The chance that mechanistically unlinked events appear in one meteorological history increases with the length of the history. Moreover, our results in Figs. 5 & 6 indicate that more than 3 years in the past the meteorological signals are not distinguishable from noise anymore.

Lines 87-88: I propose swapping (2) and (3)

Lines 101-102: why the choice of this specific domain and why ignoring boreal forests?

Figure 1 caption: please add a brief description of panel b) and it is impossible for the reader to understand it without reading the methods.

Line 106: no justification about why 0.5 degree is used.

Line 116: "at forest pixels" does not seem grammatically correct

Line 119: this is the first time missing values are referred to. Where do they stem from? The use of quality control flags? And what if there are two consecutive months missing?

Line 123: in mathematical notation, the apostrophe is usually used to express the first derivative, so this notation is confusing. Why not a for anomaly?

Line 132: replace "at" by "for" or "in"

Line 134: there is no scheme presented here. Do the authors mean the "approach presented here"?

Line 135: is it a "forest grid cell" or an "atmospheric grid cell over forested pixels"? Overall, I find these definitions confusing.

Line 136 and Equation 2: how does the flag work when there at 3 months in the season? Since the authors take the minimum value over the season, does this mean that it is enough that 1 month in JJA is flagged as low NDVI? How can the authors be confident that this is a "low NDVI season"?

Line 148: correct to "reanalysis", singular

Line 149: why interpolating ERA5 to 0.5 degree?

Line 151: if this refers to seasonal averages, should the subscript be 90d or "season"?

Line 151-159: it is not fully clear if the standardization is done also for the 90d moving windows, please clarify.

Line 167: so only 40% of the values are "not extreme"?

Line 167-169: give correlation values and respective significance

Figure 2: what do the colorbars indicate? Not mentioned in the caption. Line 181: this is not described in the Appendix A.

Line 191: for a Biogeosciences audience, is would be good to explain what the "outermost closed SLP contour ..." means.

**30.** Line 191-195: more generally, for a Biogeosciences audience it would be good to explain what additional information does this analysis bring.

> Thank you very much for the two comments above. These hints are very helpful for our goal to establish a bridge between two different research fields. We will definitely consider them.

Line 197: grammar "To evaluate", "for" + "ing" does not express purpose/intention.

Line 201: which is aggregated and normalized. Why is the normalization now done at the coarser resolution?

Line 216: correct "succeeding" to "subsequent"

Section 3.1.1 – if the purpose of using the D dataset is for evaluation of low NDVI events, why not compare the annual variability in D as well here and in Figure 3?

Figure 3b: the colors in the 4 quads are hardly distinguishable

Figure 3d: if the authors decide to keep D, then add the extent affected by D events in this panel as well.

Line 249: these "conceptual, technical and physical reasons" are not really thoroughly discussed in Sec 41.

Figure 4: What is D'[]?

Figure 5: mark in shaded areas the periods when the event mean is outside of the 95% CI. What do the vertical lines in a-d indicate?

Line 269: negative, but still within the 95% CI. Here, and elsewhere, the authors over-emphasize non-significant results.

Line 271: for short periods. Add "is significant for Xdays ... "

Line 274: continuously negative, but still within the 95% CI. Please give duration of the periods when event mean is outside of the 95% CI.

Line 276: negative in the previous winter, but not extreme. Line 280: DJF of which year?

Line 284: can you give a mechanistic explanation for this?

Line 304: add ", respectively," between "P'90d" and "from" Line 315: the accumulation of dry periods is not significant

Section 3.2.3: please state clearly that this applies basically to 2018/19 in temperate regions, and make a similar assessment for the Mediterranean biome.

Line 331: the information about the fraction of these years to the event samples needs to be given much earlier in the manuscript, and please add information for the Mediterranean biome too.

Line 334: why is 2020 excluded?

Line 340: "hot" anomalies?

Line 350: again, I find in-depth mechanistic interpretation of these patterns lacking in the discussion.

Line 355: "exerts" is not applicable here, since the NDVI events have no influence on T90.

Line 356: I quite like that the authors here give specific values of the anomalies discussed. This should be done throughout the whole results section.

Line 358: what does "small" mean? That the absolute value is close to zero?

Line 364: only T, see comment above for line 315.

Figure 8: please explain what the different color shades mean (95%CI, I believe)

Discussion Section: it is surprising that the limitations related to the short temporal records and the dominance of single years in the events analysed are not discussed.

Lines 414-325: How does the analysis done here "help to characterize the nature of these events"? If this would be true, I would expect separate analyses for pixels with low NDVI and no crown mortality and pixels with low NDVI and crown mortality. Overall, this paragraph is quite distracting given that the main goal of the paper was to analyse low NDVI extremes.

Line 432-433: the grammar can be improved Line 437: regadless of what?

Line 437-438: do the authors mean that drought early in the growing season directly "damages" forests, or simply that low P in spring promotes drier summers?

Line 440: unclear why warming in the previous 3 years would affect an instantaneous process like fire.

Line 485: can you explain in more detail how acclimation results in reduced leaf area and productivity?

Line 488-490: isn't this simply a consequence of the fact that only low NDVI events were selected? The authors did not evaluate post-event recovery trajectories separately, so that they cannot know whether increased vulnerability out competes acclimation.

Line 491: sensitivity to drought is not shown in the results, what do the authors mean here?

Line 501: what does "superior statistical modelling" mean?

Line 508: "not shown", please add these results to the supplement and this is an important point.

Conclusions

**31.** I find that the authors overemphasize the winter wet signal in the temperate biome, which first is rather short (a couple of days outside of the CI) and second cannot likely be generalized for all events. This should be toned down.

> We have addressed the counterintuitive wet winter signal in many replies above. Additionally, we would like to remind the reviewer here, that even a single value outside the confidence interval represents a 90-day average of T2m or P. The moving average simply enables to capture the exact 90-day period when any meteorological anomaly was significantly different from climatology.

Appendix A

Line 557: the superscript *r* should be placed above *n*, and be defined again in the text here, for those readers who might start here.

Line 571: Add "for" before the "null hypothesis". Also, EV10 and EV11 have not been defined previously in the appendix, making reading confusing.

References

Shepherd, T.G., Boyd, E., Calel, R.A. *et al.* Storylines: an alternative approach to representing uncertainty in physical aspects of climate change. *Climatic Change* **151,** 555–571 (2018). https://doi.org/10.1007/s10584-018-2317-9

Pausas, J.G. and Ribeiro, E. (2013), Fire and productivity. Global Ecology and Biogeography, 22: 728-736. https://doi.org/10.1111/geb.12043

Drouard, M., Kornhuber, K., & Woollings, T. (2019). Disentangling dynamic contributions to summer 2018 anomalous weather over Europe. Geophysical Research Letters, 46, 12537– 12546. https://doi.org/10.1029/2019GL084601

Kornhuber, K., Osprey, S., Coumou, D., Petri, S., Petoukhov, V., Rahmstorf, S., & Gray, L. (2019). Extreme weather events in early summer 2018 connected by a recurrent hemispheric wave-7 pattern. *Environmental Research Letters*, *14*(5), 054002.

**References:**

Büttner, G., Feranec, J., Jaffrain, G., Mari, L., Maucha, G., & Soukup, T. (2004). The CORINE land cover 2000 project. *EARSeL eProceedings*, *3*(3), 331-346. Available online at: http://eproceedings.uni-oldenburg.de/website/vol03%5F3/03%5F3%5Fbuttner2%2Ehtml

Buras, A., Rammig, A., & Zang, C. S. (2021). The European Forest Condition Monitor: Using Remotely Sensed Forest Greenness to Identify Hot Spots of Forest Decline. Frontiers in plant science, 12: 689220. https://doi.org/10.3389%2Ffpls.2021.689220

Didan, K. (2015). MOD13Q1 MODIS/Terra Vegetation Indices 16-Day L3 Global 250m SIN Grid V006. NASA EOSDIS Land Processes DAAC. https://doi.org/10.5067/MODIS/MOD13Q1.006

Dodge, Y. (2008). Central Limit Theorem. In: The Concise Encyclopedia of Statistics. Springer, New York, NY. https://doi.org/10.1007/978-0-387-32833-1_50

Martius, O., Schwierz, C., & Davies, H. C. (2008). Far-upstream precursors of heavy precipitation events on the Alpine south-side. Quarterly Journal of the Royal Meteorological Society, 134(631), 417–428. https://doi.org/10.1002/qj.229

Muñoz-Sabater, J., Dutra, E., Agustí-Panareda, A., Albergel, C., Arduini, G., Balsamo, G., Boussetta, S., Choulga, M., Harrigan, S., Hersbach, H., Martens, B., Miralles, D. G., Piles, M., Rodríguez-Fernández, N. J., Zsoter, E., Buontempo, C., and Thépaut, J.-N. (2021): ERA5-Land: a state-of-the-art global reanalysis dataset for land applications, Earth Syst. Sci. Data, 13, 4349–4383. https://doi.org/10.5194/essd-13-4349-2021

---

## Author Response (AR1)

egusphere-2022-425
**Reply document**

Original title:

**Analysis of multi-seasonal meteorological storylines leading to reduced forest greenness in Europe in 2000-2020**

In response to the reviewers' comments, we changed the title to:

**Meteorological history of low forest greenness events in Europe in 2002-2022**

*Reply to both reviewers by Mauro Hermann, Matthias Röthlisberger, Arthur Gessler, Andreas Rigling, Cornelius Senf, Thomas Wohlgemuth, and Heini Wernli*

We acknowledge the reviewers for their careful reading and detailed commenting of our manuscript. We think they have helped to greatly improve the manuscript, which now comes with a state-of-the art NDVI data set, a more robust and careful methodology, and a much clearer storyline guided by the main novel aspect of the study: a systematic assessment of multi-year meteorological precursors to low NDVI events in Europe. The reviews and, thus, the undertaken revisions were very profound, and, thus, except for the introduction and parts of the discussion, the entire manuscript has changed (making the track-changes file very extensive). We gave our best to make clear references to the relevant parts of the manuscript in this reply document and would like to point out that some comments or questions by the reviewers do not apply to the current manuscript any longer. As in the final author comments, we list the major revisions (MR) before answering in more detail to the individual reviewer comments. Details of these major revisions are provided in the replies or in the referenced parts of the manuscript:

MR1.  **New MODIS NDVI data set (L. 111ff):** We addressed the most major critique of the reviewers by downloading NDVI at ~250m spatial resolution and 16-day temporal resolution from 2002–2022 (Didan, 2015).

MR2.  **Sub-sampling to reduce weight of individual event years (L. 270ff):** In this way we account for the weight of individual event years in the meteorological histories (because there are much more low-NDVI grid cells in some years than in others); we now do a random sub-sampling of our event data.

    a.  **Delete Sect. 3.2.3:** As a consequence of the sub-sampling, event sequences outside the 2018–2020 period are extremely scarce. A robust analysis of event sequences was therefore not possible anymore.

    b.  **No wet winter signal:** The wet winter was not identified as a statistically significant meteorological precursor of low NDVI events anymore because the sub-sampling reduced the weight of the event in 2018, which was preceded by a wet anomaly in many parts of Europe.

MR3.  **Scaling of NDVI data (L. 124ff):** Instead of scaling the NDVI anomaly from the climatological mean with the local standard deviation, we use the anomaly from the median and scale the magnitude of the anomaly by the local interquartile range. This accounts for the bounded nature of NDVI data. Further, we reduce the dependence of our event definition on scaled NDVI (see next MR).

MR4.  **More careful use of threshold parameters:**

a. We introduce a new **land-cover data set termed CORINE by Copernicus** (Büttner et al., 2004) in Sect. 2.1 (L. 101ff). It offers a categorical classification of NDVI pixels into forest and non-forest pixels. The threshold parameter $FA^{0.05°} \geq 50\%$ became superfluous.

b. We introduce **a revised event definition in Sect. 2.3 (L. 130ff)**. The new definition is independent of NDVI magnitude and focuses on the persistence and large-scale nature of the low NDVI events instead.

c. All thresholds used are varied in a **sensitivity analysis (Appendix A, short summary in L. 150ff)** to assess the robustness of our results. The sensitivity of our results to reasonable variations in the threshold parameters proves to be very low.

MR5. We use the term "meteorological history" to replace "meteorological storyline" to avoid confusion with the storyline approach introduced by Shepherd et al. (2018) – see C34. We therefore also changed the title of the paper.

In the following, the comments of the reviewers are shown in black and our replies in blue. We group and number reviewer comments for referencing purposes throughout the document (comment 1 = C1, etc.). Literature references refer to those listed below and by Reviewer 2.

**Reviewer 1**

**1.** In their paper, Herrmann et al. use MODIS NDVI at 0.05° (roughly 5 km) resolution to characterize 'meteorological storylines' preceding extraordinary forest summer NDVI over the period 2000-2020. The defined events are backed-up using a Landsat-based product which allows for identifying forest disturbance following an event. In addition to studying single events, they also investigate on two consecutive events in a row. The meteorological storylines of the identified events – which represent the core of the study – are quantified using ERA5 reanalysis temperature and precipitation integrated over 90 day seasons to better understand the triggers of extremely reduced canopy greenness. Moreover, they study sea-level pressure-based cyclone-anticyclone frequencies to get an idea of predominant circulation patterns of identified events. In general, the topic of investigation –describing triggers of extreme drought impacts on forest ecosystems – is of broad interest to the public and scientific community and thus deserves publication.

**Authors:** Many thanks for this positive assessment of our overall objective.

However, I yet recognize some major issues that have to be addressed to allow for a scientifically sound research paper. I have to mention, that I already reviewed an earlier submission of this manuscript to another journal, where I already raised quite a few of those issues. Altogether, I acknowledge that the authors have put quite some efforts into the manuscript to tackle some of the points I raised previously. However, some major points of my earlier review have unfortunately not been considered in the revision of the manuscript, which I still deem mandatory for a solid and sound analysis. I will outline those in the following:

**2.** The authors based their analysis on monthly MODIS NDVI with a spatial resolution of 0.05°. As mentioned in my previous review, I wonder why the authors do not use MODIS NDVI at 250 m resolution. In combination with a fine-grained land-cover map (as used in their study) this would allow for masking most of the non-forest areas within the grid-cells under investigation. At current, the roughly 5 x 5 km grid-cells may still contain up to 50% of non-forest area, which may substantially affect the corresponding grid-cell NDVI. As shown in other studies (some of which are referenced by Herrmann et al.), the response of different land-cover types to drought varies substantially. In particular, agricultural land-cover – which probably dominates the noise in the considered pixels – is known to respond earlier to extreme drought and consequently the effect of drought on forests might be overestimated not to mention the effect of different harvest dates if different crops were planted. The authors admit this caveat in section 4.5 and mention that increasing the threshold of forest proportion did not change the identified event hot spots while reducing the sample size. However, reducing the sample size would possibly remove some of the spotted just-significant 'pre-cursors' of droughts (e.g. the just significant percentage of dry periods 24 months prior to an event in Fig. 6) and thus, the meteorological storylines might be interpreted differently. If MODIS at 0.05° resolution were the only available remote sensing product for these analyses, I would probably accept it. But since there is a relatively simple solution to the problem, I believe the authors should do their best to maximize their analytical precision. That is, instead of using the coarse MODIS resolution, simply analyze MODIS at 250 m resolution, mask non-forest areas and then – if desired (but I doubt this is needed and you would lose information) – aggregate the remaining pixels to the target resolution, i.e. 0.05°. By doing so, the precision of identified forest stress would certainly increase, likely resulting in a clearer picture of the whole study since artefacts resulting from non-forest land-cover can be largely ruled out.

**Authors:** We fully appreciate the relevance of the horizontal resolution of the NDVI data. Therefore, we downloaded and processed the 250 m resolution NDVI to identify low NDVI events (L. 112). We hold on to the aggregation to the 0.5° scale (L. 133, L. 143), as our interest and expertise lies in the field of synoptic and

mesoscale weather dynamics. Therefore, we want to contribute to understanding widespread low forest greenness and the degree to which it relates to synoptic meteorology (see MR 4).

**3.** A less critical – but yet important – point refers to the statistical pre-processing of data, i.e. a standardization of non-normally distributed data (NDVI and precipitation). While the authors stress that the standardization is not used to derive any probabilities or return intervals, relying on normal-distribution related parameters such as mean and standard deviation is inappropriate, even if Shapiro-test indicates normal distribution. A bounded distribution as NDVI (between -1 and +1) cannot be normally distributed. Also, the arbitrarily chosen threshold of -2 suggests that the authors are targeting at the lower margin outside the 95 % confidence interval (-1.96). While this probably does not severely affect the outcome of the analyses, appropriate data treatment should nevertheless be the aim in a scientific study (and is actually easy to obtain). For the NDVI an approach based on proportional differences to the median might render an appropriate alternative solution.

**Authors:** The reviewer raises a valid concern. Taking several comments of both reviewers together, we decided to use a different event definition that is not based on any intensity measure (see MR 4b, Fig. 1b, Sect. 2.3). This also made the somewhat arbitrary threshold of what an "intense" or rare event should be – also mentioned by the reviewer – superfluous. As we still use negative anomalies as an indication of low forest greenness, we defined it as deviation from the median (see MR 3, L. 124). We would like to stress again that the real challenge is the sparsity of data and that any statistics with few data is prone to substantial sampling uncertainty, i.e., any statistic derived from such small samples must be treated with care.

**4.** In terms of selecting a threshold for defining events a sensitivity analysis should be carried out to reflect the dependence on the selected threshold. This should in any case be done, if selecting any threshold (i.e. also for the currently chosen z-transformed data and -2).

**Authors:** See C3 above. Our new event identification does not rely on an intensity threshold. For other thresholds used, we conduct a sensitivity analysis (MR 4c). The sensitivity analysis is presented in Appendix A and summarized in the manuscript on L. 150–157.

**5.** Regarding precipitation, I was wondering why the authors did not utilize some of the more frequently used drought-metrics, e.g. SPI or SPEI which automatically standardize the water balance between precipitation and PET and consequently better resemble actual plant water availability. Recall that 100 mm of precipitation in northern Europe means a lot more water available as plant water in comparison to the same amount in the Mediterranean just because of the large differences in PET.

**Authors:** We agree that SPI and SPEI would be appropriate alternatives, however in light of our overarching research question regarding "meteorological" histories, we wanted to start with fundamental meteorological parameters (P and T2m). The variables you suggest are more involved, more directly relevant for plant water availability, but less straightforward for meteorological interpretation. To account for the weight of a given anomaly in P or T2m relative to the local climate, we normalized P and T2m with local variability. Assuming that forests have adapted to the local variability in P and T2m over the long run and on the large scale of 0.5° (mentioned in L. 27), this approach is an appropriate way to do meteorology-based analyses.

Note that standardization of meteorological variables is a standard procedure as we use 90-day mean values of all meteorological anomalies. Given the central limit theorem (e.g., Dodge, 2008), 90-day mean values are obtained from many degrees of freedom and – no matter the distribution of the independent (daily to weekly) values – are expected to be close to normally distributed.

**6.** I am not convinced by the meteorological storylines, at least not in the way the authors interpret them. To me, it reads as if the authors believe to have found a 'recipe' for single and consecutive drought events.

**Authors:** We apologize if we created this impression. This is not at all our understanding of our results. We don't try to present these results as a recipe or "the universal history" of low NDVI events. Rather, we present what the data tell us. The histories are necessarily distinct to a certain degree for each and every event (this is how meteorology works), nevertheless it is valid to search for „statistically significant precursors", i.e., features that are often shared between distinct low NDVI events and thus occur more frequently than expected under climatological conditions.

To avoid such misinterpretation, we now strictly use past tense throughout the result section to underline that we refer to past observations. Furthermore, we greatly improved the terminology regarding the statistically significant parts of the meteorological histories; we introduce the term "meteorological precursor" explicitly in L. 87, which is an observed (!) characteristic of events that occurred in the study period of 2002–2022 (see C7). In the introduction, discussion, and conclusion sections we make sure to refer to this study period, to avoid the false impression of speaking for low NDVI more generally.

**7.** While I agree to the finding that concurrent (spring-summer) drought results in extraordinarily low NDVI (this has been shown in several studies before) I doubt that a Central European single drought event per se needs previous summer precipitation to be below average and previous winter precipitation to be above average as suggested by Fig. 5 (or taking Fig. 6 a high chance of drought occurrence 2 years in advance of a drought).

**Authors:** We believe there is a misunderstanding here. Our analyses do not indicate that "a Central European single low NDVI event *per se needs* previous summer precipitation to be below average and previous winter precipitation to be above average". That is, no inferences can be drawn from Fig. 5 about strictly necessary conditions for a low NDVI event. Rather, the analyses presented in Fig. 5 aim to identify *statistically significant precursors*. The concept of meteorological precursors is very well established in the meteorological literature and refers to a feature that occurs with a statistically significantly higher frequency in a certain set of meteorological histories (e.g., those preceding low NDVI events) compared to climatology (e.g., Martius et al., 2008). When attempting to improve the physical understanding of any meteorological phenomenon such precursors are thus interesting features, because they stand out of the noise that invariably exists in any set of meteorological histories preceding certain events. They inform about *shared* characteristics of these histories that possibly (but not necessarily) are physically related to the events of interest. However, such precursors need to be identified first using statistical methods, which is what Fig. 5 presents. In that sense, Fig. 5 simply indicates that the features mentioned by the reviewer occurred at a higher magnitude in the meteorological histories of our events compared to what could be expected under climatological conditions.

To make this explicitly clear we revised the manuscript as follows:
1. Stick to the term "meteorological precursor", which is explicitly defined at the very beginning of the manuscript (L. 87).
2. Extensive clarification in the discussion section on what we identify with a meteorological precursor and stating that we do not intend to identify "a universally valid meteorological history leading to low NDVI events" (Sect. 4.2, first paragraph).
3. New section (Sect. 3.2) presenting individual meteorological histories, i.e., case studies of single low NDVI events, in three different regions and years. They are clearly separated from the systematic

assessment of meteorological histories, which serve a different purpose than the concern raised by the reviewer. Furthermore, they introduce the concept of a meteorological history.
4. Sect. 3.3 is introduced with a paragraph that underlines that we do *not* aim at finding a "general recipe" for low NDVI events. Rather, we explain what our systematic analysis is able to identify (see above).

**8.** It is at least very counterintuitive why an overshoot precipitation in the previous winter should be an important 'ingredient' for a drought event, since it in fact would rather replenish soil-water resources. In contrast, a dry winter prior to an event seems more meaningful as described for the Mediterranean.

**Authors:** We regret that the discussion on the wet winter signal was triggered by over-proportional contribution of the year 2018. As the wet winter is no meteorological precursor to low NDVI events as defined in our revised manuscript (MR 2b), we agree that it is not a shared characteristic of the meteorological histories in the temperate biome. However, as mentioned in C6 & C7, we did not intend to create false impressions but rather documented what the data revealed.

**9.** A reason for these partly counterintuitive results is probably the limited number of events underlying these analyses. As pointed out by the authors, 42% of single events refer to the 2018 drought, indicating that this specific event (with a wet winter in advance) has a strong fingerprint on the storylines. Even stronger is the effect of the 2018/2019 consecutive event which renders 82% of consecutive events. Consequently, the meteorological conditions prior to the 2018/2019 event dominate the meteorological storyline for consecutive events, but I am convinced that also other constellations can lead to consecutive extraordinary NDVI.

**Authors:** The reviewer raises a valid concern here and in the following paragraphs. First, however, we also think there was a misunderstanding, as the numbers above were not cited correctly. We point out that 42% of the event grid cells referred to the low NDVI grid cells in 2018, 2019, and 2020 – not only 2018. Furthermore, 82% of the consecutive events refer to *two* two-year event sequences, namely 2018/19 and 2019/2020 – not only 2018/19. This is why we – in the first place – had no reason to believe that there was an unreasonably strong contribution of individual years, given that we investigate rare events.

Nevertheless, we fully agree that individual years with an extensive low NDVI event can contribute very strongly to the overall results. The sub-sampling approach introduced in Sect. 3.3 (L. 268–279) heavily attenuates the weight of years with many low NDVI grid cells relative to those with only few. As a result of the reduced dependence on individual years, also the sensitivity to the threshold parameters of the event identification was reduced (see Appendix A). However, the meteorological history of a more robust set of events proves to be very robust to such variations (Supplementary Fig. A1), which is mentioned already in L. 150–157.

**10.** Given this dominance of the 2018/2019 event on these analyses, I was surprised to see precipitation to be above average in the winter 2018/2019 since in fact this winter was drier than usual in Central Europe, explaining why the drought actually simply went on in 2019 since soil water was not replenished – as usually – in winter. Therefore, I would be very careful with talking about legacy effects but rather interpret the drought 2019 as an ongoing drought. This can for instance be seen when studying soil water deficit indices or other soil-related drought metrics over the winter 2018/2019. Again, this makes me wonder whether the used climate parameter precipitation is the right candidate for quantifying drought impact. I have to stress, that the authors do discuss the disadvantage of having only a short period for their overall analyses in section

4.5, resulting in the inability to 'perform superior statistical modelling or resolve species specific responses' (the coarse resolution of 0.05° would anyhow not allow for the latter). Unfortunately, they do not mention the potential effect of single events dominating their meteorological storylines even though the numbers are given in section 3.2.3. To account for the dominance-effect, it would be interesting to see the storyline analyses based on a weighed random subsampling of events to avoid dominance of single events in the storylines.

**Authors:** Thank you for this suggestion. We used a sub-sampling method that has the same goal as the suggestion by the reviewer (see MR 2, C9, L. 268ff).

**11.** But all in all, I have doubts that an average storyline– i.e. a 'recipe' – for extreme NDVI exists beyond concurrent spring/summer conditions. The droughts of the past two decades all had in common that summers were extraordinarily warm and dry. However, the timing of drought differed and so did the conditions in the previous winter (warm-wet vs. cool-wet vs. warm-dry) as well as the previous summer. Thus, even if 100-year lasting NDVI time-series were available, I would be surprised if we were able to find a specific storyline resulting in single or consecutive drought events.

**Authors:** We think we offer a careful analysis that is able to test how far back in time meteorological anomalies prior to low NDVI events stood out from the noise. Again, we thereby do not speculate about the "generic, globally valid storyline for low NDVI events" but rather we evaluate the available data for Europe in the last two decades and with – we think – sophisticated resampling, we can show that, with a relatively high degree of confidence, the signals we find are not just noise. With the new sub-sampling applied (MR 2), we strengthen the robustness of our results. See C7 for where this is made clear in the revised manuscript. Most importantly, we discuss for the reader in L. 423–430 that our results are statistically significant but, of course, we are not able to infer from meteorological data only whether there is a mechanistic link to low NDVI. However, we have robust evidence that what we observe prior to low NDVI events is not just noise, but a systematic deviation of the meteorological conditions. In the case of the temperate biome, the identified meteorological precursors reach further back than one year. We would, therefore, be highly curious to learn more from any discussions in the forest dynamics community that was triggered by the meteorological precursors presented in this study.

**12.** Finally, I miss the mention of potentially additional noise on the analyses. For instance, in 2019 a late-frost event stroke parts of Central Europe which potentially affected NDVI in summer 2019. Same holds true for the winter-storm early in 2018. This also refers to my mention on the uniqueness of meteorological storylines preceding extraordinary NDVI values. While I agree that the main cause of shown events is spring/summer drought, the possible additional effects of late-frost/storm impacts and others – which is briefly mentioned at the initiation of the discussion – is not well elaborated, at least not in context of the meteorological storylines.

**Authors:** Summer NDVI is definitely also affected by other disturbances, as highlighted, e.g., in Buras et al. (2021). This is why we think the way we assess significance is such an appropriate tool to *only detect shared meteorological signals (i.e., meteorological precursors)* of many low NDVI grid cells. The more other factors were driving low NDVI events, the less significant T2m and P signals would result from our analysis in Figs. 5 & 6. Our study, however, clearly highlights that there are some shared meteorological precursors. We were therefore much more careful with our wording, as e.g., when phrasing the research questions (L. 90) and when discussing the meteorological histories (Sect. 4.2). We do address other disturbances in the result (L. 235) and discussion section (L. 407–410). We also state that our analyses include indirect effects of

meteorology on low NDVI events that are delayed in time (e.g., structural overshoot) or happen indirectly (wildfire), e.g., in L. 405ff, L. 434ff, L. 444ff, L. 450, or L. 463.

On the other hand, we also explain why we think that due to the large-scale nature of the low NDVI events we identify, some disturbances do not appear in our event set (L. 411–415). The caveats of our large-scale approach are then again extensively discussed in an entire section (Sect. 4.4, highlighting, e.g., the multi-dimensional nature of tree mortality (L. 497) or influences of micro climate or forest management (L. 500). However, we really want to stress that only meteorological precursors are highlighted and discussed, and we do not state that other disturbances were *not* of importance to low NDVI events. Also, the research questions only aim to identify signals in T2m, P, and weather system frequencies (see L.90–92).

I regret, that I cannot be more positive. As mentioned earlier in this review, I appreciate that the authors have refined their analyses in comparison to the last time I reviewed this manuscript: compared to the previous version of the paper, it now reads much clearer and the analyses as well as their presentation have certainly improved. But given the major revision they undertook, I wonder what made the authors decide against a higher spatial resolution of MODIS data, which is freely accessible and also why they stick to the standardization of non-normally distributed data. Both issues could have been dealt with in course of the revision without extreme efforts and would allow for a more precise picture of forest drought meteorological storylines. At current, the selection of events is blurred by effects from non-forest land-cover and potentially inadequate standardization resulting in different absolute variations being treated equally (for instance if the standard deviation differs between pixels which can easily happen given the varying contribution of non-forest land-cover to the pixels but also when comparing different forest types).

**13.** As a way forward, I (again) recommend to utilize high-resolution MODIS products and mask non-forest areas from the analyses. Inappropriate standardization of data should be avoided (suggestions are given in point 2 above). Regarding the meteorological storylines, I recommend to randomly sub-select the data to avoid single events – such as 2018 or 2018/2019 – dominate the storylines.

This may result in a lower overall sample size but that could be accounted for by bootstrapping the random subsamples over several iterations (just make sure this does not result in a stronger contribution of single events, i.e. keep the weighing for each iteration constant).

**Authors:** Thank you for these suggestions. It was indeed very useful to redo the meteorological histories (Fig. 5) with a sub-sampling approach as discussed previously (see MR 2).

**14.** When interpreting corresponding results, I would highly appreciate if the authors communicate their findings in a less straightforward way, i.e. avoid selling a 'recipe for droughts' but rather tone down to something like: meteorological conditions that have preceded extreme forest drought impacts over the last two decades and recall that under anticipated climate change 'storylines' may change if drought frequencies increase.

**Authors:** Thanks, as noted in C6, C7, and C11, we carefully reconsidered our wording to avoid the impression of "selling a recipe".

**15.** I hope my comments help the authors refining their analyses towards a more robust interpretation. At this stage, I refrain from commenting on textual aspects of the manuscript since I believe that the revisions change a larger part of the methods, results, and discussion.

**Authors:** The comments of the reviewer were of great help to revise the presented manuscript. We invite the reviewer to comment on textual aspects in the upcoming round of revisions. Note that we have included textual aspects and minor comments of Reviewer 2 already in the current version of the manuscript.

**Reviewer 2**

**16.** Herrmann et al. analyse storylines of low summer NDVI events in Europe for the period 2000- 2020. The storylines are estimated for up to two years prior to the low NDVI events and are analysed separately for the temperate and Mediterranean biomes. This is an extremely relevant topic, since the ongoing disturbances in Europe are in many regions unprecedented (Senf & Seidl, 2021) and the drivers of tree mortality are still elusive (Hartmann et al., 2022). The use of storylines to evaluate drivers of low NDVI events is an innovative and well-fitting approach that could have a high potential to shed light into the general processes driving these events.

**Authors:** Thank you for this positive overall assessment of the objectives of our study. Note that we now use the term "meteorological history" instead of "meteorological storyline" to more clearly refer to our intention of assessing the characteristics of the meteorological past of low NDVI grid cells (see C34 for a detailed introduction of this term).

After reading the manuscript in detail, I believe the study unfortunately falls short of this ambition for generality, due to a number of fundamental issues with the specific methodological approach, underlying data, and the general lack of mechanistic insights, as discussed in more detail below. Moreover, the reading is at times engaging but at times it can be quite cumbersome due to the (heavy) use of confusing notation, unclear grammar and some distracting sentences that read as if the authors were addressing specific reviewers' comments, perhaps from a previous submission. That said, I believe that the analysis performed here can be of great value to the community, should these issues be addressed.

Major comments

**17.** 1) Generality of the storylines & mechanistic interpretation. At the moment, the results and discussion are presented as if the authors can derive general lessons about meteorological storylines leading to reduced forest greenness in summer.

**Authors:** This comment was also raised by Reviewer 1. Please refer to C6 & C7 & C11 above. In short, we apologize for creating a false impression, as we just let the data tell their story. We do not want to claim any generality for earlier or future events, or for events in other parts of the world. The referenced replies state how we aim to communicate this more clearly in the current version of the manuscript.

**18.** Given the short length of the time record (3 years), and that very few (1-3) large-scale events dominate the overall number of events detected – 42% of individual events correspond to the 2018 drought/heat event – it is hard to accept that the conclusions can be generalized to different event types. This is even more problematic for the "consecutive events", which refer basically to the 2018/19 event (making up 82% of the "individual events").

**Authors:** Unfortunately, there was a misunderstanding here, the above reference of our numbers is not correct (see C9). C9 is stating among others that the meteorological histories are now based on a sub-sample of all events, which reduces the weight of individual events (see L. 268–279). Given the sub-sampling approach and that we do not investigate event sequences anymore (MR 2a), confusion of which signal we investigate should be avoided completely in the current manuscript.

**19.** Therefore, the event samples cannot be treated as individual events. If I understood well (see my comment below) the authors calculate the anomalies for the climate variables by selecting 10k randomly

selected samples and calculate their means over the different events for each of the variables. By using a random sampling of *n* "pixel" events that are dominated by a single large-scale event (or two for the repeated events), a large fraction of the 10k samples will stem from this single event, so that their sampling will necessarily be biased.

**Authors:** We think there is a misunderstanding here. First, we do not only sample the low NDVI grid cells during the year of the event in the bootstrapping. We keep the annual chunks of low NDVI grid cells, i.e., the annual event masks (1=event, 0=no event), to extract meteorological anomalies of a randomly selected year in 2002–2022 where mask=1. So, the spatial coherence of meteorological anomalies in a specific year is retained over an area defined by low NDVI grid cells in a different year. Therefore, the anomalies in that region are sampled in a year that is not necessarily (and most likely is not) followed by a low NDVI event due to the random shuffling. All in all, our method allows for spatially coherent patterns in the meteorological fields of the randomly sampled years.

Second, even if it was the case that we sample grid cells mostly from a single large-scale event, then we would not find *statistically significant* anomalies, because then the random sampling would identify the same "large-scale signal". Figure 5 clearly looks different than expected in this case.

To be very clear about the bootstrapping approach, we greatly extended Appendix B including a new figure on how our approach works specifically.

**20.** It follows that the meteorological anomalies estimated for these storylines will strongly reflect the anomalies of those 1-2 synoptic-scale events. This results in the authors possibly overinterpreting short-term departures of the storylines from the 95% confidence interval, particularly the rainy previous winter season – that occurred in much of the region affected by the 2018 event (https://climate.copernicus.eu/european-wet-and-dry-conditions) – as a general percussor of low summer NDVI events. It is unclear what would be the mechanistic link between rainy winters and dry/hot summers leading to low NDVI events and the authors do not offer satisfactory explanations (rather list a number of hypothetical processes in the discussion).

**Authors:** The new subsampling of low NDVI grid cells promoted by both reviewers results in a meteorological history without the counterintuitive wet winter signal (see MR 2). However, this is a result of more carefully assessing the meteorological history that is tested in the bootstrapping, and not a shortcoming of the bootstrapping test itself. Because generally, we are confident that departures from the confidence interval are significant because the resampling is done in a way that we use the same area affected in certain years also in the bootstrapping reference. Therefore, a large-scale anomaly can affect also the resampled histories, in a similar fashion as is the case in the observed event history (see C19 and C25). Finally, even if one year contributes strongly to the meteorological histories, they need not be the same everywhere within the event.

We also agree that we could not offer an explanation for the role of the mild and wet winters and rather referred to existing literature (see C8). But this is a very common feature in climate science that some studies identify signals in observations and/or simulations (e.g., many important trend studies), and these studies are valuable although they hardly ever can "explain" the reasons for the identified signals. So, in our view, there is nothing bad with presenting our results to the forest and climate community as an invitation to either falsify or explain our results. We further elaborate on this in similar comments of Reviewer 1 in C6-C9.

**21.** I would be curious to whether the authors could find significant signals, especially the rainy winter peak before the low NDVI events in temperate regions, if events occurring in 2018 are removed from the analysis.

**Authors:** We like this suggestion, which is similar to a suggestion from Reviewer 1 and we now used a random sampling of similar sample sizes per year (see MR 2). We indeed still find significant signals that were apparently not only occurring prior to the 2018 event.

The authors could also try an alternative sampling approach that would explicitly correct for the sampling bias due to the predominance of few large-scale synoptic events, but this would reduce drastically the number of events (by about half for EV10 and by 80% for EV11).

**22.** An alternative could be to tone down the ambition to derive general conclusions for 2000-2020 and rather focus on highlighting the value of the storyline approach to learn about anomalies in vegetation activity and, specifically, extremes. While the results for 2018/19 in central Europe are not necessarily new, the fact that the approach proposed here shows consistent results past studies while providing a means to assess more generally past history is a strength that could/should be stressed. For the Mediterranean case, the results might also reflect the year 2017 the most (given figure 3d), so that a similar approach/reasoning can be applied.

**Authors:** Thank you. In addition to using a sub-sampling approach, we also rephrased our conclusions more carefully to reflect this valuable input. Furthermore, the term "meteorological precursor" was introduced to make clear that we talk about "features in the meteorological histories that occur at a statistically significantly higher rate preceding reduced forest greenness events than in climatology, and that are shared among many events" (L. 87). We appreciate that the reviewer summarizes the novelty of this study in such an appropriate way.

**23.** 2) Spatial resolution. The authors analyse storylines leading to low NDVI extremes, focusing on forest ecosystems. To do this, they rely on NDVI data from MODIS at 0.05km. This is already a relatively coarse resolution for the analysis proposed, as pointed out by Reviewer 1, but the authors then further coarsen the data in the analysis to 0.5 degree spatial resolution. The authors tried to minimize this issue by estimating low NDVI values first at the 5km resolution and then imposing the condition that at least 50% of the coarser grid cell needs to register low NDVI events. However, their definition of "forest pixels" includes pixels with less than 50% of forest, and even as low as only 20% (Section 2.1). This results in many areas that are dominated by croplands or mixed tree/herbaceous cover (parts of central and eastern Europe and of Italy, southern Sweden), being included as "forests".

**Authors:** It is worth mentioning first that we now use 250 m resolution NDVI data (MR 1, Sect. 2.2). Furthermore, we use a new land cover data set to categorically identify forest areas (MR 4a, Sect. 2.1). Consequently, we can much more reliably identify 250 m *forest* pixels and 0.5° *forest* grid cells where forest was affected by low NDVI, and, thus, these concerns no longer apply to the new methodology.

Nevertheless, for clarification, we think that there was a misunderstanding in the above-cited numbers. We only considered 0.05°x0.05° pixels that are at least 50% covered by forests. The threshold of 20% forest area applied to the 0.5° grid cells considered. Only 0.5° grid cells with more than 20% forest area are considered at all. The eventual NDVI signal considered within these grid cells includes forest pixels only, i.e., they each must have at least 50% forest coverage.

**24.** While I understand that the authors could not perform this analysis at the higher MODIS spatial resolution given the lack of such high-resolution climate data over Europe, it is unclear why the authors chose to aggregate results to 0.5 degrees, since ERA5's native resolution is 0.25 degree, and ERA5 land provides

downscaled fields at 0.1 degree spatial resolution, much closer to the spatial scale used to define events and to distinguish (at least better) forest from non-forest pixels.

**Authors:** The official ECMWF webpage states that "The data [ERA5] cover the Earth on a 30 km grid". The grid spacing is not the same as effective resolution, which is typically about four times the grid spacing (since four grid points are required to resolve a wavelength). In principle one can download ERA5 data on any grid, e.g., a 0.25° instead of a 0.5° grid, but this does not increase the information content of the data, it rather oversamples the data. This is also why we only found small differences in a comparison of seasonal mean P and T2m fields in ERA5 data at 0.25° vs. at 0.5°. Since ERA5-Land is based on the same spectral truncation of the ECMWF model, it is not clear whether the 0.1° grid provides meaningful additional information. Note that ERA5-Land does not include any correction of the precipitation by the ERA5 reanalysis (Muñoz-Sabater et al., 2021).

We therefore still use ERA5 at 0.5° resolution, which is a meaningful target resolution of our analyses of large-scale meteorological conditions and synoptic-scale weather systems. Even if there were local meteorological signals important to the low NDVI events that are not present at the 0.5° scale (which is most likely the case, e.g., Föhn winds), we would not identify them in our study as meteorological precursors (see also C19). This is because we only identify precursors that are statistically significantly different from climatological conditions at 0.5°. Therefore, it is generally not problematic to assess large-scale meteorology at any scale, as long as it is meaningful to synoptic meteorology.

The same for E-OBS, and the group of COSMO-REA reanalyzes provide temperature and precipitation fields at 6km or even 2km. Surely, repeating the analysis at such fine resolutions (2km) would be more computationally expensive, so that keeping the 5km scale of the initial steps is probably the most feasible option.

**25a.** 3) Synthetic event sets and meteorological means. In appendix A and the methods, it is generally unclear if the 10k event sets are derived per pixel individually, as suggested by keeping $n$ in subscript (appendix A), or across pixels or across individual events (i.e. pairs of (pixel, year)). It is also unclear how the authors calculate the means and climatologies of the meteorological variables (Line 562-63). Generally, this makes it difficult to evaluate and reproduce the analysis.

**Authors:** Thank you for mentioning the difficulties in understanding the details of our method. We greatly improved Appendix B (former Appendix A), where we introduced a more detailed terminology, to avoid such confusion for further readers. We, nevertheless, would like to clarify the raised questions by the reviewer below.

**25b.** The authors state that they take 10k synthetic event sets and then (1) take the mean among all events, then (2) take a distribution of these mean values that is used to calculate climatologies. I have several questions about this step:

- It is not clear what is the size of the event set matrix, is it 3d (10k, 21, n)?
  - **Authors:** The reference event matrix is a 3D $s \times g \times n_{ev}$ matrix, where $s = 10'000$ (now 1'000) corresponds to the number of random samples, $g = 3$ corresponds to the grid point identifiers year, latitude and longitude, and $n_{ev}$ corresponds to the total number of low NDVI grid cells (which now is 1'260 and 544 for the temperate and Mediterranean biome, respectively).
- And over which dimension is the first mean calculated, the second?

- o **Authors:** For every sample we extract meteorological fields in the respective year, lat, lon combination indicated by the matrix above. The resulting matrix, e.g., for temperature, is 3D again with size $s \times n_{ev} \times n_{dt}$. It includes $T2m'_{90d}$ for $s = 10'000$ synthetic samples, including $n_{ev}$ events, over $n_{dt} = 1'095$ timesteps (three years). The first mean is taken along the second axis (event mean, including lat-weighing). The resulting $s \times n_{dt}$ matrix is used to calculate the 2.5th and 97.5th percentile along its first axis. The outcome is a time-evolving confidence interval with $n_{dt}$ entries (see the more elaborate description below).
- What is the size of the resulting vector/matrix?
  - o **Authors:** See above
- Do the climatologies in T and P have seasonality?
  - o **Authors:** Yes they do: For each calendar day, the climatology $\overline{P_{90d}}$ is equal to the mean over the 21 annual $P_{90d}$ values at that calendar day. $P_{90d}$ itself is the mean over the past 90 days, exerting seasonality as it is a moving average. Consequently, also $\overline{P_{90d}}$ is changing for every calendar day (see also Sect. 2.4.1).
- Over which dimension is the standardization performed?
  - o **Authors:** The standardization is performed as initial data treatment prior to the bootstrapping. At every grid point there are 21 $P_{90d}$ values per calendar day which are used to calculate mean and standard deviation for local standardization. As for the climatologies above, the period used is Sep 2002 – Aug 2022.

For the consecutive event sets, how exactly is this done, do you select from EV10 subsets of synthetic events that happen consecutively? Where are the values of the different n values derived from? Finally, the authors mention in the methods section that the sampling strategy preserves the spatial correlation of T and P fields, but this is not discussed in Appendix A.

**25c.** As a consequence of these issues, it is difficult to evaluate the robustness of the method described. I suggest either describing the steps using explicitly mathematical notation, reporting the sizes of the vectors/matrices, and/or a flowchart to facilitate understanding of the exact steps performed here. It would also be good to discuss the rationale behind using bootstrapping in the methods and start of Appendix A.

**Authors:** This reviewer comment clearly indicates that our explanation of the method and statistical approach was insufficient. We have rewritten Sects. 2.4.1, 2.4.2, and Appendix B (former Appendix A).

**26.** 4) Use of forest disturbance dataset. It is unclear why the authors introduce a whole new dataset that is only used for cross-comparison purposes. Low NDVI events do not necessarily have to correspond to crown-mortality events (D), and conversely crown mortality events might not necessarily result in low NDVI if understory vegetation benefits from the canopy opening. This discrepancy is briefly mentioned by the authors, and very clear in Figure 4, but not fully addressed. Another point not really mentioned is that D can be affected strongly by anthropogenic signals, such as management and selecting logging. Since the authors presented the goals of the analysis as to evaluate extremes in vegetation greenness, a clearly well-defined and constrained problem, I find that the comparison with D adds more confusion, rather than clarity to the study.

**Authors:** We think that this brief comparison is useful, because it supports the notion that observed forest disturbances are indeed much more likely to occur in grid cells where we identify low NDVI events. However, we agree that disturbance data (canopy mortality) and low NDVI do not correspond in every case, due to

reasons mentioned by the reviewer. To accommodate the reviewers' comment, we shifted this comparison to Appendix D as it is indeed not essential for constructing the meteorological histories.

Specific comments:

**27.** "forest performance" is not a standard expression and is rather unclear whether the authors mean vitality, health, or any other aspect reflected by NDVI (vegetation cover, LAI, ...). To be accurate, the best would be to stick to "forest greenness".

**Authors:** Thank you, this is a good suggestion, and we will stick to forest greenness.

**28.** Line 23: Over what time-scales is this sentence referring to? This is not true for the past several decades – acid rain, changes in management, reforestation, elevated CO2, nutrient deposition, ... there is a long list of processes that have been destabilizing forests in Europe.

**Authors:** We clarified the sentence, as we were mostly referring to the long-term acclimatization and adaptation to climatic variables. It now reads "European forest ecosystems have typically been in balance with their climatic environment and are, thus, largely adapted and acclimated to meteorological variability on a larger scale."

**29.** Line 48-50: but in Mediterranean regions, drought can also reduce fuel load (Pausas and Ribeiro, 2013).

**Authors:** Yes, indeed. We added the sentence in L. 51: "In strongly fuel-limited regions, however, forest fires can in the long run negatively feed back on fire activity (Pausas & Ribeiro, 2013)."

**30.** Line 56: intensively discussed... in the literature?

**Authors:** Yes, it now reads "…is intensively discussed in the literature" (L. 59).

**31.** Line 59: "stressed", or "identified"?

**Authors:** "stressed" was changed to "identified" (L. 61)

**32.** Line 60: "drought prone region", such as the Mediterranean?

**Authors:** We use "drought prone region" to characterize a climatic characteristic of this region. As the study addresses the southwestern US, we would not like to confuse it by mentioning the Mediterranean.

**33.** Line 63: "margins" of the growing season is an unusual expression

**Authors:** We now use the term "bounds" instead (L. 66).

**34.** Line 65: increasing understanding "of ...", specify what is meant here. Furthermore, the concept of storylines should be described in more detail and appropriately referenced (e.g. Shepherd et al. 2018)

**Authors:** We realize our use of "storyline" can be confused with the established storyline concept of Shepherd et al. (2018). Even though we use an (extreme) event-oriented approach, we do not intend to address the plausibility and the causality of the meteorological storylines we find, nor to entangle *all* the drivers of low NDVI events. Therefore, we no longer use the terminology "meteorological storyline" but

rather "meteorological history", which better focuses on the past evolution and the occurrence of statistically significant precursors (of T2m and P).

**35.** Line 73-74: Other studies have attributed this to a Wave-7 pattern and a positive NAO phase (Drouard et al., 2019 and Kornhuber et al., 2019)

**Authors:** We introduce and later on use the perspective of weather systems that occur at the synoptic scale, like cyclones and anticyclones or blocks. Also weak synoptic forcing, i.e., when none of the aforementioned weather systems is present, can lead to summer heatwaves – as described.

**36.** Line 84: what do the 90 in subscript stand for, 90-day moving average not yet mentioned, making it confusing.

**Authors:** We now write "We focus on the evolution of 90-day average 2-m temperature (…) and precipitation (…)…" (L.89).

**37.** Line 85: why 3 year only?

**Authors:** This choice is guided by two opposing constraints: (1) Going back longer potentially shows more significant results in Figs. 5 & 6 simply by design (at 5% of the time steps). However, (2) the length of the data record becomes more critical the longer we go back. The chance that mechanistically unlinked events appear in one meteorological history increases with the length of the history. Moreover, our results in Figs. 5 indicate that more than 3 years in the past the meteorological signals are not distinguishable from noise anymore.

**38.** Lines 87-88: I propose swapping (2) and (3)

**Authors:** The research questions have changed significantly.

**39.** Lines 101-102: why the choice of this specific domain and why ignoring boreal forests?

**Authors:** The domain relates to the CLC land cover data set used, which is a European product (Sect. 2.1). Further, we prefer to only consider the temperate and Mediterranean biome in this study on European forests, which are the two largest European biomes.

**40.** Figure 1 caption: please add a brief description of panel b) and it is impossible for the reader to understand it without reading the methods.

**Authors:** We now include a more elaborate description in the caption of Fig. 1b.

**41.** Line 106: no justification about why 0.5 degree is used.

**Authors:** We elaborate on the use of ERA5 at 0.5° resolution in more detail in C23. In essence, it is a meaningful scale for synoptic analyses and prevents over-sampling the ERA5 model.

**42.** Line 116: "at forest pixels" does not seem grammatically correct

**Authors:** This sentence has been deleted.

**43.** Line 119: this is the first time missing values are referred to. Where do they stem from? The use of quality control flags? And what if there are two consecutive months missing?

**Authors:** We better elaborate on gap filling in L. 118ff as follows: "The Application for Extracting and Exploring Analysis Ready Samples (AppEEARS) additionally provides MODIS pixel quality. We mask NDVI values that are of poor quality due to snow and clouds, and only retain NDVI values with good and marginal quality according to MODIS pixel quality. The resulting NDVI time series contain missing values, which we linearly interpolate from neighbouring time steps as in Buras et al. (2021)." If values at two time steps t1 and t2 are missing, they are linearly interpolated from the values at neighbouring time steps t0 and t3.

**44.** Line 123: in mathematical notation, the apostrophe is usually used to express the first derivative, so this notation is confusing. Why not a for anomaly?

**Authors:** In our own literature we typically refer to anomalies with an apostrophe and would like to keep it that way.

**45.** Line 132: replace "at" by "for" or "in"

**Authors:** The sentence has been deleted.

**46.** Line 134: there is no scheme presented here. Do the authors mean the "approach presented here"?

**Authors:** We now use "approach" instead of "scheme" (L. 131).

**47.** Line 135: is it a "forest grid cell" or an "atmospheric grid cell over forested pixels"? Overall, I find these definitions confusing.

**Authors:** All fields used are in 2D in the horizontal. Therefore, we always consider the same grid cells, which have different properties such as 2m-temperature or forest cover. Based on that we define the term "forest grid cell" in Sect. 2.1 (forest cover larger equal than 10%).

**48.** Line 136 and Equation 2: how does the flag work when there at 3 months in the season? Since the authors take the minimum value over the season, does this mean that it is enough that 1 month in JJA is flagged as low NDVI? How can the authors be confident that this is a "low NDVI season"?

**Authors:** This sentence has changed due to the new event definition.

**49.** Line 148: correct to "reanalysis", singular

**Authors:** We now use "ERA5 reanalysis data".

**50.** Line 149: why interpolating ERA5 to 0.5 degree?

**Authors:** We elaborate on the use of ERA5 at 0.5° resolution in more detail in C23. In essence, it is a meaningful scale for synoptic analyses and prevents over-sampling the ERA5 model.

**51.** Line 151: if this refers to seasonal averages, should the subscript be 90d or "season"?

**Authors:** We no longer use the term seasonal average for e.g., $P'_{90d}$, but rather 90-day average.

**52.** Line 151-159: it is not fully clear if the standardization is done also for the 90d moving windows, please clarify.

**Authors:** The standardization is now described in L. 170ff as follows: "For all four variables, we calculate 90-day mean values as a right-aligned moving average. Each 90-day mean value, therefore, is labelled by the time step of the last value that contributes to the average. Leap days are discarded from the analysis. The climatologies of the four variables cover 90-day averages from 1 September 2001 to 31 August 2022. Based on these 90-day mean values, we compute normalized anomalies at every forest grid cell for variables [...] as follows:". The standardization is done for 90-day averages, which is also shown by Eq. 5.

**53.** Line 167: so only 40% of the values are "not extreme"?
Line 167-169: give correlation values and respective significance
Figure 2: what do the colorbars indicate? Not mentioned in the caption.

**Authors:** The three comments refer to a section that we have deleted in the revised version. We no longer use the bivariate representation of the meteorological histories as they highlight similar aspects of the meteorological history as previous sections.

**54.** Line 181: this is not described in the Appendix A.

**Authors:** Appendix B (former Appendix A) has been completely reformulated, and now includes the sentence: "Note that the shuffling of years is done prior to extracting the spatial fields of X from the ERA5 data set. This has – in contrast to a random sampling of all forest grid cells – the convenient effect that spatial correlation in these meteorological variables is retained." (L. 637ff).

**55.** Line 191: for a Biogeosciences audience, is would be good to explain what the "outermost closed SLP contour ..." means.
Line 191-195: more generally, for a Biogeosciences audience it would be good to explain what additional information does this analysis bring.

**Authors:** Thank you very much for the two comments above. These hints are very helpful for our goal to establish a bridge between two different research fields. We now elaborate more clearly on the use and identification of weather systems in L. 164-170.

**56.** Line 197: grammar "To evaluate", "for" + "ing" does not express purpose/intention.

**Authors:** This sentence has been deleted.

**57.** Line 201: which is aggregated and normalized. Why is the normalization now done at the coarser resolution?

**Authors:** This section has been shifted to Appendix D. The reason for normalizing after the aggregation is that all our analyses are done at the 0.5° scale. We want to understand the link of synoptic meteorology and large-scale low NDVI event at that scale, and therefore compare low NDVI events also to an aggregated measure of forest disturbances.

**58.** ine 216: correct "succeeding" to "subsequent"

**Authors:** This sentence has been deleted.

**59.** Section 3.1.1 – if the purpose of using the D dataset is for evaluation of low NDVI events, why not compare the annual variability in D as well here and in Figure 3?

**Authors:** The importance of this chapter has been reduced and the use of disturbance data now rather serves a comparison than an evaluation, as mentioned in L. 642ff.

**60.** Figure 3b: the colors in the 4 quads are hardly distinguishable

**Authors:** Thank you, this is true: we now display the intensity of the events as histograms in Fig. 2b.

**61.** Figure 3d: if the authors decide to keep D, then add the extent affected by D events in this panel as well.

**Authors:** We decided not to keep it in the main part of the manuscript.

**62.** Line 249: these "conceptual, technical and physical reasons" are not really thoroughly discussed in Sec 41.

**Authors:** The sentence has been removed.

**63.** Figure 4: What is D'[]?

**Authors:** D' is the anomaly of disturbance area as defined in Eq. D1 (Appendix D). The bracket has been removed. It was meant to indicate that it is a unitless quantity.

**64.** Figure 5: mark in shaded areas the periods when the event mean is outside of the 95% CI. What do the vertical lines in a-d indicate?

**Authors:** We have introduced coloured dots in all panels to highlight periods of significant anomalies. For keeping the caption at a reasonable length, we do not describe the grey vertical lines, as they are grid lines.

**65.** Line 269: negative, but still within the 95% CI. Here, and elsewhere, the authors over-emphasize non-significant results.

**Authors:** We have almost completely reduced the mention of non-significant results and now mostly discuss results outside the 95% CI, which are termed "meteorological precursors".

**66.** Line 271: for short periods. Add "is significant for Xdays ... "

**Authors:** We decided not to indicate the number of days, as every value represents a 90-day average. The mention of a period of n days could cause confusion.

**67.** Line 274: continuously negative, but still within the 95% CI. Please give duration of the periods when event mean is outside of the 95% CI.

**Authors:** We now focus almost exclusively on significant anomalies in our descriptions.

**68.** Line 276: negative in the previous winter, but not extreme. Line 280: DJF of which year?

**Authors:** This sentence does no longer exist in the revised version of the manuscript.

**69.** Line 284: can you give a mechanistic explanation for this?

**Authors:** We discuss the relevance of the identified meteorological precursors in the discussion section (Sect. 4.2). From the meteorological anomalies alone, we cannot deduce any mechanistic explanations.

**70.** Line 304: add ", respectively," between "P'90d" and "from" Line 315: the accumulation of dry periods is not significant

**Authors:** We now include "respectively" at the suggested position.

**71.** Section 3.2.3: please state clearly that this applies basically to 2018/19 in temperate regions, and make a similar assessment for the Mediterranean biome.
Line 331: the information about the fraction of these years to the event samples needs to be given much earlier in the manuscript, and please add information for the Mediterranean biome too.
Line 334: why is 2020 excluded?
Line 340: "hot" anomalies?
Line 350: again, I find in-depth mechanistic interpretation of these patterns lacking in the discussion.
Line 355: "exerts" is not applicable here, since the NDVI events have no influence on T90.
Line 356: I quite like that the authors here give specific values of the anomalies discussed. This should be done throughout the whole results section.
Line 358: what does "small" mean? That the absolute value is close to zero?
Line 364: only T, see comment above for line 315.
Figure 8: please explain what the different color shades mean (95%CI, I believe)

**Authors:** All comments above apply to sections that have been deleted during the revision.

**72.** Discussion Section: it is surprising that the limitations related to the short temporal records and the dominance of single years in the events analysed are not discussed.

**Authors:** As highlighted in many previous replies, we now greatly highlight that our results only apply to the study period 2002–2022. Furthermore, the results now are based on the sub-sampling approach described in Sect. 3.3, and, thus, are not dominated by single years (see C9).

**73.** Lines 414-325: How does the analysis done here "help to characterize the nature of these events"? If this would be true, I would expect separate analyses for pixels with low NDVI and no crown mortality and pixels with low NDVI and crown mortality. Overall, this paragraph is quite distracting given that the main goal of the paper was to analyse low NDVI extremes.

**Authors:** This paragraph has been deleted as the event evaluation is not central to the manuscript anymore.

**74.** Line 432-433: the grammar can be improved

**Authors:** The sentence has been completely rephrased (see now L.445ff)

**75.** Line 437: regardless of what?

**Authors:** Sentence has been deleted.

**76.** Line 437-438: do the authors mean that drought early in the growing season directly "damages" forests, or simply that low P in spring promotes drier summers?

**Authors:** We refer to insights on which meteorological signals were identified as harmful for trees in one way or the other, where the referenced literature points to different mechanisms: e.g., early leaf senescence (Bigler & Vitasse, 2021) or canopy mortality (Senf et al., 2020). So rather the former than the latter.

**77.** Line 440: unclear why warming in the previous 3 years would affect an instantaneous process like fire.

**Authors:** Providing a lot of dry fuel can happen via the drying out of deadwood through evaporative water loss, which is a temperature-driven process. We refer to a longer time period that was unusual (now 25 months), but cannot pinpoint to the exact moment from which on such a warming might be important to fire, which might again affect NDVI over a longer time period.

**78.** Line 485: can you explain in more detail how acclimation results in reduced leaf area and productivity?
**Authors:** The sentence has been deleted.

**79.** Line 488-490: isn't this simply a consequence of the fact that only low NDVI events were selected? The authors did not evaluate post-event recovery trajectories separately, so that they cannot know whether increased vulnerability out competes acclimation.

**Authors:** This sentence has been deleted but exists in a similar form in L. 465ff: "Moreover, the succession of drought in consecutive summers is particularly harmful for temperate forests, while Mediterranean forests show a decreased sensitivity to the second drought (Anderegg et al., 2020)." We do not draw conclusions from our analyses on what outcompetes what anymore. However, we think it is a valid discussion point when mentioning inter-biome differences, that drought sensitivity in the Mediterranean is decreasing in contrast to temperate forests. This aligns well with a previous summer signal in temperate but not in Mediterranean forests.

Also note that we start Sect. 4.2 by stating the purpose of including literature *related to* our results, but not only *of what is shown* in our results: "Hereby it should be noted that this statistical analysis alone does not allow to infer causation between the precursors and the low NDVI events, it rather just identifies unusual co-occurrence of these precursors and the low NDVI events. The causation surmised in our interpretation of these precursors below is inferred from the large body of process-focused literature we cite."

**80.** Line 491: sensitivity to drought is not shown in the results, what do the authors mean here?
**Authors:** See above.

**81.** Line 501: what does "superior statistical modelling" mean?
**Authors:** The sentence has been deleted.

**82.** Line 508: "not shown", please add these results to the supplement and this is an important point.
**Authors:** The sentence has been deleted.

Conclusions

**83.** I find that the authors overemphasize the winter wet signal in the temperate biome, which first is rather short (a couple of days outside of the CI) and second cannot likely be generalized for all events. This should be toned down.

**Authors:** This is no longer part of the results and, thus, not concluded anymore.

Appendix A

**84.** Line 557: the superscript *r* should be placed above *n*, and be defined again in the text here, for those readers who might start here.

**Authors:** Formalism has changed, and superscripts used are now defined in the text.

**85.** Line 571: Add "for" before the "null hypothesis". Also, EV10 and EV11 have not been defined previously in the appendix, making reading confusing.

**Authors:** Sentence has been deleted.

References

Shepherd, T.G., Boyd, E., Calel, R.A. *et al.* Storylines: an alternative approach to representing uncertainty in physical aspects of climate change. *Climatic Change* **151,** 555–571 (2018). https://doi.org/10.1007/s10584-018-2317-9

Pausas, J.G. and Ribeiro, E. (2013), Fire and productivity. Global Ecology and Biogeography, 22: 728-736. https://doi.org/10.1111/geb.12043

Drouard, M., Kornhuber, K., & Woollings, T. (2019). Disentangling dynamic contributions to summer 2018 anomalous weather over Europe. Geophysical Research Letters, 46, 12537– 12546. https://doi.org/10.1029/2019GL084601

Kornhuber, K., Osprey, S., Coumou, D., Petri, S., Petoukhov, V., Rahmstorf, S., & Gray, L. (2019). Extreme weather events in early summer 2018 connected by a recurrent hemispheric wave-7 pattern. *Environmental Research Letters*, *14*(5), 054002.

**References:**

Büttner, G., Feranec, J., Jaffrain, G., Mari, L., Maucha, G., & Soukup, T. (2004). The CORINE land cover 2000 project. *EARSeL eProceedings*, *3*(3), 331-346. Available online at: http://eproceedings.uni-oldenburg.de/website/vol03%5F3/03%5F3%5Fbuttner2%2Ehtml

Buras, A., Rammig, A., & Zang, C. S. (2021). The European Forest Condition Monitor: Using Remotely Sensed Forest Greenness to Identify Hot Spots of Forest Decline. Frontiers in plant science, 12: 689220. https://doi.org/10.3389%2Ffpls.2021.689220

Didan, K. (2015). MOD13Q1 MODIS/Terra Vegetation Indices 16-Day L3 Global 250m SIN Grid V006. NASA EOSDIS Land Processes DAAC. https://doi.org/10.5067/MODIS/MOD13Q1.006

Dodge, Y. (2008). Central Limit Theorem. In: The Concise Encyclopedia of Statistics. Springer, New York, NY. https://doi.org/10.1007/978-0-387-32833-1_50

Martius, O., Schwierz, C., & Davies, H. C. (2008). Far-upstream precursors of heavy precipitation events on the Alpine south-side. Quarterly Journal of the Royal Meteorological Society, 134(631), 417–428. https://doi.org/10.1002/qj.229

Muñoz-Sabater, J., Dutra, E., Agustí-Panareda, A., Albergel, C., Arduini, G., Balsamo, G., Boussetta, S., Choulga, M., Harrigan, S., Hersbach, H., Martens, B., Miralles, D. G., Piles, M., Rodríguez-Fernández, N. J., Zsoter, E., Buontempo, C., and Thépaut, J.-N. (2021): ERA5-Land: a state-of-the-art global reanalysis dataset for land applications, Earth Syst. Sci. Data, 13, 4349–4383. https://doi.org/10.5194/essd-13-4349-2021

---

## Author Response (AR2)

egusphere-2022-425

**Reply document**

**Meteorological history of low forest greenness events in Europe in 2002-2022**

*Reply to minor revisions of both reviewers in February 2023 by Mauro Hermann, Matthias Röthlisberger, Arthur Gessler, Andreas Rigling, Cornelius Senf, Thomas Wohlgemuth, and Heini Wernli*

We acknowledge the reviewers again for the second round of constructive feedback regarding our manuscript. We are thankful that our revisions based on their comments could improve the manuscript to the point where only very minor revisions were necessary. We addressed most of them in the presented final version of the manuscript. Below you can find the final comments of both reviewers, which mainly addressed the formulation of the null hypothesis tested in our study, including our replies. Again, the comments of the reviewers are shown in black and our replies in blue. We number reviewer comments for referencing purposes throughout the document (comment 1 = C1, etc.).

Prior to that, we list a few minor changes made to the manuscript during the final read that go beyond typos:

- L. 103: We use CLC land cover from the year 2012 (in the center of the study period), not 2018 as stated in the previous version.

- Sect. 2.3 & Appendix A: We use an improved formulation for the count of negative 16-daily $NDVI'$ values in year $n$ (new $c_{n,ev}$ instead of $n_{t,ev}$), as well as its minimum threshold value (new $c_{ev}^{min}$ instead of $n_{t,ev}^{min}$). This is to avoid confusion with further abbreviations used (e.g., $n_{ev}$) and is more accurate regarding the use of the indices $n$ and $t$. The new terms are introduced in L. 137 and L. 142.

- L. 219: Removal of unit IQR, as the normalized NDVI anomalies are in fact unitless.

Finally, note that the low NDVI dataset will be uploaded under the doi https://doi.org/20.500.11850/505559 once the published study can be cited in the data manual. The doi will be reserved up to that point.

**Reviewer 1**

Altogether, the authors have done a good job in revising the manuscript and I highly appreciate the efforts they've taken to refine the paper. In my opinion, all the possible technical flaws from previous submissions have been overcome with the extensive revisions provided, wherefore I deem the manuscript publishable pending minor corrections. Of somewhat greater concern is however the formulation of the null-hypothesis of the bootstrapping test, which I don't agree with (see specifications below). Once these minor adjustments have been taken into consideration, the manuscript in my opinion deserves publication.

**Authors:** Many thanks for this very supportive feedback including one final suggestion for correction.

**1.** Lines 69-81: In this paragraph I was missing the context provided in Kornhuber et al., 2019: https://doi.org/10.1088%2F1748-9326%2Fab13bf which seems to be highly relevant in context of meteorological histories.

**Authors:** We have significant reservations regarding the theoretical underpinning of the arguments put forward in Kornhuber et al. (2019). These reservations include (but are not limited to) arguments made in Wirth & Polster (2021) and thus prefer not to refer to Kornhuber et al. (2019).

Wirth, V., & Polster, C. (2021). The Problem of Diagnosing Jet Waveguidability in the Presence of Large-Amplitude Eddies. *Journal of the Atmospheric Sciences*, *78*(10), 3137–3151. https://doi.org/10.1175/JAS-D-20-0292.1

**2.** Line 73: I guess you mean insolation not insulation

**Authors:** Yes of course, we changed insulation to insolation.

**3.** Section 2.2: Here I was wondering, whether the authors only considered forest-pixels (as per Corine land-cover) for further analyses. Reading further I got that it is mentioned in 2.3 but I think this is relevant information for this section, so that readers immediately become aware that only forest pixels were considered and thus noise from other land-cover types can be more or less ruled out (this you can even mention specifically to make your approach more robust). Also, I recommend providing a URL for the AppEEARS website.

**Authors:** This is a great suggestion. We included the following sentence at the beginning of Sect. 2.2 "As mentioned in Sect. 2.1, we only use NDVI at forest pixels according to CLC in order to minimize noise from other land cover types. The NDVI is based on…" (L. 113).

We include a link to the AppEEARS website in L. 119.

Furthermore, we again highlight the masking of non-forest land cover types in the second paragraph of this section in L. 120: "In addition to masking non-forest land cover, we mask NDVI values that are of poor quality…"

**4.** Line 171: Why are leap-days discarded? I don't really see a reason but I'm sure you had a good reason for doing so. Please elaborate.

**Authors:** We included the two reasons for our decision in the same sentence, now in L. 172 : "Leap days are discarded from the analysis to maintain consistency in each calendar day's climatology and the length of the meteorological histories."

For example, the use of only 5 days for calculating the climatologies on 29 February would result in standardized meteorological anomalies that are not readily comparable to those at other calendar days.

**5.** Line 192: I really like the bootstrapping approach but I disagree with the formulation of the null-hypothesis. In my opinion, your bootstrapping does not allow for assessing whether a meteorological history is related to the low NDVI event, since there is no direct link (like a correlation score or similar) incorporated in the test. To me, the H0 should be something like:

The meteorological history at tev – dt is not different from a randomly (arbitrarily) sampled meteorological history.

The way your H0 is formulated suggests a direct link (which might be the case, but this you do not test with the bootstrapping) whereas my suggestion emphasizes on a potential difference of meteorological histories between events and 'normal' periods. Please elaborate this throughout the manuscript and the appendices.

**Authors:** We want to highlight that no matter the formulation of H0, failure to reject an H0 in a statistical test does not imply that the H0 is true. Nevertheless, the reviewer raises an important point, which we would like to accommodate in our manuscript. We, therefore, reformulated H0 in L. 194 as follows:

"The meteorological history at t_ev - dt is equal to a randomly sampled meteorological history"

This change was also incorporated in Appendix B.

**6.** Fig. 2: it is quite difficult to digest panel a. I think it is okay like this, but for deeply interested readers it might be a good idea to have a multi-panel figure in the Appendix depicting the events for each year in a single map.

**Authors:** We agree on this point which is why we had Appendix D included (exactly the plot suggested by the reviewer). We now refer to this Appendix earlier in Sect. 3.1 (L. 208) and additionally in the caption of Fig. 2, which is why it now is Appendix C, i.e., the earlier reference has changed the order of Appendices.

**7.** Line 264-266: This sentence is really hard to read. I guess I get its meaning (which by the way supports my claim to reformulate H0, see above) but I believe it is possible to reformulate to make it easier readable.

**Authors:** We rephrased the sentence in L. 266 to make it more comprehensive to "Again recall that these precursors are features of the low NDVI events' meteorological histories that were significantly more frequent than during any random meteorological history in the climatology."

**8.** Line 333-334: see my comment on H0 and please revise accordingly. It is rather a difference to climatology and not a relation with low NDVI events that you're testing.

**Authors:** We rephrased H0 in L. 334 accordingly: "…under the null hypothesis H_0,EV that the fraction of dry/warm periods preceding the low NDVI events (i.e., during dt) was not different from a randomly sampled meteorological history (grey shading…"

**9.** Line 348-349: The formulation used here again suggests the reformulation of H0: it is a comparison to normal conditions and not a relation with low NDVI events.

**Authors:** Yes, see answer to C5.

**10.** Line 405: To be very clear, please specify which approach you refer to here. I assume the low NDVI-approach and not the meteorological history bootstrapping approach.

**Authors:** We changed "Our approach" to "Our approach to identify low NDVI events".

**11.** Line 490: I propose to reformulate this sentence: on the one hand the anticyclone favors low NDVI in northern Europe but on the other hand it is unfavorable for low NDVI grid cells in western Europe. It is not clear how the anticyclone should be unfavorable for low NDVI. Please reword.

**Authors:** We assume that the reviewer refers to the sentence in L. 493: "So while in JJA a European-centered anticyclone can favor low NDVI grid cells in northern Europe, it might be unfavorable for low NDVI grid cells in western Europe.". As we explain in the two previous sentences, there are regions in western Europe that receive a large portion of summer precipitation during the presence of an anticyclone (see L.491-493 and references, or also L. 547). Our results show that the JJA anticyclone frequency is reduced in these regions at the time of low NDVI events. Consequently, we conclude with the above-cited sentence that in these regions, more frequent anticyclones (i.e., more JJA precipitation) are not favorable for the occurrence of a low NDVI event. A possible explanation could, we think, be that weak cyclonic activity with no or little precipitation instead of heavy convective precipitation at the western inflow of the anticyclone sustains drought conditions better, and, hence, is favorable for low NDVI events (in the given spatio-temporal context). As this is an interesting point to make, which is shown in our results and relates to precipitation patterns found in previous studies, we did not change this part of the manuscript.

**Reviewer 2**

The authors have made an excellent job in addressing the comments and the manuscript is now clearer, and the methods, analysis and results more robust. The manuscript is still heavy on acronyms, but it is probably unavoidable

The approach proposed to study low NDVI events is still very relevant for the community, as I mentioned in the first revision. The more didactic explanation of atmospheric processes and the introduction of Sections 3.2 and 3.4 are very good addition. The extension of NDVI until August 2022 results in very timely new results (summer 2022 was again hot and dry).

In summary, I strongly recommend the publication of this study.

**Authors:** Many thanks for these very positive comments on our revised manuscript.

I have only a minor comment:

**12.** Line 90 and Line 162 should read "90-day *moving* average", right?

**Authors:** Thank you for highlighting this aspect, we included the reviewer's suggestion in both passages.